# TOWARDS INFINITE-LONG PREFIX IN TRANSFORMER

## ABSTRACT

Prompting and context-based fine-tuning methods, which we call Prefix Learning, have been proposed to enhance the performance of language models on various downstream tasks. They are empirically efficient and effective, matching the performance of full parameter fine-tuning, but the theoretical understandings are limited. In this paper, we aim to address this limitation by studying their ability from the perspective of prefix length. In particular, we provide a convergence guarantee for training an ultra-long prefix in a stylized setting using the Neural Tangent Kernel (NTK) framework. Based on this strong theoretical guarantee, we design and implement an algorithm that only needs to introduce and fine-tune a few extra trainable parameters instead of an infinite-long prefix in each layer of a transformer, and can approximate the prefix attention to a guaranteed polynomial-small error. Preliminary experimental results on vision, natural language, and math data show that our method achieves superior or competitive performance compared to existing methods like full parameters fine-tuning, P-Tuning V2, and LoRA. This demonstrates our method is promising for parameter-efficient fine-tuning.

## 1 INTRODUCTION

The advent of Large Language Models (LLMs) and Vision LLMs (vLLMs) has significantly advanced the field of Artificial Intelligence (AI), with prominent examples like ChatGPT (ChatGPT, 2022), GPT-4 (Achiam et al., 2023; Bubeck et al., 2023), Claude (Claude-3, 2024), Llama (Touvron et al., 2023a;b), Gemini (Gemini, 2024), ViT (Dosovitskiy et al., 2020), DETR (Carion et al., 2020), BLIP (Li et al., 2022; 2023a), CLIP (Radford et al., 2021). They have exhibited impressive performances across a spectrum of tasks, encompassing chat systems (Maaz et al., 2023; Xu et al., 2023a; Zheng et al., 2024), text-to-image conversion (Qiao et al., 2019; Frolov et al., 2021; Zhang et al., 2023), AI mathematical inference (Hendrycks et al., 2020; Yu et al., 2023a; Yao et al., 2023), and many more. However, despite these advancements, pre-existing LLMs often fall short in specialized domains that demand a deeper understanding of professional knowledge (Tajbakhsh et al., 2016; Devlin et al., 2018; Gururangan et al., 2020; Hu et al., 2021; Sun, 2023; Kasneci et al., 2023; Li et al., 2023b; Thirunavukarasu et al., 2023; Li et al., 2024b; Wang et al., 2024). This has led to the development of fine-tuning/adaptation (Shi et al., 2022; Xu et al., 2023b; Shi et al., 2024a) methodologies aimed at enhancing the proficiency of these models in executing more specialized tasks (Mangrulkar et al., 2022). Several notable contributions in this area, such as LoRA (Low-Rank Adaptation, Hu et al. (2021)), P-Tuning (Liu et al., 2021b; 2023), and $(IA)^3$ (Liu et al., 2022), have displayed performances rivaling those of full-parameter fine-tuning techniques. This underscores the potential of these fine-tuning strategies to further refine the capabilities of Large Language Models.

Among the methods proposed, most context-based fine-tuning methods, e.g., Prompt-Tuning (Lester et al., 2021; Liu et al., 2021a), Prefix-Tuning (Li & Liang, 2021), P-Tuning (Liu et al., 2023; 2021b), use enhanced input sequences (or virtual prompt, a.k.a soft prompt) to optimize their model outputs. These methods are gaining significant interest due to their ease of implementation across various model architectures, and also prevention of catastrophic forgetting with static pre-trained parameters (Wang et al., 2023b; Sohn et al., 2023; Yang et al., 2024). We call the above approaches **Prefix Learning** since they improve the performance by optimizing a prefix matrix added to the input in each attention layer of the LLMs (see detailed formulation in Section 2).

Despite its wide use and strong empirical performance, we still have a limited understanding of why and how prefix learning operates (Wang et al., 2023a; Petrov et al., 2024a;b). One common phenomenon in prior empirical studies is that prefix learning results in better downstream performance

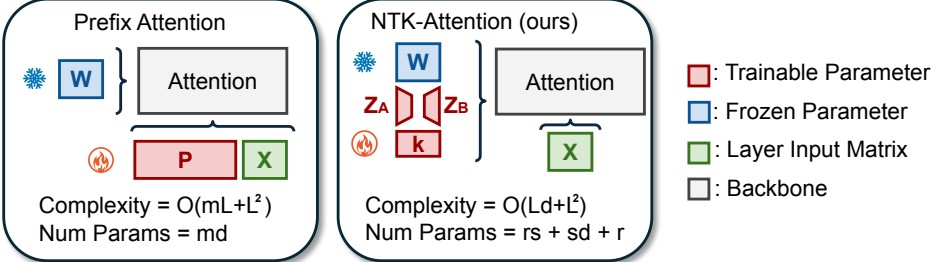

Figure 1: Illustration of existing prefix attention methods (Algorithm 1) and our NTK-Attention (Algorithm 2). Compared to the former, NTK-Attention significantly reduces the number of parameters and the time complexity. Here, $X \in \mathbb{R}^{L \times d}$ is the input of this layer, $W = [W_Q, W_K, W_V]$ is frozen weights of attention, $P \in \mathbb{R}^{m \times d}$ is the trainable prefix matrix and $Z_A \in \mathbb{R}^{r \times s}, Z_B \in \mathbb{R}^{s \times d}, k \in \mathbb{R}^r$ are the trainable parameters in our method. $L$ is the input length, $d$ the input dimension, $m$ the prefix length, and $r$ a hyperparameter in NTK-attention (i.e., the dimension of the constructed feature mapping; see Section 4). Note that $m \gg L$ and $m \gg d$, and $r = \text{poly}(d)$ (usually be chosen to $d$ or $2d$), $s \leq \lfloor d/2 \rfloor$ (low-rank of $Z_A, Z_B$) are used in our experiments.

when the prefix length increases (Lester et al., 2021; Liu et al., 2023). We call this phenomenon *scaling law in prefix learning*: the longer the prefix, the larger downstream dataset the model can fit, and thus the better performance the model would have. Then intuitively, we would like to ask:

*What happens when the prefix length is large or even tends to infinity?*

The answer to this cannot be directly figured out via empirical evaluations, since it is impractical to implement networks with ultra-long or even infinite prefixes in practice. Therefore, we first perform a theoretical analysis of prefix learning. We study the optimization of ultra-long prefix learning via the Neural Tangent Kernel (NTK) technique (Jacot et al., 2018), which has been used for analyzing overparameterized networks and thus is suitable for ultra-long prefix learning. Based on the insights gained from the analysis, we propose our method, NTK-attention, which reparameterizes prefix learning and can approximate infinite-long prefix learning using a finite number of parameters. We also conduct some empirical evaluations of our method on vision, natural language understanding, and math inference datasets to demonstrate its effectiveness.

Specifically, we have made the following contributions:

- We first perform a theoretical analysis of optimizing an ultra-long prefix in a stylized attention network; see Section 3. We consider a simplified attention network, and show that when prefix length $m$ is sufficiently large (i.e., prefix learning is sufficiently over-parameterized), the training can be analyzed via NTK, which leads to our theoretical guarantee of convergence to small errors. This also provides theoretical support for scaling law in prefix learning.

- We then propose our NTK-Attention (Algorithm 2), motivated by the above strong theoretical guarantee; see Section 4. Our method approximates existing prefix attention (Algorithm 1) by utilizing three trainable parameters $Z_A$, $Z_B$ and $k$, to replace the parameter in prefix attention (the prefix matrix $P$). This allows scaling the prefix length without large memory usage and computational time that increases with the prefix length. It reduces the computation complexity from $O(mL)$ to $O(L^2)$, where $L$ is the input length and $m$ is the prefix length. See Figure 1 for an illustration.

- We further conduct experiments on vision, language and math datasets to verify our theoretical results; see Section 5. The experiments include (1) a comparison among our NTK-Attention, full parameters fine-tuning, and LoRA on CIFAR-100, Food-101 and Tiny-Imagenet datasets with the same pretrained ViT backbone; (2) a comparison among our NTK-Attention, P-Tuning V2, and LoRA on SuperGLUE, WikiText-103, Penn TreeBank and LAMBADA datasets with the same pretrained ChatGLM3-6B and OPT-{125M, 350M, 1.3B, 2.7B, 6.7B} family; (3) a comparison among our NTK-Attention and LoRA on GSM8K and MATH datasets with supervised fine-tune pretrained models LLAMA-3.2; (4) an ablation study to validate sensitivity of hyper-parameters in NTK-Attention; (5) a comparison of the computational costs between our method and standard prefix learning on random data. The empirical results show that on average our NTK-Attention

method achieves better performance than the competitors. For example, on SuperGLUE datasets, it achieves an average accuracy that is 1.07% higher than LoRA and 12.94% higher than P-Tuning V2. It is also observed that our method maintains low time and memory costs while those of prefix learning scales with prefix length. The experimental results demonstrate that our method is effective and efficient and supports our theoretical analysis.

## 1.1 RELATED WORK

**Prefix Learning.** Prefix Learning (Lester et al., 2021; Ding et al., 2021; Wang et al., 2022b; Zhou et al., 2022; Liu et al., 2021a; Petrov et al., 2024a; Wu et al., 2023), including Prompt-Tuning (Lester et al., 2021), Prefix-Tuning (Li & Liang, 2021), P-Tuning (Liu et al., 2023; 2021b), Reweighted In-Context Learning (RICL) (Chu et al., 2023) and so on, is proposed to enhance the performance of language models on the downstream tasks and to reduce the costs of computational resources of fine-tuning the whole model. Those methods optimize task-specific prompts for downstream task improvement. On the other hand, besides the Parameter-Efficient-Fine-Tuning (PEFT) approaches (Mangrulkar et al., 2022) we mentioned above, Retrieval Augmented Generation (RAG) (Lewis et al., 2020; Jiang et al., 2023; Gao et al., 2023b) and Chain-of-Thought (CoT) prompting (Wei et al., 2022b; Wang et al., 2022a; Fu et al., 2022) can also be considered as prefix learning. We conclude all these works to an optimization problem that improves the prefix based on task-specific measurements.

**Neural Tangent Kernel.** Neural Tangent Kernel (NTK) (Jacot et al., 2018) studies the gradient flow of neural networks in the training process. They showed neural networks are equivalent to Gaussian processes in the infinite-width limit at initialization. A bunch of works has explained the strong performance and the learning ability of neural networks at over-parameterization, such as (Li & Liang, 2018; Du et al., 2019; Song & Yang, 2019; Allen-Zhu et al., 2019; Wei et al., 2019; Bietti & Mairal, 2019; Lee et al., 2020; Chizat & Bach, 2020; Shi et al., 2021; Zhou et al., 2021; Seleznova & Kutyniok, 2022; Gao et al., 2023a; Li et al., 2024a; Shi et al., 2024c) and many more. Furthermore, Arora et al. (2019) gave the first exact algorithm on computing Convolutional NTK (CNTK), Alemohammad et al. (2020) proposed Recurrent NTK, and Hron et al. (2020) presented infinite attention via NNGP and NTK for attention networks. These works have demonstrated advanced performance by utilizing NTK in different neural network architectures. In particular, Malladi et al. (2023) have studied the training dynamic of fine-tuning LLMs via NTK and confirmed the efficiency of such methods.

**Theory of Understanding Large Language Models.** Since the complicated transformer-based architecture and stochastic optimization process of LLMs lead the study of their behaviors to be a challenge, analyzing LLMs through some theoretical guarantee helps in providing insights to improve and design the next generation of AI systems. This topic includes efficient LLMs (Alman & Song, 2023; 2024a;b; Han et al., 2024; Kacham et al., 2023; Addanki et al., 2023; Deng et al., 2024; Shi et al., 2024b), optimization of LLMs (Deng et al., 2023; Li et al., 2024a), white-box transformers (Yu et al., 2023b;c; Ferrando et al., 2024; Pai et al., 2024), analysis of emergent abilities of LLMs (Brown et al., 2020; Wei et al., 2022a; Allen-Zhu & Li, 2023a;b;c; 2024), etc. Especially, (Alman & Song, 2023) proved that the hardness of fast attention can be achieved within $n^{1+o(1)}$ times executions, one effective way is to construct a high-order polynomial mapping based on Taylor expansion of the exponential function $\exp(\cdot)$, and it inspired the design of our NTK-Attention method.

## 2 PRELIMINARY: PREFIX LEARNING

In this section, we provide the detailed formulation for prefix learning, which optimizes prefix matrices in the attention layers of transformer-based LLMs. Focusing on one single-layer attention network, we formalize it as a regression problem that optimizes a prefix matrix.

**Prefix for Attention Computation.** Let $X \in \mathbb{R}^{L \times d}$ be an input matrix to the attention network, where $L$ and $d$ are the input length and dimension. Prefix learning freezes the query, key, and value parameter matrices in the pretrained attention network (denoted as $W_Q, W_K, W_V \in \mathbb{R}^{d \times d}$, respectively). It introduces a trainable prefix matrix $P \in \mathbb{R}^{m \times d}$, which stands for $m$ virtual token vectors (or soft prompt). Let $S := \begin{bmatrix} P \\ X \end{bmatrix}$ be the concatenation of the prefix and the input. Then the query, key, and value matrices are given by $Q := XW_Q, K_P := SW_K, V_P := SW_V$, and the

attention with the prefix is:

$$\mathsf{PrefixAttn}(X, P) := \mathsf{Softmax}(\frac{QK_P^\top}{\sqrt{d}}) \cdot V_P \quad \in \mathbb{R}^{L \times d}. \tag{1}$$

Here Softmax is the row-wise softmax computation, i.e., for any $d_1, d_2 > 0$, $Z \in \mathbb{R}^{d_1 \times d_2}$, $\mathsf{Softmax}(Z) := [\mathsf{S}(Z_{1,*}), \mathsf{S}(Z_{2,*}), \cdots, \mathsf{S}(Z_{d_1,*})]^\top \in \mathbb{R}^{d_1 \times d_2}$ where $\mathsf{S}(z) := \frac{\exp(z)}{\langle \exp(z), \mathbf{1}_{d_2} \rangle} \in \mathbb{R}^{d_2}$ for any $z \in \mathbb{R}^{d_2}$. The attention computation with prefix is summarized in Algorithm 1.

**Prefix Learning.** The prefix $P$ is trained on a fine-tuning dataset. Denote the dataset as $\mathcal{D}_{\mathrm{pl}} = \{(X_i, Y_i)\}_{i=1}^n$ where $n$ is the dataset size, and $X_i, Y_i \in \mathbb{R}^{L \times d}$. Let $\ell(\cdot, \cdot)$ denote the loss function for the specific task (e.g., prompting, context-based fine-tuning, etc.). The training objective of prefix learning is then:

$$\min_{P \in \mathbb{R}^{m \times d}} \mathcal{L}_{\mathrm{pl}}(W) := \sum_{i=1}^n \ell(\mathsf{PrefixAttn}(X_i, P), Y_i). \tag{2}$$

**Scaling Prefix Length.** A rich line of studies (Liu et al., 2021b; Lester et al., 2021; Liu et al., 2023; Reynolds & McDonell, 2021; Arora et al., 2022; Brown et al., 2020; Dong et al., 2022; Shi et al., 2023; Von Oswald et al., 2023; Xu et al., 2024; Fu et al., 2022; Agarwal et al., 2024; Kaplan et al., 2020; Hoffmann et al., 2022) have reported a common observation that as the prefix length increases, the model's ability to master complex skills also improves. Specifically, the performance of fine-tuned models is enhanced when the prefix length grows within a certain range. A similar trend is observed in prompting methods and in-context learning (ICL), where longer and more complex prompts lead to better inference abilities in LLMs, and providing more examples in ICL results in improved LLM performance. We summarize this as the *scaling law in prefix learning*: the longer the prefix length for fine-tuning, the larger dataset the model can fit, thus, the more complicated skill it can master. This motivates investigating prefix learning with long prefixes.

In this paper, we examine the implications of using a significantly large prefix length, denoted as $m \gg L$ and $m \gg d$, which is prevalent across various prompt-based methods. The primary objective of Prefix Learning is to enhance the LLMs' outputs by identifying an advanced prefix during the generation process. For instance, the search for optimal example pairs to improve ICL (Nguyen & Wong, 2023) and the development of prompt engineering tailored for agent frameworks to address specific task requirements (dif, 2024) often necessitate the use of exceptionally long prefixes. Moreover, given the modern application demands related to long-context scenarios, optimizing previous tokens to improve next-token prediction can be framed as a prefix optimization problem. Thus, a thorough investigation into the optimization of infinitely long prefixes is essential for understanding the theoretical significance of the prefix matrix in LLMs.

## 3 THEORETICAL ANALYSIS OF PREFIX LEARNING VIA NTK

In this section, we explore the theory behind prefix learning with ultra-long prefixes. We first present the theoretical setting for a simplified model $\mathsf{F}(W, x, a)$ in Section 3.1, and then in Section 3.2 introduce the formal definition of the neural tangent kernel for our problem and confirm the convergence of the kernel matrices needed for performing NTK analysis. In Section 3.3 we state the main result, a convergence guarantee of prefix learning in this setting (the detailed analysis is in the appendix).

### 3.1 PROBLEM SETUP

**Model.** The attention computation with prefix $P$ given is by Eq. (1). Since the attention parameters are fixed, it can be rewritten as $\mathsf{Softmax}(\widetilde{X}P^\top + b) \cdot \begin{bmatrix} PW_V \\ b' \end{bmatrix}$ where $\widetilde{X} = XW_Q W_K^\top / \sqrt{d}, b = XW_Q W_K^\top X^\top / \sqrt{d}$, and $b' = XW_V$. We view the input sequence as one token (i.e., assuming $L = 1$) such that the input $X$ and thus $\widetilde{X}$ become vectors, simplifying our analysis from matrix-form calculations to vector-form. Furthermore, ignoring the bias terms, and introducing notations $x := \widetilde{X}^\top$ and $W = P^\top$, the attention simplifies to $\mathsf{Softmax}(xW) \cdot W^\top W_V = \frac{\sum_{r \in [m]} \exp(w_r^\top x) w_r W_V}{\sum_{r \in [m]} \exp(w_r^\top x)}$ where

$w_r$ is the $r$-th column of $W$. We therefore consider the following two-layer attention model:

$$\mathsf{F}(W, x, a) := m \frac{\sum_{r \in [m]} \exp(w_r^\top x) w_r a_r}{\sum_{r \in [m]} \exp(w_r^\top x)} \tag{3}$$

with the hidden-layer weights $W = [w_1, w_2, \ldots, w_m] \in \mathbb{R}^{d \times m}$ and output-layer weights $a = [a_1, a_2, \ldots, a_m]^\top \in \mathbb{R}^m$. Such a stylized setting has been widely used for studying the learning behavior of transformer-based models (Deng et al., 2023; Chu et al., 2023; 2024; Li et al., 2024a), and they gave detailed derivations and guarantees for its connection to attention. Furthermore, our analysis can be extended to models with bias terms and matrix inputs rigorously.

**Training.** Consider a training dataset $\mathcal{D} = \{(x_i, y_i)\}_{i=1}^n$ where the $i$-th data point $(x_i, y_i) \in \mathbb{R}^d \times \mathbb{R}^d$. Assume $\|x_i\|_2 \leq 1$ and $\|y_i\|_2 \leq 1$ for any $i \in [n]$. The training loss is measured by the $\ell_2$ norm of the difference between model prediction $\mathsf{F}(W, x_i, a)$ and ideal output vector $y_i$. Formally, the training objective is:

$$\mathcal{L}(W) := \frac{1}{2} \sum_{i=1}^n \|\mathsf{F}(W, x_i, a) - y_i\|_2^2. \tag{4}$$

The weights $W$ are initialized to $W(0)$ as follows: $\forall r \in [m]$, sample $w_r(0) \sim \mathcal{N}(0, I_d)$ independently. For output-layer $a$, randomly sample $a_r \sim \mathsf{Uniform}\{-1, +1\}$ independently for $r \in [m]$ and fix $a$ during the training. Then use gradient descent (GD) to update the trainable weights $W(t)$ with a fixed learning rate $\eta > 0$. Then for $t \geq 0$:

$$W(t + 1) := W(t) - \eta \cdot \nabla_W \mathcal{L}(W(t)). \tag{5}$$

### 3.2 Neural Tangent Kernel

Here, we give the formal definition of NTK in our analysis, which is a kernel function that is driven by hidden-layer weights $W(t) \in \mathbb{R}^{d \times m}$. To present concisely, we first introduce an operator function in the following. For all $r \in [m]$, $k \in [d]$ and $i \in [n]$:

$$v_{k,r}(W) := W_{k,r} \cdot a_r \cdot \mathbf{1}_m - W_{k,*} \circ a \in \mathbb{R}^m, \quad \mathcal{G}_{i,r}(W) := m\mathsf{S}_r(W^\top x_i) \cdot \langle v_{k,r}, \mathsf{S}(W^\top x_i) \rangle \in \mathbb{R}$$

where $\mathsf{S}(z) = \frac{\exp(z)}{\langle \exp(z), \mathbf{1}_m \rangle} \in \mathbb{R}^m$ for any $z \in \mathbb{R}^m$, and $\circ$ denotes element-wise product.

Then, we define the kernel matrix $H(W(t))$ as an $nd \times nd$ Gram matrix, where its $(k_1, k_2)$-th block is an $n \times n$ matrix for $k_1, k_2 \in [d]$, and the $(i, j)$-th entry of the block is:

$$[H_{k_1,k_2}]_{i,j}(W(t)) := \frac{1}{m} x_i^\top x_j \sum_{r=1}^m \mathcal{G}_{i,r}(W(t)) \cdot \mathcal{G}_{j,r}(W(t)).$$

We can show that $\mathsf{S}_r(W^\top x_i) = O(\frac{1}{m})$ and $\langle v_{k,r}, \mathsf{S}(W^\top x_i) \rangle = O(1)$, thus $\mathcal{G}_{i,r}(W)$ is $O(1)$. Then $H(W)$ is close to $H^* := H(W(0))$ when $W$ is close to $W(0)$. This kernel convergence is the key needed for the NTK analysis and is formalized below (details in Appendix H).

**Lemma 3.1** (Kernel convergence, informal version of Lemma H.3)**.** *For $\delta \in (0, 0.1)$ and $B = \max\{C\sigma\sqrt{\log(nd/\delta)}, 1\}$. Let $\widetilde{W} = [\widetilde{w}_1, \cdots, \widetilde{w}_m] \in \mathbb{R}^{d \times m}$ and satisfy $\|\widetilde{w}_r - w_r(0)\|_2 \leq R$ for any $r \in [m]$, where $R$ is some constant in $(0, 0.01)$. Define $\widetilde{H} := H(\widetilde{W}) \in \mathbb{R}^{nd \times nd}$. Then with probability at least $1 - \delta$, we have $\|H^* - \widetilde{H}\| \leq 8R\sqrt{nd} \cdot \exp(22B)$.*

### 3.3 Main Result: Loss Convergence Guarantee

**Assumption on NTK $H^*$.** In the NTK analysis framework for the convergence of training neural networks, one widely-used and mild assumption is that $H^*$ is a positive definite (PD) matrix, i.e., its minimum eigenvalue $\lambda := \lambda_{\min}(H^*) > 0$ (Du et al., 2019; Oymak & Soltanolkotabi, 2020). With this, our main result is presented as follows.

**Theorem 3.2** (Main result, informal version of Theorem J.2)**.** *Assume $\lambda > 0$. For any $\epsilon, \delta \in (0, 0.1)$, $B = \max\{C\sigma\sqrt{\log(nd/\delta)}, 1\}$, $m = \lambda^{-2}\operatorname{poly}(n, d, \exp(B))$, $\eta = \lambda m^{-1}/\operatorname{poly}(n, d, \exp(B))$ and $\widehat{T} = \Omega((m\eta\lambda)^{-1}\log(nd/\epsilon))$. Then, after $\widehat{T}$ iterations of update (Eq. (5)), we have $\mathcal{L}(W(\widehat{T})) \leq \epsilon$ holds with probability at least $1 - \delta$.*

*Proof sketch of Theorem 3.2.* We use the math induction to show that the weight $w$ perturbation is small so that the loss landscape is almost convex around the network's initialization in Lemma J.3, Lemma J.4 and Lemma J.5, which are based on Lemma 3.1. Then, we conclude the results by standard convex optimization analysis. See the complete proof in Appendix J.1. □

**Discussion.** Theorem 3.2 mainly describes the following fact for any dataset with $n$ data points. After initializing the prefix matrix from a normal distribution, assuming the minimum eigenvalue of NTK $\lambda > 0$, setting $m$ to be a large enough value so that the network is sufficiently over-parameterized. Then with proper learning rate, the loss can be minimized in finite training time to an arbitrarily small error $\epsilon$. Corresponding to the real-world implementation, it explains that adequately long prefix learning can master downstream tasks when fine-tuning LLMs. Furthermore, it also helps us understand the working mechanism of prefix learning, inspiring us to explore the direction of using ultra-long prefixes.

Now we connect our theory to the *scaling law in prefix learning*. Following (Kaplan et al., 2020), we focus on the relationship between the loss and the computational cost. We prove that the loss decreases with the computational cost scaling up, providing a theoretical confirmation about the scaling law in prefix learning.

**Proposition 3.3** (Scaling Law in Prefix Learning). *We define* $\mathsf{N} := O(md)$ *as the number of parameters,* $\mathsf{D} := O(n)$ *as the size of training dataset,* $\mathsf{C}_{\mathrm{cpt}} := O(\mathsf{N}\mathsf{D}T)$ *as the total compute cost, and* $\alpha := nd$. *We choose* $T$ *as Theorem 3.2, then the loss of training, denotes* $\mathsf{L}$, *satisfies:*

$$\mathsf{L} \approx \frac{\alpha}{[\exp(\eta\lambda\mathsf{C}_{\mathrm{cpt}})]^{\frac{1}{\alpha}}}$$

*Proof sketch of Proposition 3.3.* This proof follows from the definitions of $\mathsf{C}_{\mathrm{cpt}}$, $\mathsf{N}$, $\mathsf{D}$ and $\alpha$ and Theorem 3.2. □

Proposition 3.3 shows that the training loss of the prefix learning converges exponentially as we increase the computational cost $\mathsf{C}_{\mathrm{cpt}}$, which primarily depends on the number of parameters and the training time in prefix learning, further indicating a possible relationship for formulating scaling law in prefix learning.

# 4 NTK-ATTENTION: APPROXIMATE INFINITE-LONG PREFIX ATTENTION

The preceding section discussed the convergence guarantee of training sufficiently long prefixes $P$ in attention networks (recall that the trainable parameter $W$ is just $P^\top$). This strong theoretical property inspires us to scale up the prefix length $m$. However, such prefix learning (Algorithm 1) necessitates a time complexity of $O(mLd + L^2d)$ in each layer of the model, this is impractical due to a large $m$.

This section proposes an approximate algorithm to make long prefix learning practical. Our algorithm, NTK-Attention, is designed to output an approximation of $\mathsf{PrefixAttn}(X, P)$ (Eq. (1)) in time within $O(L^{1+o(1)})$ and without using the long prefix matrix $P$. We present the derivation and motivation of our algorithm in Section 4.1, formalize the NTK-Attention algorithm in Section 4.2, and provide an approximation guarantee in Section 4.3.

## 4.1 DERIVATION: REPLACING PREFIX $P$ WITH TRAINABLE PARAMETERS $Z, k$

There exists a wealth of attention approximation algorithms capable of executing attention computations within $n^{1+o(1)}$ time (Han et al., 2024; Liang et al., 2024a;b). However, our focus lies predominantly with the polynomial method (Tsai et al., 2019; Katharopoulos et al., 2020; Alman & Song, 2023; 2024b). This method has exhibited exceptional performance in terms of both time and space complexity through the use of a streaming algorithm.

**Polynomial method.** In the context of attention networks, the query, key, and value state matrices, denoted as $Q, K, V \in \mathbb{R}^{L \times d}$, are assumed to have all entries bounded (Alman & Song, 2023). Under this condition, the polynomial method first constructs a linear mapping $\phi : \mathbb{R}^d \to \mathbb{R}^r$, where

$r = \text{poly}(d)$ (Alman & Song, 2023), and it satisfies the following relation ($i, j \in [L]$, $Q_i, K_j \in \mathbb{R}^d$ represent the $i$-th row of $Q$ and the $j$-th row of $K$ respectively):

$$\phi(Q_i)^\top \phi(K_j) \approx \exp(Q_i^\top K_j / \sqrt{d}). \tag{6}$$

Here, the mapping $\phi(\cdot)$ is constructed based on the Taylor expansion of the exponential function, and the larger value of $r \geq d$ would bring the approximation (Eq. (6)) with a smaller error. This is guaranteed by Lemma 3.4 in Alman & Song (2023), refer to a copy in Lemma K.7. The $i$-th row of the approximate attention (denoted as $\text{PolyAttn}_i \in \mathbb{R}^{1 \times d}$) then can be computed as follows:

$$\text{PolyAttn}_i := \frac{\phi(Q_i)^\top \sum_{j=1}^L \phi(K_j) V_j^\top}{\phi(Q_i)^\top \sum_{j=1}^L \phi(K_j)} \in \mathbb{R}^{1 \times d}, \forall i \in [L].$$

Now recall that given an input matrix $X \in \mathbb{R}^{L \times d}$, thus, $Q = XW_Q$, and we have $[K_P, V_P] = \begin{bmatrix} P \\ X \end{bmatrix} \cdot$ $[W_K, W_V] = \begin{bmatrix} PW_K & PW_V \\ XW_K & XW_V \end{bmatrix}$. Let $K_C := PW_K, V_C := PW_V \in \mathbb{R}^{m \times d}$ and $K := XW_K, V := XW_V \in \mathbb{R}^{L \times d}$. We thus expand the $i$-th row of the prefix attention, $\text{PrefixAttn}_i(X, P) \in \mathbb{R}^{1 \times d}$ as:

$$\begin{aligned} \text{PrefixAttn}_i(X, P) &= \frac{\exp(Q_i^\top K^\top / \sqrt{d}) V + \exp(Q_i^\top K_C^\top / \sqrt{d}) V_C}{\exp(Q_i^\top K^\top / \sqrt{d}) \mathbf{1}_L + \exp(Q_i^\top K_C^\top / \sqrt{d}) \mathbf{1}_m} \\ &\approx \frac{\exp(Q_i^\top K^\top / \sqrt{d}) V + \phi(Q_i)^\top Z}{\exp(Q_i^\top K^\top / \sqrt{d}) \mathbf{1}_n + \phi(Q_i)^\top k} \end{aligned}$$

where

$$Z = \sum_{j=1}^m \phi(K_{C,j}) V_{C,j}^\top \in \mathbb{R}^{r \times d}, \qquad k = \sum_{j=1}^m \phi(K_{C,j}) \in \mathbb{R}^r. \tag{7}$$

Here, the first step explicitly computes the softmax function, and the second step holds since replacing $\exp(Q_i^\top K^\top / \sqrt{d})$ by Eq. (6), which is $\exp(Q_i^\top K_{C,j}^\top / \sqrt{d}) \approx \phi(Q_i)^\top \phi(K_{C,j}), \forall j \in [m]$.

Therefore, checking the training process of $P$, we observe that $P$ is updating iff $Z$ and $k$ are updating. Hence, we can replace $P$ by utilizing **trainable parameters** $Z$ and $k$ in Eq. (7) to re-parameterize the prefix attention. This is the key to how NTK-Attention approximates prefix attention without a large number of parameters.

## 4.2 ALGORITHM

To present our algorithm, based on $\phi$, we define: $\Phi(A) = [\phi(A_{1,*}), \cdots, \phi(A_{L,*})]^\top \in \mathbb{R}^{L \times r}, \forall A \in \mathbb{R}^{L \times d}$. Below we present our NTK-Attention method in Algorithm 2, and for comparison also present the traditional prefix attention for prefix learning in Algorithm 1.

**Implementation Detail of $\phi$.** In order to find a balance between approximation and efficient computation of NTK-Attention, we use the first-order polynomial method. In particular, we choose $r = d$, and the function $\phi$ is given by $\phi(z) := d^{-\frac{1}{4}} \cdot (z \circ \mathbf{1}_{z \geq \mathbf{0}_d} + \exp(z) \circ \mathbf{1}_{z < \mathbf{0}_d}) + \mathbf{1}_d \in \mathbb{R}^d, \forall z \in \mathbb{R}^d$, where $\mathbf{1}_{z \geq \mathbf{0}_d} \in \mathbb{R}^d$ is an indicative vector and its $i$-th entry for $i \in [d]$ equals 1 only when $z_i \geq 0$, and 0 otherwise.

**Initialization, Approximation and Training of $Z$ and $k$.** In Section 3.1, we initialize the parameter $W = P^\top$ by $w_r(0) \sim \mathcal{N}(0, I_d)$ for $r \in [m]$. Since the pretrained weights $W_Q, W_K, W_V \in \mathbb{R}^{d \times d}$ are known, the initialization of $Z$ and $k$, denotes $Z(0)$ and $k(0)$, can then be computed by Eq. (7) using $P(0) = W(0)^\top$. However, consider that $Z$ caches $rd$ parameters for $r = \text{poly}(d)$, which is insufficient parameter-efficient. In response to it, we choose $s \leq \lfloor d/2 \rfloor$ as an appropriately small integer, then $Z(0) \approx Z_A(0) \cdot Z_B(0)$ is decomposed into two low-rank matrices $Z_A(0) \in \mathbb{R}^{r \times s}, Z_B(0) \in \mathbb{R}^{s \times d}$. For training, let $g_{Z_A}(t) \in \mathbb{R}^{r \times s}, g_{Z_B}(t) \in \mathbb{R}^{s \times d}$ and $g_k(t) \in \mathbb{R}^r$ denote the gradients of $Z_A(t), Z_B(t)$ and $k(t)$ at time $t$, and $\eta$ denote the learning rate. Then the update rule is:

$$Z_A(t+1) := Z_A(t) - \eta \cdot g_{Z_A}(t), Z_B(t+1) := Z_B(t) - \eta \cdot g_{Z_B}(t), k(t+1) := k(t) - \eta \cdot g_k(t).$$

**Number of Trainable Parameters.** Since given $r$ and $s$ as two hyper-parameters in NTK-Attention, for each attention layer in transformer-based architecture, we denote $\beta := \frac{r}{d}$, then the number of

Table 1: Performance of different fine-tuning methods on the SuperGLUE datasets. The base model is ChatGLM3-6B. The methods include P-Tuning V2, LoRA, and our NTK-Attention method. The metric on these datasets is accuracy (measured in %). The best score on each dataset is **boldfaced**.

| Method | Num Params | Task | | | | | Average |
|---|---|---|---|---|---|---|---|
| | | BoolQ | CB | Copa | MultiRC | RTE | |
| P-Tuning V2 $m = 1$ | 0.12M | $65.69_{\pm 0.32}$ | $67.06_{\pm 0.37}$ | $52.00_{\pm 1.00}$ | $53.59_{\pm 0.28}$ | $65.97_{\pm 0.22}$ | $60.86_{\pm 0.44}$ |
| P-Tuning V2 $m = 10$ | 1.15M | $66.67_{\pm 0.23}$ | $74.07_{\pm 0.00}$ | $54.00_{\pm 0.00}$ | $54.17_{\pm 0.71}$ | $66.55_{\pm 0.25}$ | $63.10_{\pm 0.24}$ |
| P-Tuning V2 $m = 100$ | 11.47M | $69.42_{\pm 0.02}$ | $74.54_{\pm 0.47}$ | $64.50_{\pm 0.50}$ | $61.62_{\pm 2.28}$ | $76.77_{\pm 0.83}$ | $69.37_{\pm 0.82}$ |
| P-Tuning V2 $m = 200$ | 22.94M | $67.51_{\pm 0.15}$ | $70.11_{\pm 0.28}$ | $60.00_{\pm 0.50}$ | $58.37_{\pm 0.91}$ | $70.83_{\pm 0.44}$ | $65.36_{\pm 0.46}$ |
| LoRA $r' = 8$ | 3.67M | $\mathbf{76.52}_{\pm 0.10}$ | $90.23_{\pm 0.39}$ | $86.50_{\pm 0.50}$ | $65.09_{\pm 0.41}$ | $\mathbf{87.76}_{\pm 0.37}$ | $81.24_{\pm 0.35}$ |
| NTK-Attention (ours), $r = 128, s = 16$ | 3.78M | $75.06_{\pm 0.12}$ | $\mathbf{96.04}_{\pm 0.84}$ | $\mathbf{88.00}_{\pm 2.00}$ | $65.85_{\pm 0.33}$ | $86.59_{\pm 0.52}$ | $\mathbf{82.31}_{\pm 0.76}$ |

trainable parameters could be computed by $(\beta s + \beta + s)d$ where integer $\beta \geq 1$ and $s \leq \lfloor d/2 \rfloor$. This is more flexible when adjusting the practical efficiency needs. For LoRA with its hyper-parameter $r' \leq \lfloor d/2 \rfloor$, where $r'$ is the rank number used for approximation, its number of trainable parameters is $4r'd$ and for prefix attention with its hyper-parameter $m \geq 1$, its number of trainable parameters is $md$ in each attention layer. By choosing $(\beta s + \beta + s) \leq 4r'$, the higher efficiency of NTK-Attention compared to LoRA will be satisfied.

---

**Algorithm 1** Prefix Attention

> **Input:** Input matrix $X \in \mathbb{R}^{L \times d}$
> **Parameters:** Frozen query, key and value weights $W_Q, W_K, W_V \in \mathbb{R}^{d \times d}$, trainable prefix matrix $P \in \mathbb{R}^{m \times d}$
> **Output:** Exact output $\text{Attn} \in \mathbb{R}^{L \times d}$
> 1: **procedure** PREFIXATTEN($X$)
> 2:    $S \leftarrow \left[ P^\top, X^\top \right]^\top$
> 3:    $Q, K_P, V_P \leftarrow XW_Q, SW_K, SW_V$
> 4:    $A \leftarrow \exp(QK_P^\top / \sqrt{d})$
> 5:    $D \leftarrow \text{diag}(A \mathbf{1}_{m+L})$
> 6:    **return** $D^{-1} A V_P$
> 7: **end procedure**

**Algorithm 2** NTK-Attention (w/o low-rank)

> **Input:** Input matrix $X \in \mathbb{R}^{L \times d}$
> **Parameters:** Frozen query, key and value weights $W_Q, W_K, W_V \in \mathbb{R}^{d \times d}$, trainable weights $Z \in \mathbb{R}^{r \times d}$ and $k \in \mathbb{R}^r$
> **Output:** Approx output $T \in \mathbb{R}^{L \times d}$
> 1: **procedure** NTK-ATTEN($X$)
> 2:    $Q, K, V \leftarrow XW_Q, XW_K, XW_V,$
> 3:    $\widehat{A} \leftarrow \exp(QK^\top / \sqrt{d})$
> 4:    $\widehat{D} \leftarrow \text{diag}(\widehat{A} \mathbf{1}_L + \Phi(Q)k)$
> 5:    $T \leftarrow \widehat{D}^{-1}(\widehat{A}V + \Phi(Q)Z)$
> 6:    **return** $T$
> 7: **end procedure**

---

### 4.3 ERROR BOUND AND COMPLEXITY REDUCTION

Introducing an ultra-long prefix matrix $P \in \mathbb{R}^{m \times d}$ to satisfy the conditions in Theorem J.2 requires $md$ parameters for $m \geq \Omega(\lambda^{-2} \text{poly}(n, d, \exp(B)))$, while it also bring a $O(m(m + L)d)$ time complexity to compute Algorithm 1. Our NTK-Attention relieve this by replacing $P$ with $Z$ and $k$, where we state our theoretical guarantee as follows:

**Theorem 4.1** (Error bound with reduced time complexity, informal version of Theorem K.2). *Let $m$ denote the prefix length. Given an input matrix $X \in \mathbb{R}^{L \times d}$ and prefix matrix $P \in \mathbb{R}^{m \times d}$, we denote $Q = XW_Q$, $K_C = PW_K$ and $V_C = PW_V$. If the condition Eq. (7), $\|Q\|_\infty \leq o(\sqrt{\log m}), \|K_C\|_\infty \leq o(\sqrt{\log m}), \|V_C\|_\infty \leq o(\sqrt{\log m})$ and $d = O(\log m)$ holds, then Algorithm 2 outputs a matrix $T \in \mathbb{R}^{L \times d}$ within time complexity of $O(L^2 d)$ that satisfies:*

$$\|T - \text{PrefixAttn}(X, P)\|_\infty \leq 1 / \text{poly}(m). \tag{8}$$

Furthermore, if we replace the original attention operation (attention computation on input $X$ with $K = XW_K$ and $V = XW_V$) with fast attention algorithms like HyperAttention (Han et al., 2024), then NTK-Attention can be even more efficient, achieving Eq. (8) within complexity $O(L^{1+o(1)}d)$ (see Corollary K.3 for proofs).

## 5 EMPIRICAL EVALUATIONS

In this section, we evaluate our method NTK-Attention on natural language understanding, math inference, and fine-grained image classification tasks. All our experiments use the Huggingface (Wolf

et al., 2019) trainer with AdamW optimizer (Kingma & Ba, 2014), and all optimizer hyper-parameters are set to the defaults. We provide more details in Appendix B.

**Evaluation on Natural Language Understanding Datasets.** In this experiment, we utilize five binary classification datasets in SuperGLUE (Wang et al., 2019) for evaluation: the BoolQ, CB, Copa, MultiRC, and RTE datasets. We use a pretrained LLM ChatGLM3-6B (Zeng et al., 2022; Du et al., 2022) as the base model. For comparison, we choose P-Tuning V2 (Liu et al., 2023; 2021b) which is a standard prefix learning method, and choose LoRA (Hu et al., 2021) which is a popular parameter-efficient fine-tuning method often achieving state-of-the-art. P-Tuning V2 uses different lengths of virtual prefix $\{1, 10, 100, 200\}$, and LoRA uses rank $r' = 8$. We choose $r = 128$ (the dimension of each head of ChatGLM3-6B) and $s = 16$ for our NTK-Attention.

The results are provided in Table 1. Our NTK-Attention method achieves much higher performance than P-Tuning V2. Interestingly, as $m$ increases, the performance of P-Tuning V2 also improves, which is consistent with our analysis. Our analysis also suggests that NTK-Attention approximates ultra-long prefix learning and thus can perform better than P-Tuning V2. The experimental results also show that NTK-Attention achieves better performance than LoRA on CB, Copa,

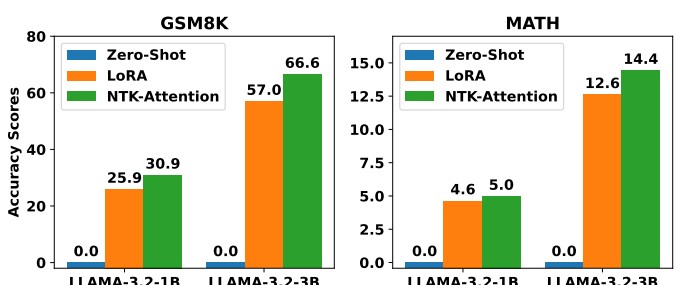

Figure 2: Compare our results with LoRA and Zero-Shot on Math inference datasets. The $y$-axis is the accuracy.

and MultiRC datasets, and achieves better average performance over all the datasets. These results show that NTK-Attention can be a promising efficient fine-tuning method.

**Evaluation on Language Modeling Tasks.** In this experiment, we focus on the scalability of NTK-Attention on a family of language models of different sizes, the OPT family with the model sizes 125M, 350M, 1.3B, 2.7B and 6.7B (Zhang et al., 2022). We introduce three text datasets, which are WikiText-103 (Merity et al., 2016), Penn TreeBank (Marcus et al., 1993), and LAMBADA (Paperno et al., 2016), to compare the scalability of NTK-Attention with LoRA (Hu et al., 2021) and P-Tuning V2 (Liu et al., 2023; 2021b). As we choose $r' = 8$ for LoRA, $m = 32$ for P-Tuning V2, and $r = 2d$ and $s = 10$ for our NTK-Attention, the numbers of trainable parameters are aligned to the same as $32d$ for each attention layer. The results are stated in Table 3, which shows the improvement of NTK-Attention compared to baselines when scaling the model size.

**Evaluation on Math Inference Datasets.** In order to thoroughly verify the effectiveness of NTK-Attention, we conduct experiments on the math inference task, which includes GSM8K (Cobbe et al., 2021) and MATH (Hendrycks et al., 2021) datasets. These are considered as fair benchmarks to test the complex capability of LLMs. We follow Yu et al. (2023a) to supervised fine-tune two pretrained models LLAMA-3.2-1B and LLAMA-3.2-3B (Touvron et al., 2023a;b) with dataset MetaMathQA (Yu et al., 2023a). We state our results in Figure 2, and we use accuracy scores for counting the matched answers for evaluation. As we can see, our NTK-attention ($r = d, s = 16$) is better than the two baselines, LoRA and Zero-Shot, where LoRA uses $r' = 16$ for LLAMA-3.2-1B and $r' = 32$ for LLAMA-3.2-3B.

**Evaluation on Vision Datasets.** We evaluate the method on three image classification datasets: CIFAR-100 (Krizhevsky et al., 2009), Food-101 (Bossard et al., 2014), and Tiny-Imagenet (mn-moustafa, 2017). The base model to be fine-tuned on these datasets is ViT-Base (Dosovitskiy et al., 2020) that is pretrained on the ImageNet-21k (Deng et al., 2009). We compare our method to two baselines: (1) FFT (**F**ull parameters **F**ine-**T**uned) that fine-tunes all parameters; (2) LoRA that fine-tunes the base model with the popular LoRA method (Hu et al., 2021) with rank $r' = \{16, 32\}$.

The results are presented in Table 2. Our method performs much better than FFT: $7.40\%$, $5.81\%$ and $13.26\%$ higher accuracy on the three datasets, respectively. Note that FFT updates all parameters and has much higher computational costs than LoRA or our method. Our method has a similar

Table 2: Performance of different fine-tuning methods on the CIFAR-100, Food-101 and Tiny-Imagenet datasets. The base model is ViT-Base. The methods include FFT, LoRA, and our method NTK-Attention. The metric is accuracy (measured in %). The best score on each dataset is **boldfaced**.

| Method | Num Params | Dataset | | | Average |
| --- | --- | --- | --- | --- | --- |
| | | CIFAR-100 | Food-101 | Tiny-Imagenet | |
| FFT | 86.39M | $85.15_{\pm 0.13}$ | $84.76_{\pm 0.07}$ | $76.20_{\pm 0.23}$ | $82.04_{\pm 0.14}$ |
| LoRA $r' = 16$ | 7.08M | $92.17_{\pm 0.05}$ | $89.38_{\pm 0.33}$ | $88.22_{\pm 0.09}$ | $89.92_{\pm 0.16}$ |
| LoRA $r' = 32$ | 14.16M | $92.01_{\pm 0.20}$ | $89.86_{\pm 0.11}$ | $\mathbf{90.16}_{\pm 0.12}$ | $90.68_{\pm 0.14}$ |
| NTK-Attention (ours), $r = 64, s = 32$ | 7.09M | $\mathbf{92.55}_{\pm 0.03}$ | $\mathbf{90.57}_{\pm 0.01}$ | $89.46_{\pm 0.10}$ | $\mathbf{90.86}_{\pm 0.05}$ |

Table 3: Performance of different fine-tuning methods on OPT-{125M, 350M, 1.3B, 2.7B, 6.7B} pretrained models with WikiText-103, Penn TreeBank and LAMBADA datasets. The metric is perplexity (PPL), with its smaller value standing for better performance. The best score on each dataset and model is **boldfaced**.

| Model | Method | Num Params | Datasets | | | Average |
| --- | --- | --- | --- | --- | --- | --- |
| | | | WikiText-103 | Penn TreeBank | LAMBADA | |
| OPT-125M | LoRA, $r' = 8$ | | **30.50** | 35.97 | 46.02 | 37.50 |
| | P-Tuning V2, $m = 32$ | 0.29M | 2264.22 | 963.09 | 1762.19 | 1663.17 |
| | NTK-Attention, $r = 2d, s = 10$ | | 31.41 | **33.52** | **45.39** | **36.77** |
| OPT-350M | LoRA, $r' = 8$ | | **24.76** | 30.41 | 38.80 | 31.32 |
| | P-Tuning V2, $m = 32$ | 0.77M | 7383.48 | 1339.43 | 14020.36 | 7581.09 |
| | NTK-Attention, $r = 2d, s = 10$ | | 25.67 | **28.85** | **36.97** | **30.50** |
| OPT-1.3B | LoRA, $r' = 8$ | | **16.71** | 21.27 | 24.16 | 20.71 |
| | P-Tuning V2, $m = 32$ | 1.57M | 2230.76 | 540.17 | 3480.77 | 2083.9 |
| | NTK-Attention, $r = 2d, s = 10$ | | 17.04 | 20.09 | 24.04 | **20.39** |
| OPT-2.7B | LoRA, $r' = 8$ | | 15.06 | 19.61 | 22.13 | 18.93 |
| | P-Tuning V2, $m = 32$ | 2.62M | 772.48 | 277.99 | 3378.18 | 1476.22 |
| | NTK-Attention, $r = 2d, s = 10$ | | **14.83** | **18.52** | **21.85** | **18.40** |
| OPT-6.7B | LoRA, $r' = 8$ | | 12.81 | 17.36 | 19.38 | 16.52 |
| | P-Tuning V2, $m = 32$ | 4.19M | 2051.10 | 409.37 | 4709.46 | 2389.98 |
| | NTK-Attention, $r = 2d, s = 10$ | | **12.56** | **16.68** | 18.81 | **16.02** |

performance to LoRA with $r' = 32$, achieving slightly better average accuracy. These results on vision datasets also provide positive empirical support for our method.

**Ablation Study.** We validate the sensitivity of hyper-parameters $r$ and $s$ and give the results in Appendix B.3. The results firstly indicate that choosing $r = d$ and $s = 4$ is enough for high-performance fine-tuning on LLAMA-3.1-8B. Also, we follow Table 4 to suggest choosing a larger value of $r$ primarily instead of $s$ to achieve supernal accuracy.

**Empirical Evaluation of Computational Cost.** We also provide experimental results of the computational costs of NTK-Attention (Algorithm 2) and the standard Prefix Attention (Algorithm 1) in Appendix B.2. The results show that Prefix Attention's run time is quadratic and memory usage is linear in the prefix length, so its costs are typically much higher, while NTK-Attention maintains a small run time and memory usage.

# 6 CONCLUSION

In this study, we illuminated the principles of prefix learning for fine-tuning when the prefix length is large. We conducted an in-depth theoretical analysis, demonstrating that when the prefix length is sufficiently large, the attention network is over-parameterized, and the Neural Tangent Kernel technique can be leveraged to provide a convergence guarantee of prefix learning. Based on these insights, we proposed a novel efficient fine-tuning method called NTK-Attention, which approximates prefix attention using two trainable parameters to replace the large prefix matrix, thus significantly mitigating memory usage issues and reducing computational cost for long prefixes. We also provided empirical results to support our theoretical findings, demonstrating NTK-Attention's superior performance on downstream tasks over baselines across natural language, math, and vision datasets.

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

# Appendix

CONTENTS

**Roadmap.** In Appendix A, we present the details of our method and prefix attention, and give a complexity and memory analysis.

The experimental details for our empirical evaluation is shown in Appendix B. We give a naive implementation of NTK-Attention within Python code in Appendix C. We provide more discussions on our work in Appendix D, including the limitations and societal impacts of this paper.

We provide the preliminary we use in our analysis in Appendix E, including helpful probability tools. We provide the basic definitions in Appendix F, and give helpful Lemmas about gradient computation in Appendix G. Then we present our adaptation of NTK in our analysis in Appendix H, in Appendix I show how to decompose the training objective to simplify proofs, and finally post our main results and the proofs for analyzing the training in Appendix J.

In Appendix K, we compute the error bound on our NTK-Attention approximating ultra-long prefix in attention. In Appendix L, we state helpful tools about the Taylor series.

## A  ALGORITHM DETAILS AND COMPUTATIONAL COMPLEXITY ANALYSIS

Here, we give the detailed version of two algorithms of this paper, which are prefix attention and NTK-Attention. Moreover, we comment on each computation step with its corresponding complexity to demonstrate our memory and complexity reduction in detail.

From Algorithm 3 and Algorithm 4, we can see the comparison analysis of memory reduction (from $O(md)$ to $O(rd + r)$) and complexity reduction (from $O(mL + L^2)$ to $O(Ld + L^2)$ since $m \gg L$ and $m \gg d$) between two fine-tuning methods, indicating the efficiency of our NTK-Attention.

---

**Algorithm 3** Prefix Attention (Detailed version of Algorithm 1)

---

**Input:** Input matrix $X \in \mathbb{R}^{L \times d}$
**Parameters:** Frozen query, key and value weights $W_Q, W_K, W_V \in \mathbb{R}^{d \times d}$, trainable prefix matrix $P \in \mathbb{R}^{m \times d}$                              ▷ Additional memory usage $O(md)$
**Output:** Exact output Attn $\in \mathbb{R}^{L \times d}$
1: **procedure** PREFIXATTENTION($X$)
2:   Concatenate input matrix with prefix matrix $S \leftarrow \begin{bmatrix} P \\ X \end{bmatrix} \in \mathbb{R}^{(m+L) \times d}$
3:   Compute query, key, and value matrices for attention $Q \leftarrow XW_Q \in \mathbb{R}^{L \times d}, K_P \leftarrow SW_K \in \mathbb{R}^{(m+L) \times d}, V_P \leftarrow SW_V \in \mathbb{R}^{(m+L) \times d}$        ▷ Time complexity $O(Ld^2 + 2(m + L)d^2)$
4:   Compute exponential matrix $A \leftarrow \exp(QK_P^\top/\sqrt{d}) \in \mathbb{R}^{L \times (m+L)}$        ▷ Time complexity $O(L(m + L)d)$
5:   Compute summation of exponential matrix $D \leftarrow \mathrm{diag}(A\mathbf{1}_{m+L}) \in \mathbb{R}^{L \times L}$        ▷ Time complexity $O(L(m + L))$
6:   Compute prefix attention output Attn $\leftarrow D^{-1}AV_P \in \mathbb{R}^{L \times d}$ ▷ Here $D^{-1}A \in \mathbb{R}^{L \times (m+L)}$ is the attention matrix (a.k.a attention scores). This step implements $A$ multiply $V_P$ first, then get $D^{-1} \cdot (AV_P)$ with time complexity $O(L(m + L)d + L^2d)$
7:   **return** Attn
8: **end procedure**

---

**Algorithm 4** NTK-Attention (Detailed version of Algorithm 2, w low-rank)

---

**Input:** Input matrix $X \in \mathbb{R}^{L \times d}$
**Parameters:** Frozen query, key and value weights $W_Q, W_K, W_V \in \mathbb{R}^{d \times d}$, trainable weights $Z_A \in \mathbb{R}^{r \times s}, Z_B \in \mathbb{R}^{s \times d}$ and $k \in \mathbb{R}^r$        ▷ Additional memory usage $O(rs + sd + r)$
**Output:** Approximating output $T \in \mathbb{R}^{L \times d}$
1: **procedure** NTK-ATTENTION($X$)
2:   Compute query, key, and value matrices for attention $Q \leftarrow XW_Q \in \mathbb{R}^{L \times d}, K \leftarrow XW_K \in \mathbb{R}^{L \times d}, V \leftarrow XW_V \in \mathbb{R}^{L \times d}$        ▷ Time complexity $O(3Ld^2)$
3:   Compute approximating exponential matrix $\widehat{A} \leftarrow \exp(QK^\top/\sqrt{d}) \in \mathbb{R}^{L \times L}$        ▷ Time complexity $O(L^2d)$
4:   Compute approximating summation of exponential matrix $\widehat{D} \leftarrow \mathrm{diag}(\widehat{A}\mathbf{1}_L + \Phi(Q)k) \in \mathbb{R}^{L \times L}$        ▷ Time complexity $O(L^2 + Lr)$
5:   Compute approximation of prefix attention output $T \leftarrow \widehat{D}^{-1}(\widehat{A}V + \Phi(Q)Z_A \cdot Z_B) \in \mathbb{R}^{L \times d}$ ▷ This step implements $Z := Z_A \cdot Z_B$ first, compute $\widehat{A}V + \Phi(Q)Z$ secondly, then implements $\widehat{D}^{-1} \cdot (\widehat{A}V + \Phi(Q)Z_A \cdot Z_B)$, time complexity $O(2L^2d + Lr^2 + rsd)$
6:   **return** $T$
7: **end procedure**

---

## B  EXPERIMENTAL DETAILS

### B.1  SETUP DETAILS

Here, we give the details of the setup for the experiments in Section 5.

- Learning rate $\eta = 0.001$ (default).

- Learning rate scheduler: Cosine.

- Adam hyper-parameter $\beta_1 = 0.9$ (default).

- Adam hyper-parameter $\beta_2 = 0.999$ (default).

- Adam hyper-parameter $\epsilon = 1 \times 10^{-8}$ (default).

- Platform: PyTorch (Paszke et al., 2019) and Huggingface (Wolf et al., 2019).

- GPU device information: 8 V100 GPUs, 8 4090 GPUs and 4 H800 GPUs.

- Number of training epochs 30.

- Batch size for vision tasks: 256 (for best effort).

- Batch size for natural language task: 32 (for best effort).

- Max input length for natural language task: 128 for each feature, e.g. BoolQ has two dataset features: question and passage, for each data, we select the first 128 tokens in question and passage of the data respectively, and concatenate them to be the input.

- Quantization: fp16 and bf16.

## B.2 ADDITIONAL EMPIRICAL COMPLEXITY ANALYSIS

We state an additional empirical complexity analysis here to support our claim practically. We evaluate the complexity reduction on one layer to show how much efficiency our NTK-Attention will demonstrate per layer.

**Setup.** Firstly, we choose $d = 32$ and $r = d$, and randomly initialize attention weights $W_Q, W_K, W_V \in \mathbb{R}^{d \times d}$. For the trainable parameters in NTK-Attention and Prefix Attention, we initialize $P \in \mathbb{R}^{m \times d}$, $Z \in \mathbb{R}^{d \times d}$ and $k \in \mathbb{R}^d$ randomly, either. We then scale the prefix length, denotes $m$, within the range $\{2^0, 2^1, \cdots, 2^{16}\}$ for comparison. The input length $L$ is chosen from $\{32, 64, 128, 256\}$. For computation, we initialize a new input matrix $X \in \mathbb{R}^{L \times d}$ and compute NTK-Attention and Prefix Attention respectively. We repeat each computation with a different setup 10000 times and record the maximum, minimum, and mean values. The inference is run on an AMD CPU to compare FLOPS fairly between two algorithms (this also works on GPU devices).

Figure 3: Run time and the number of parameters of one-layer NTK-Attention and Prefix Attention (on random input data). $x$-axis: the number of parameters; $y$-axis: run time. Input length $L$ is chosen from $\{32, 64, 128, 256\}$, dimension $d = 32$ and prefix length $m$ is chosen from $\{2^0, 2^1, \cdots, 2^{16}\}$.

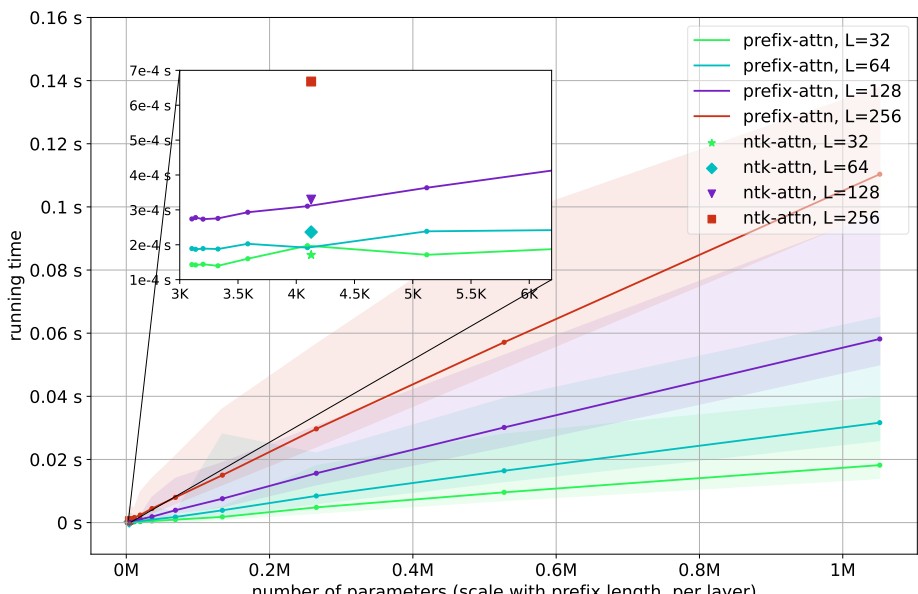

**Results.** We demonstrate our result in Figure 3. The $x$-axis is the number of parameters (representing memory usage), and the $y$-axis shows the run time in seconds. Note that the number of parameters is computed by the summation of every number in NTK-Attention or Prefix Attention. For example, $m = 1024, d = 32$, the number of parameters of Prefix Attention is $md + 3d^2 = 35840$; the number of parameters if NTK-Attention is $4d^2 + d = 4128$.

As expected, the number of parameters of Prefix Attention increases linearly with the prefix length $m$, and its running time increases quadratically with $m$. While our method, NTK-Attention, has computational costs unaffected by the prefix length. It maintains a small running time and low memory usage as shown in the figure. Roughly speaking, the cost of NTK-Attention is close to Prefix Attention with a very small prefix length $m = 32$.

### B.3 ADDITIONAL ABLATION STUDY

**Setup.** We provide an additional ablation study for the sensitivity of the hyper-parameters of NTK-Attention $r$ and $s$ here and the results are given in Table 4. In particular, this experiment is run on pretrained LLAMA-3.1-8B-Instruct model ($d = 128$ for each head in attention) (Touvron et al., 2023a;b) with dataset WikiText-103 (Merity et al., 2016). We utilize 4 H800 GPU devices to train the model with different settings within 2 epochs on the training dataset and evaluate them on the test dataset. The metric is cross-entropy loss and its smaller value stands for better performance.

**Results.** We show the NTK-Attention with the weakest setting $r = 128, s = 4$ is able to achieve competitive performance with $r = 256, r = 64$. This further ensures the parameter efficiency of NTK-Attention.

Moreover, Table 4 also demonstrates that choosing a big value for hyper-parameter $r$ primarily will lead to better evaluation loss since NTK-Attention with $(r, s) = (256, 32)$ requires 12.85M parameters but achieve superior performance compared to NTK-Attention with $(r, s) = (128, 64)$ (requires 16.91M parameters).

However, we discover that an increased value for $r$ might cause huge complexity - when setting $r = 512$, the computational complexity $4Ld$ will lead the GPU out-of-memory (OOM) since it's usually unaffordable even for H800 (80GiB memory). Thus, we also suggest using $r = d$ or $r = 2d$ to make LLMs to learn downstream tasks.

Table 4: The results of ablation study to the NTK-Attention hyper-parameters $r$ and $s$ with pretrained LLM LLAMA-3.1-8B-Instruct and dataset WikiText-103 on H800 GPUs (80GiB).

| Hyper-parameters | Num Parameters | Evaluation Loss | Training Loss |
|---|---|---|---|
| $(r, s)$=(128, 4) | 1.18M | 2.48 | 2.38 |
| $(r, s)$=(128, 8) | 2.23M | 2.57 | 2.50 |
| $(r, s)$=(128, 16) | 4.33M | 2.74 | 2.72 |
| $(r, s)$=(128, 32) | 8.52M | 2.47 | 2.38 |
| $(r, s)$=(128, 64) | 16.91M | 2.41 | 2.31 |
| $(r, s)$=(256, 4) | 1.84M | 2.47 | 2.39 |
| $(r, s)$=(256, 8) | 3.41M | 2.43 | 2.36 |
| $(r, s)$=(256, 16) | 6.55M | 2.51 | 2.53 |
| $(r, s)$=(256, 32) | 12.85M | 2.28 | 2.33 |
| $(r, s)$=(256, 64) | 25.43M | 2.21 | 2.15 |
| $(r, s)$=(512, 4) | 3.15M (OOM since $4Ld$ complexity) | - | - |

## C NAIVE NTK-ATTENTION IMPLEMENTATION WITH FLASH-ATTENTION

Below, we provide a naive Python code to implement our NTK-Attention that is written in only 10 lines, which supports the simplicity of implementation. Our code utilizes the function of Flash Attention function (Dao et al., 2022; Dao, 2023; Shah et al., 2024).

```
1  def ntk_attn_forward(self, query_states, key_states, value_states,
       attention_mask):
2      attn_outputs, lse = _flash_attention_forward(
3          query_states, key_states, value_states, attention_mask,
4          is_causal=self.is_causal, return_attn_probs=True
5      ) # Call flash-attn function to get attn_output and logsumexp
6
7      Z = torch.matmul(self.Z_A, self.Z_B) # Low-rank approximate Z
8      k = self.k
9      phi_query_states = self.phi(query_states)
10
11     se = lse.exp() # Compute sumexp
12     scale_factor = (se + torch.matmul(phi_query_states, k)) / se
13
14     attn_output = scale_factor * (attn_output * se + torch.matmul(
       phi_query_states, Z))
15
16     return attn_output
```

## D  FURTHER DISCUSSIONS

Prior works (Arora et al., 2019; Alemohammad et al., 2020; Hron et al., 2020) had already given exact algorithms for computing the extension of NTK to neural nets and conducted experiments showing enhanced performance from adding NTK into models, while in this paper, our contributions are not limited to this. Our theory about NTK of attention with the infinite-long prefix provides more insights. We clarify this further in the following.

**Can LLMs master any advanced reasoning skill through self-planning and prompting?** We will answer that it may be possible. Since an attention network can converge on any dataset with the infinite-long prefix, we can tell that for any advanced reasoning skill that is equivalent to training on a well-constructed dataset, there exists an ultra-long prefix matrix satisfying the training objective smaller than any positive value $\epsilon > 0$. It's noteworthy that this conclusion is not only suitable for LLMs with outstanding performance but also can be worked on those small language models with common performance.

**What is NTK-Attention used for? What is the meaning of proposing this method?** The attention with an infinite-long prefix is superior due to its over-parameterization phenomenon, whereas it is nearly impossible to implement practically, our NTK-Attention method gives us a chance to approximate the infinite-long prefix and makes it possible for us to study its empirical properties in experiments. Besides, any form of prefix learning can be formulated into the training of $Z \in \mathbb{R}^{d \times d}$ and $k \in \mathbb{R}^d$ in NTK-Attention, we can compress prompts into $Z$ and $k$ if $\phi(\cdot)$ by utilizing Lemma K.7, hence, the approaches in Prefix Learning would be much more efficient.

**Comparison between NTK-Attention and LoRA.** LoRA in (Hu et al., 2021; Zeng & Lee, 2023; Hu et al., 2024) is a popular efficient fine-tuning method for large base models. Usually, LoRA makes adaptation on Query and Value projections $W_Q, W_V \in \mathbb{R}^{d \times d}$; denote the adaptation as $W_{\Delta Q}, W_{\Delta V} \in \mathbb{R}^{d \times d}$. Given an input $X \in \mathbb{R}^{L \times d}$, LoRA computes $\widetilde{D}^{-1}\widetilde{A}X(W_V + W_{\Delta V})$, where $\widetilde{A} := \exp(X(W_Q + W_{\Delta Q})W_K^\top X^\top)$, $\widetilde{D} := \operatorname{diag}(\widetilde{A}\mathbf{1}_L)$, and $W_K \in \mathbb{R}^{d \times d}$ is the Key projection weights. So LoRA updates query and value weights during training, while our NTK-Attention compresses the additional prefix $P$ into $Z$ and $k$ (Algorithm 2), which is a completely different mechanism. Our method also achieves comparable performance to LoRA in our experiments in Section 5. Also, note that the two methods are orthogonal to each other and can be used together.

**Connection to the newest SOTA LLM on math inference tasks, GPT-o1** [1]. On September 12-th, 2024, OpenAI released the newest SOTA LLM on math inference tasks, GPT-o1, which is trained by Reinforcement Learning (RL) methods to enhance the Chain-of-Thought (CoT) ability. Li et al. (2024c) explained the necessity of CoT for LLM on complicated inference tasks, meanwhile, they also emphasized how the embedding size and the CoT length affect the capability to solve high-order problems. Connecting to our work, we believe that these empirical and theoretical results support the

---

[1] https://openai.com/o1/

conclusion of our work since we consider CoT as a specific application of Prefix Learning. Moreover, we think our *scaling law in prefix learning* is more universal for explaining the LLMs' context-based advanced skills. However, even when we present our theory, we still have a limited understanding of prefix learning, for example, what is the relationship between prefix length and complexity of problems that aim to solve; if we want to solve an NP problem by LLM, how long is the prefix needed for inference? We don't know the answers. Thus, explaining prefix learning, or particularly, CoT, is still a fascinating and challenging problem for future work.

**Limitations.** The work has limited experimental analysis and results. While empirical evaluations have been provided for some datasets and LLM models, the proposed method is widely applicable to different data and models, so comprehensive evaluations on more datasets and more practical methods can provide stronger empirical support.

Besides, the computational efficiency of NTK-Attention is insufficiently better than prefix attention when $m < d$, since the design of NTK-Attention is toward the ultra-big value of $m$, such we only compare to the prefix attention with prefix length $m \gg d$ to meet the over-parameterization setting in our analysis.

**Societal impact.** This paper presents work whose goal is to advance the understanding of context-based fine-tuning methods (prefix learning) theoretically. There are many positive potential societal consequences of our work, such as inspiring new algorithm design. Since our work is theoretical in nature, we do not foresee any potential negative societal impacts which worth pointing out.

## E PRELIMINARY OF ANALYSIS

We provide our notations for this paper as follows:

**Notations** In this paper, we use integer $d$ to denote the dimension of networks. We use integer $m$ to denote the prefix length in prefix learning, we think $m$ is an ultra-big number. We use $L$ to denote the input length in language models. $\nabla_x f(x)$ and $\frac{\mathrm{d}f(x)}{\mathrm{d}x}$ are both means to take the derivative of $f(x)$ with $x$. Let a vector $z \in \mathbb{R}^n$. We denote the $\ell_2$ norm as $\|z\|_2 := (\sum_{i=1}^n z_i^2)^{1/2}$, the $\ell_1$ norm as $\|z\|_1 := \sum_{i=1}^n |z_i|$, $\|z\|_0$ as the number of non-zero entries in $z$, $\|z\|_\infty$ as $\max_{i \in [n]} |z_i|$. We use $z^\top$ to denote the transpose of a $z$. We use $\langle \cdot, \cdot \rangle$ to denote the inner product. Let $A \in \mathbb{R}^{n \times d}$, we use $\mathrm{vec}(A)$ to denote a length $nd$ vector. We denote the Frobenius norm as $\|A\|_F := (\sum_{i \in [n], j \in [d]} A_{i,j}^2)^{1/2}$. For any positive integer $n$, we use $[n]$ to denote set $\{1, 2, \cdots, n\}$. We use $\mathbb{E}[]$ to denote the expectation. We use $\Pr[]$ to denote the probability. We use $\epsilon$ to denote the error. We define $\lambda_{\min}(\cdot)$ as a function that outputs the minimum eigenvalues of the input matrix, e.g. matrix $A \in \mathbb{R}^{n \times n}$ has eigenvalues $\{\lambda_1, \lambda_2, \cdots, \lambda_n\}$, $\lambda_{\min}(A) = \min\{\lambda_1, \lambda_2, \cdots, \lambda_n\}$.

### E.1 FACTS

**Fact E.1.** *For any $x \in (-0.01, 0.01)$, we have*

$$\exp(x) = 1 + x + \Theta(1)x^2.$$

**Fact E.2.** *For any $x \in (0, 0.1)$, we have*

$$\sum_{i=1}^n x^i \leq \frac{1}{1-x}.$$

### E.2 PROBABILITY

Here, we state a probability toolkit in the following, including several helpful lemmas we'd like to use. Firstly, we provide the lemma about Chernoff bound in (Chernoff, 1952) below.

**Lemma E.3** (Chernoff bound, (Chernoff, 1952))**.** *Let $X = \sum_{i=1}^n X_i$, where $X_i = 1$ with probability $p_i$ and $X_i = 0$ with probability $1 - p_i$, and all $X_i$ are independent. Let $\mu = \mathbb{E}[X] = \sum_{i=1}^n p_i$. Then*

- $\Pr[X \geq (1+\delta)\mu] \leq \exp(-\delta^2\mu/3), \forall \delta > 0$;

- $\Pr[X \leq (1 - \delta)\mu] \leq \exp(-\delta^2\mu/1), \forall 0 < \delta < 1.$

Next, we offer the lemma about Hoeffding bound as in (Hoeffding, 1994).

**Lemma E.4** (Hoeffding bound, (Hoeffding, 1994)). *Let $X_1, \cdots, X_n$ denote $n$ independent bounded variables in $[a_i, b_i]$ for $a_i, b_i \in \mathbb{R}$. Let $X := \sum_{i=1}^n X_i$, then we have*

$$\Pr[|X - \mathbb{E}[X]| \geq t] \leq 2\exp(-\frac{2t^2}{\sum_{i=1}^n (b_i - a_i)^2})$$

We show the lemma of Bernstein inequality as (Bernstein, 1924).

**Lemma E.5** (Bernstein inequality, (Bernstein, 1924)). *Let $X_1, \cdots, X_n$ denote $n$ independent zero-mean random variables. Suppose $|X_i| \leq M$ almost surely for all $i$. Then, for all positive $t$,*

$$\Pr[\sum_{i=1}^n X_i \geq t] \leq \exp(-\frac{t^2/2}{\sum_{j=1}^n \mathbb{E}[X_j^2] + Mt/3})$$

Then, we give the Khintchine's inequality in (Khintchine, 1923; Haagerup, 1981) as follows:

**Lemma E.6** (Khintchine's inequality, (Khintchine, 1923; Haagerup, 1981)). *Let $\sigma_1, \cdots, \sigma_n$ be i.i.d sign random variables, and let $z_1 \cdots, z_n$ be real numbers. Then there are constants $C > 0$ so that for all $t > 0$*

$$\Pr[|\sum_{i=1}^n z_i \sigma_i| \geq t\|z\|_2] \leq \exp(-Ct^2).$$

We give Hason-wright inequality from (Hanson & Wright, 1971; Rudelson & Vershynin, 2013) below.

**Lemma E.7** (Hason-wright inequality, (Hanson & Wright, 1971; Rudelson & Vershynin, 2013)). *Let $x \in \mathbb{R}^n$ denote a random vector with independent entries $x_i$ with $\mathbb{E}[x_i] = 0$ and $|x_i| \leq K$ Let $A$ be an $n \times n$ matrix. Then, for every $t \geq 0$*

$$\Pr[|x^\top A x - \mathbb{E}[x^\top A x]| > t] \leq 2\exp(-c\min\{t^2/(K^4\|A\|_F^2), t/(K^2\|A\|)\}).$$

We state Lemma 1 on page 1325 of Laurent and Massart (Laurent & Massart, 2000).

**Lemma E.8** (Lemma 1 on page 1325 of Laurent and Massart, (Laurent & Massart, 2000)). *Let $X \sim \mathcal{X}_k^2$ be a chi-squared distributed random variable with $k$ degrees of freedom. Each one has zero mean and $\sigma^2$ variance. Then*

$$\Pr[X - k\sigma^2 \geq (2\sqrt{kt} + 2t)\sigma^2] \leq \exp(-t)$$
$$\Pr[X - k\sigma^2 \geq 2\sqrt{kt}\sigma^2] \leq \exp(-t).$$

Here, we provide a tail bound for sub-exponential distribution (Foss et al., 2011).

**Lemma E.9** (Tail bound for sub-exponential distribution, (Foss et al., 2011)). *We say $X \in \mathrm{SE}(\sigma^2, \alpha)$ with parameters $\sigma > 0$, $\alpha > 0$, if*

$$\mathbb{E}[e^{\lambda X}] \leq \exp(\lambda^2\sigma^2/2), \forall|\lambda| < 1/\alpha.$$

*Let $X \in \mathrm{SE}(\sigma^2, \alpha)$ and $\mathbb{E}[X] = \mu$, then:*

$$\Pr[|X - \mu| \geq t] \leq \exp(-0.5\min\{t^2/\sigma^2, t/\alpha\}).$$

In the following, we show the helpful lemma of matrix Chernoff bound as in (Tropp, 2011; Lu et al., 2013).

**Lemma E.10** (Matrix Chernoff bound, (Tropp, 2011; Lu et al., 2013)). *Let $\mathcal{X}$ be a finite set of positive-semidefinite matrices with dimension $d \times d$, and suppose that*

$$\max_{X \in \mathcal{X}} \lambda_{\max}(X) \leq B.$$

*Sample $\{X_1, \cdots, X_n\}$ uniformly at random from $\mathcal{X}$ without replacement. We define $\mu_{\min}$ and $\mu_{\max}$ as follows:*

$$\mu_{\min} := n \cdot \lambda_{\min}(\underset{X \in \mathcal{X}}{\mathbb{E}}(X))$$

$$\mu_{\max} := n \cdot \lambda_{\max}(\underset{X \in \mathcal{X}}{\mathbb{E}}(X)).$$

*Then*

$$\Pr[\lambda_{\min}(\sum_{i=1}^{n} X_i) \le (1-\delta)\mu_{\min}] \le d \cdot \exp(-\delta^2 \mu_{\min}/B) \text{ for } \delta \in (0,1],$$

$$\Pr[\lambda_{\max}(\sum_{i=1}^{n} X_i) \ge (1+\delta)\mu_{\max}] \le d \cdot \exp(-\delta^2 \mu_{\max}/(4B)) \text{ for } \delta \ge 0.$$

## F  DEFINITIONS OF NTK ANALYSIS

This section provides the fundamental definitions of our NTK analysis in this paper.

To begin with, we re-denote our weight of prefix in attention as $W \in \mathbb{R}^{d \times m}$ and $a \in \{-1, +1\}^m$ as follows[2]:

**Definition F.1.** *We choose $a \in \{-1, +1\}^m$ to be weights that each entry $a_r$ is randomly sampled from $-1$ with probability $1/2$ and $+1$ with probability $1/2$.*

*Let $W \in \mathbb{R}^{d \times m}$ denote random Gaussian weights, i.e., each entry independently draws from $\mathcal{N}(0, \sigma^2)$. For each $r \in [m]$, we use $w_r \in \mathbb{R}^d$ to denote the $r$-th column of $W$.*

Since we have established the equivalence between the ultra-long prefix matrix in attention and our theory in Section 3.1, it's reasonable we utilize the following definition of F to decompose the model function and facilitate our analysis.

**Definition F.2.** *We define function $\mathsf{F} : \mathbb{R}^{d \times m} \times \mathbb{R}^d \times \mathbb{R}^m \to \mathbb{R}^d$*

$$\mathsf{F}(W, x, a) = m \frac{\sum_{r \in [m]} a_r \exp(w_r^\top x) w_r}{\sum_{r \in [m]} \exp(w_r^\top x)}$$

*Here we use $w_r \in \mathbb{R}^d$ to denote the $r$-th column of $W \in \mathbb{R}^{d \times m}$.*

To further break down the complicated F for more convenience analysis. We give an operator function $\alpha$ as follows:

**Definition F.3.** *We define $\alpha(x)$ as follows*

$$\alpha(x) := \langle \exp(\underbrace{W^\top}_{m \times d} \underbrace{x}_{d \times 1}), \mathbf{1}_m \rangle$$

Thus, we can rewrite F in the following claim.

**Claim F.4.** *We can rewrite $\mathsf{F}(W, x, a) \in \mathbb{R}^d$ as follows*

$$\mathsf{F}(W, x, a) = m \underbrace{\alpha(x)^{-1}}_{\text{scalar}} \underbrace{W}_{d \times m} (\underbrace{a}_{m \times 1} \circ \underbrace{\exp(W^\top x)}_{m \times 1})$$

*Proof.* We can show

$$\mathsf{F}(W, x, a) = m \frac{\sum_{r \in [m]} a_r \exp(w_r^\top x) w_r}{\sum_{r \in [m]} \exp(w_r^\top x)}$$

$$= m \alpha(x)^{-1} \sum_{r \in [m]} a_r \exp(w_r^\top x) w_r$$

---

[2]Note that the proof of the case with $a$ and without $a$ are similar. We mainly focus on the proofs under the setting that includes $a$.

$$= m\alpha(x)^{-1}W(a \circ \exp(W^\top x))$$

where the first step follows from Definition F.2, the second step follows from Definition F.3 and simple algebras, the third step follows from $w_r \in \mathbb{R}^d$ is denoting the $r$-th column of $W \in \mathbb{R}^{d \times m}$ and simple algebras. $\square$

In the following Definition F.6 and Definition F.5, we further derive and define two operator functions to convenient our analysis.

**Definition F.5.** *We define $\beta$ as follows*

$$\beta_k := W_{k,*} \circ a, \forall k \in [d]$$

*Let $\beta \in \mathbb{R}^{d \times m}$ be defined as* $\underbrace{\beta}_{d \times m} = \underbrace{W}_{d \times m}\underbrace{\mathrm{diag}(a)}_{m \times m}$

Here, we define softmax.

**Definition F.6.** *We define $\mathsf{S} \in \mathbb{R}^m$ as follows*

$$\mathsf{S} := \underbrace{\alpha(x)^{-1}}_{\text{scalar}} \cdot \underbrace{\exp(W^\top x)}_{m \times 1}.$$

Here, we use $\beta$ and $\mathsf{S}$ to re-denote the model function $\mathsf{F}$.

**Definition F.7.** *For each $k \in [d]$, let $W_{k,*}^\top$ denote the $k$-th row of $W$, we define*

$$\mathsf{F}_k(W, x, a) := m \underbrace{\alpha(x)^{-1}}_{\text{scalar}} \langle \underbrace{W_{k,*}}_{m \times 1} \circ \underbrace{a}_{m \times 1}, \underbrace{\exp(W^\top x)}_{m \times 1} \rangle$$

*Then, we can rewrite it as*

$$\mathsf{F}_k(W, x, a) := m \langle \beta_k, \mathsf{S} \rangle.$$

## F.1 LOSS FUNCTION

Here, we state the training objective that we aim to solve in the analysis.

**Definition F.8.** *Given a dataset $\mathcal{D} = \{(x_i, y_i)\}_{i=1}^n \subset \mathbb{R}^d \times \mathbb{R}^d$. Let function $\mathsf{F} : \mathbb{R}^{d \times m} \times \mathbb{R}^d \times \mathbb{R}^m \to \mathbb{R}^d$ be defined as Definition F.2, we define the training objective $\mathcal{L} : \mathbb{R}^{m \times d} \to \mathbb{R}$ as follows:*

$$\mathcal{L}(W) := 0.5 \sum_{i=1}^n \|\mathsf{F}(W, x_i, a) - y_i\|_2^2$$

## G GRADIENT COMPUTATION

In this section, we first compute the gradients that we need for the analysis of NTK. Then we define the training dynamic of our model in the process of gradient descent.

### G.1 COMPUTING GRADIENT

We give our computation of the gradients as the following lemma.

**Lemma G.1.** *If the following conditions hold*

- *Let $W \in \mathbb{R}^{d \times m}$ and $a \in \mathbb{R}^m$ be defined as Definition F.1.*

- *Let $\alpha(x) \in \mathbb{R}$ be defined as Definition F.3*

- *Let $\mathsf{S} \in \mathbb{R}^m$ be defined as Definition F.6*

- *Let $\mathsf{F} \in \mathbb{R}^d$ be defined as Definition F.7*

*Then, we can show that for each $r \in [m]$*

- **Part 1.** *For $k_1 \in [d]$, we have*

$$\frac{\mathrm{d}W^\top x}{\mathrm{d}w_{r,k_1}} = x_{k_1} e_r$$

- **Part 2.** *For $k_1 \in [d]$, we have*

$$\frac{\mathrm{d}\exp(W^\top x)}{\mathrm{d}w_{r,k_1}} = (x_{k_1} e_r) \circ \exp(W^\top x)$$

- **Part 3.** *For $k_1 \in [d]$, we have*

$$\frac{\mathrm{d}\alpha(x)}{\mathrm{d}w_{r,k_1}} = \langle x_{k_1} e_r, \exp(W^\top x)\rangle$$

- **Part 4.** *For $k_1 \in [d]$, we have*

$$\frac{\mathrm{d}\alpha(x)^{-1}}{\mathrm{d}w_{r,k_1}} = -\alpha(x)^{-1}\langle x_{k_1} e_r, \mathsf{S}\rangle$$

- **Part 5.** *For $k_1 \in [d]$, we have*

$$\frac{\mathrm{d}\mathsf{S}}{\mathrm{d}w_{r,k_1}} = -\langle x_{k_1} e_r, \mathsf{S}\rangle \cdot \mathsf{S} + (x_{k_1} e_r) \circ \mathsf{S}$$

- **Part 6.** *For $k_1, k \in [d]$ and $k_1 \neq k$, we have*

$$\frac{\mathrm{d}\mathsf{F}(W,x,a)_k}{\mathrm{d}w_{r,k_1}} = +0 - mx_{k_1} \cdot \mathsf{S}_r \cdot \langle \beta_k, \mathsf{S}\rangle + mx_{k_1}\mathsf{S}_r \beta_{k,r}$$

- **Part 7.** *For $k_1, k \in [d]$ and $k_1 = k$, we have*

$$\frac{\mathrm{d}\mathsf{F}(W,x,a)_k}{\mathrm{d}w_{r,k}} = +m\langle a \circ e_r, \mathsf{S}\rangle - mx_k \cdot \mathsf{S}_r \cdot \langle \beta_k, \mathsf{S}\rangle + mx_k \mathsf{S}_r \beta_{k,r}$$

- **Part 8.** *For $k \in [d]$, we have*

$$\frac{\mathrm{d}\mathsf{F}(W,x,a)_k}{\mathrm{d}w_r} = ma_r \mathsf{S}_r \cdot e_k - m\langle \beta_k, \mathsf{S}\rangle \mathsf{S}_r \cdot x + m\beta_{k,r}\mathsf{S}_r \cdot x$$

*Proof.* **Proof of Part 1.**

$$\frac{\mathrm{d}W^\top x}{\mathrm{d}w_{r,k_1}} = x_{k_1} e_r$$

where this step follows from simple differential rules.

**Proof of Part 2.**

$$\frac{\mathrm{d}\exp(W^\top x)}{\mathrm{d}w_{r,k_1}} = \exp(W^\top x) \circ \frac{\mathrm{d}W^\top x}{\mathrm{d}w_{r,k_1}}$$
$$= (x_{k_1} e_r) \circ \exp(W^\top x)$$

where the first step follows from chain rules, the second step follows from Part 1 of this Lemma.

**Proof of Part 3.**

$$\frac{\mathrm{d}\alpha(x)}{\mathrm{d}w_{r,k_1}} = \langle \frac{\mathrm{d}\exp(W^\top x)}{\mathrm{d}w_{r,k_1}}, \mathbf{1}_m \rangle$$

$$= \langle x_{k_1} e_r, \exp(W^\top x) \rangle$$

where the first step follows from Definition F.3 and simple algebras, the second step follows from Part 2 of this Lemma.

**Proof of Part 4.**

$$\frac{\mathrm{d}\alpha(x)^{-1}}{\mathrm{d}w_{r,k_1}} = -\alpha(x)^{-2} \frac{\mathrm{d}\alpha(x)}{\mathrm{d}w_{r,k_1}}$$
$$= -\alpha(x)^{-1} \langle x_{k_1} e_r, \mathsf{S} \rangle$$

where this step follows from chain rules, the second step follows from Part 3 of this Lemma.

**Proof of Part 5.**

$$\frac{\mathrm{d}\mathsf{S}}{\mathrm{d}w_{r,k_1}} = \frac{\mathrm{d}\alpha(x)^{-1}}{\mathrm{d}w_{r,k_1}} \cdot \exp(W^\top x) + \alpha(x)^{-1} \cdot \frac{\mathrm{d}\exp(W^\top x)}{\mathrm{d}w_{r,k_1}}$$
$$= -\alpha(x)^{-1} \langle x_{k_1} e_r, \mathsf{S} \rangle \cdot \exp(W^\top x) + \alpha(x)^{-1} \cdot (x_{k_1} e_r) \circ \exp(W^\top x)$$
$$= -\langle x_{k_1} e_r, \mathsf{S} \rangle \cdot \mathsf{S} + (x_{k_1} e_r) \circ \mathsf{S}$$

where the first step follows from Definition F.6 and differential rules, the second step follows from Part 2 and Part 4 of this Lemma, the last step follows from simple algebras.

**Proof of Part 6.** For $k_1 \neq k$

$$\frac{\mathrm{d}\mathsf{F}(W,x,a)_k}{\mathrm{d}w_{r,k_1}} = +m\langle \frac{\mathrm{d}\beta_k}{\mathrm{d}w_{r,k_1}}, \mathsf{S} \rangle + m\langle \beta_k, \frac{\mathrm{d}\mathsf{S}}{\mathrm{d}w_{r,k_1}} \rangle$$
$$= -m\langle x_{k_1} e_r, \mathsf{S} \rangle \cdot \langle \beta_k, \mathsf{S} \rangle + m\langle \beta_k, (x_{k_1} e_r) \circ \mathsf{S} \rangle$$
$$= +0 - mx_{k_1} \cdot \mathsf{S}_r \cdot \langle \beta_k, \mathsf{S} \rangle + mx_{k_1} \mathsf{S}_r \beta_{k,r}$$

where the first step follows from Definition F.7 and simple algebras, the second step follows from Definition F.5, simple algebras and Part 5 of this Lemma, the last step follows from simple algebras.

**Proof of Part 7.** For $k_1 = k$

$$\frac{\mathrm{d}\mathsf{F}(W,x,a)_k}{\mathrm{d}w_{r,k}} = +m\langle \frac{\mathrm{d}\beta_k}{\mathrm{d}w_{r,k}}, \mathsf{S} \rangle + m\langle \beta_k, \frac{\mathrm{d}\mathsf{S}}{\mathrm{d}w_{r,k}} \rangle$$
$$= +m\langle a \circ e_r, \mathsf{S} \rangle - m\langle x_k e_r, \mathsf{S} \rangle \cdot \langle \beta_k, \mathsf{S} \rangle + m\langle \beta_k, (x_k e_r) \circ \mathsf{S} \rangle$$
$$= +m\langle a \circ e_r, \mathsf{S} \rangle - mx_k \cdot \mathsf{S}_r \cdot \langle \beta_k, \mathsf{S} \rangle + mx_k \mathsf{S}_r \beta_{k,r}$$

where the first step follows from Definition F.7 and simple algebras, the second step follows from Definition F.5, simple algebras and Part 5 of this Lemma, the last step follows from simple algebras.

**Proof of Part 8.**

This part of proof follows from the combination of Part 6 and Part 7 of this Lemma. □

### G.2 GRADIENT DESCENT

After we computed the gradient of the model function above, we are now able to define the training dynamic of $\mathsf{F}$ by updating weight using gradient descent.

We use $e_r$ to denote a vector where the $r$-th coordinate is 1 and everywhere else is 0. $\forall r \in [m], \forall k \in [d]$, we have $\frac{\mathrm{d}\mathsf{F}(W,x,a)_k}{\mathrm{d}w_r} \in \mathbb{R}^d$ can be written as

$$\underbrace{\frac{\mathrm{d}\mathsf{F}_k(W,x,a)}{\mathrm{d}w_r}}_{d \times 1} = ma_r \mathsf{S}_r \cdot e_k - m\langle \beta_k, \mathsf{S} \rangle \mathsf{S}_r \cdot x + m\beta_{k,r} \mathsf{S}_r \cdot x. \quad (9)$$

Hence, by defining several following dynamical operator functions, we can further convenient our proofs.

We first define $\mathsf{u}_i(\tau) \in \mathbb{R}^m$ for simplification as follows:

**Definition G.2.** *For each $i \in [n]$, we define $\mathsf{u}_i(\tau) \in \mathbb{R}^m$ as*

$$\underbrace{\mathsf{u}_i(\tau)}_{m \times 1} := \exp(\underbrace{W(\tau)^\top}_{m \times d} \underbrace{x_i}_{d \times 1})$$

Secondly, we re-denote $\alpha_i(\tau) \in \mathbb{R}$ below, which holds due to the definition of $\alpha(x)$ and the updating of $W \in \mathbb{R}^{d \times m}$.

**Definition G.3.** *For each $i \in [n]$, we define $\alpha_i(\tau) \in \mathbb{R}$ as*

$$\underbrace{\alpha_i(\tau)}_{\text{scalar}} := \langle \underbrace{\mathsf{u}_i(\tau)}_{m \times 1}, \underbrace{\mathbf{1}_m}_{m \times 1} \rangle.$$

We define $\beta_k(\tau) \in \mathbb{R}^m$ for convenience.

**Definition G.4.** *For each $k \in [d]$, we define $\beta_k(\tau) \in \mathbb{R}^m$ as*

$$\underbrace{\beta_k(\tau)}_{m \times 1} = \underbrace{(W_{k,*}(\tau))}_{m \times 1} \circ \underbrace{a}_{m \times 1}$$

**Remark G.5.** *The purpose of defining notation $\beta$ is to make our proofs more aligned with softmax NTK proofs in previous work ([Li et al., 2024a](#)).*

We define $\theta_{k,i}(\tau) \in \mathbb{R}^m$ for convenience as follows :

**Definition G.6.** *For each $i \in [n]$, for each $k \in [d]$, we define $\theta_{k,i}(\tau) \in \mathbb{R}^m$ as follows*

$$\underbrace{\theta_{k,i}(\tau)}_{m \times 1} := \underbrace{\beta_k(\tau)}_{m \times 1} \cdot \underbrace{\alpha_i(\tau)^{-1}}_{\text{scalar}}$$

We denote $\mathsf{S}_r(\tau)$.

**Definition G.7.** *For each $i \in [n]$. Let $\mathsf{S}_i(\tau) \in \mathbb{R}^m$ be defined as*

$$\underbrace{\mathsf{S}_i(\tau)}_{m \times 1} := \underbrace{\alpha_i(\tau)^{-1}}_{\text{scalar}} \cdot \underbrace{\mathsf{u}_i(\tau)}_{m \times 1}$$

*for integer $\tau \geq 0$. For $r \in [m]$, we denote $\mathsf{S}_{i,r}(\tau) \in \mathbb{R}$ as the $r$-th entry of vector $\mathsf{S}_i(\tau)$.*

Now, we can define $\mathsf{F}$ at different timestamps.

**Definition G.8** ($\mathsf{F}(\tau)$, dynamic prediction). *For each $k \in [d]$, for each $i \in [n]$, we define $\mathsf{F}_i(\tau) \in \mathbb{R}^d$, for any timestamp $\tau$, as*

$$\mathsf{F}_{k,i}(\tau) := m \langle \mathsf{u}(\tau), \mathbf{1}_m \rangle^{-1} \langle W(\tau)_{k,*} \circ a, \mathsf{u}(\tau) \rangle.$$

*Here $x_i \in \mathbb{R}^d$. It can be rewritten as*

$$\mathsf{F}_{k,i}(\tau) = m \cdot \langle \underbrace{\beta_k(\tau)}_{m \times 1}, \underbrace{\mathsf{S}_i(\tau)}_{m \times 1} \rangle.$$

*and also*

$$\mathsf{F}_{k,i}(\tau) = m \cdot \langle \underbrace{\theta_{k,i}(\tau)}_{m \times 1}, \underbrace{u_i(\tau)}_{m \times 1} \rangle$$

We consider $d$-dimensional MSE loss.

**Definition G.9** (Loss function over time). *We define the objective function $\mathcal{L}$ as below:*

$$\mathcal{L}(W(\tau)) := \frac{1}{2} \sum_{i \in [n]} \sum_{k \in [d]} (\mathsf{F}_{k,i}(\tau) - y_{k,i})^2.$$

Thus, we define the gradient of $w$.

**Definition G.10** ($\Delta w_r(\tau)$). *For any $r \in [m]$, we define $\Delta w_r(\tau) \in \mathbb{R}^d$ as below:*

$$\Delta w_r(\tau)$$
$$:= m \sum_{i=1}^{n} \sum_{k=1}^{d} (\mathsf{F}_{k,i}(\tau) - y_{k,i}) \cdot \Big( a_r \mathsf{S}_{i,r}(\tau) \cdot e_k - \langle \beta_k(\tau), \mathsf{S}_i(\tau) \rangle \mathsf{S}_{i,r}(\tau) \cdot x + \beta_{k,r} \mathsf{S}_{i,r}(\tau) \cdot x \Big)$$

Here, we utilize $v$ to simplify $\Delta w_r(\tau)$, we have the following:

**Definition G.11.** *For each $k \in [d]$, for each $r \in [m]$, we define $v_{k,r}(\tau) \in \mathbb{R}^m$ as follows*

$$v_{k,r}(\tau) := \beta_{k,r}(\tau) \cdot \mathbf{1}_m - \beta_k(\tau).$$

Note that we can simplify the gradient calculation by the fact $1 = \langle \mathbf{1}_m, \mathsf{S}_i(\tau) \rangle$ for $i \in [n]$. Thus, we have the following claim.

**Claim G.12.** *We can rewrite $\Delta w_r(\tau)$ as follows*

$$\Delta w_r(\tau) = m \sum_{i=1}^{n} \sum_{k=1}^{d} (\mathsf{F}_{k,i}(\tau) - y_{k,i}) \cdot \Big( \langle v_{k,r}(\tau), \mathsf{S}_i(\tau) \rangle \cdot \mathsf{S}_{i,r}(\tau) \cdot x_i + a_r \mathsf{S}_{i,r}(\tau) e_k \Big)$$

*Proof.* We have

$$\Delta w_r(\tau)$$
$$= m \sum_{i=1}^{n} \sum_{k=1}^{d} (\mathsf{F}_{k,i}(\tau) - y_{k,i}) \cdot \Big( a_r \mathsf{S}_{i,r}(\tau) \cdot e_k - \langle \beta_k(\tau), \mathsf{S}_i(\tau) \rangle \mathsf{S}_{i,r}(\tau) \cdot x + \beta_{k,r} \mathsf{S}_{i,r}(\tau) \cdot x \Big)$$
$$= m \sum_{i=1}^{n} \sum_{k=1}^{d} (\mathsf{F}_{k,i}(\tau) - y_{k,i})$$
$$\cdot \Big( a_r \mathsf{S}_{i,r}(\tau) \cdot e_k - \langle \beta_k(\tau), \mathsf{S}_i(\tau) \rangle \mathsf{S}_{i,r}(\tau) \cdot x + \beta_{k,r} \langle \mathbf{1}_m, \mathsf{S}_i(\tau) \rangle \mathsf{S}_{i,r}(\tau) \cdot x \Big)$$
$$= m \sum_{i=1}^{n} \sum_{k=1}^{d} (\mathsf{F}_{k,i}(\tau) - y_{k,i})$$
$$\cdot \Big( a_r \mathsf{S}_{i,r}(\tau) \cdot e_k - \langle \beta_k(\tau), \mathsf{S}_i(\tau) \rangle \mathsf{S}_{i,r}(\tau) \cdot x + \langle \beta_{k,r} \cdot \mathbf{1}_m, \mathsf{S}_i(\tau) \rangle \mathsf{S}_{i,r}(\tau) \cdot x \Big)$$
$$= m \sum_{i=1}^{n} \sum_{k=1}^{d} (\mathsf{F}_{k,i}(\tau) - y_{k,i}) \cdot \Big( a_r \mathsf{S}_{i,r}(\tau) \cdot e_k + \langle \beta_{k,r} \cdot \mathbf{1}_m - \beta_k(\tau), \mathsf{S}_i(\tau) \rangle \mathsf{S}_{i,r}(\tau) \cdot x \Big)$$
$$= m \sum_{i=1}^{n} \sum_{k=1}^{d} (\mathsf{F}_{k,i}(\tau) - y_{k,i}) \cdot \Big( a_r \mathsf{S}_{i,r}(\tau) \cdot e_k + \langle v_{k,r}(\tau), \mathsf{S}_i(\tau) \rangle \mathsf{S}_{i,r}(\tau) \cdot x \Big)$$

where the first step follows from Definition G.10, the second step follows from the fact $1 = \langle \mathbf{1}_m, \mathsf{S}_i(\tau) \rangle$ for $i \in [n]$, the third and fourth steps follow from simple algebras, the last step follows from Definition G.11.

$\square$

We use the gradient descent (GD) algorithm with the learning rate $\eta$ to train the network. As we only train the hidden layer $W$ and fix $a$, we have the following gradient update rule.

**Definition G.13** (Gradient descent). *The gradient descent algorithm for optimizing the weight matrix $W$ is defined as:*

$$W(\tau + 1) = W(\tau) - \eta \Delta W(\tau).$$

*where $\Delta W(\tau) \in \mathbb{R}^{d \times m}$ and $\Delta w_r(\tau) \in \mathbb{R}^d$ is the $r$-th column of $\Delta W(\tau)$ defined in Definition G.10.*

# H  NEURAL TANGENT KERNEL

Now in this section, we give the exact computation of NTK in our analysis below.

**Definition H.1** (Kernel function, Definition 3.6 in (Li et al., 2024a) )**.** *For simplicity, we denote* $\mathsf{S}(W^\top x_i)$ *as* $\mathsf{S}_i \in \mathbb{R}^m_{\geq 0}$ *and* $v_{k,r} = \beta_{k,r} \cdot \mathbf{1}_m - \beta_k \in \mathbb{R}^m$. *We define the function (Gram matrix)* $H : \mathbb{R}^{d \times m} \to \mathbb{R}^{nd \times nd}$ *as following*

$$H(W) := \begin{bmatrix} H_{1,1} & H_{1,2} & \cdots & H_{1,d} \\ H_{2,1} & H_{2,2} & \cdots & H_{2,d} \\ \vdots & \vdots & \ddots & \vdots \\ H_{d,1} & H_{d,2} & \cdots & H_{d,d} \end{bmatrix},$$

*and for each* $k_1, k_2 \in [d]$, *we have* $H_{k_1,k_2} \in \mathbb{R}^{n \times n}$ *is defined as*

$$[H_{k_1,k_2}]_{i,j}(W) := \frac{1}{m} x_i^\top x_j \sum_{r=1}^m \langle v_{k_1,r}, \mathsf{S}_i \rangle \cdot m\mathsf{S}_{i,r} \cdot \langle v_{k_2,r}, \mathsf{S}_j \rangle \cdot m\mathsf{S}_{j,r}.$$

*For any timestamp* $\tau$, *for simplicity, we denote* $H(\tau) := H(W(\tau))$ *and denote* $H(0)$ *as* $H^*$.

## H.1  KERNEL PERTURBATION

The purpose of this section is to prove Lemma H.3. In the proof, we do not use concentration inequality. Please see Remark H.2 for more details.

**Remark H.2.** *In the proof of Lemma H.3, we do not use concentration bound as previous work (Song & Yang, 2019; Munteanu et al., 2022; Gao et al., 2023a). The reason is that we consider the worst case. In general,* $\mathbb{E}[H(W) - H(\widetilde{W})] \neq \mathbf{0}_{nd \times nd}$. *Thus, using the concentration bound may not gain any benefits.*

**Lemma H.3.** *If the following conditions hold*

- *Let* $C > 10$ *denote a sufficiently large constant*

- *Let* $B := \max\{C\sigma\sqrt{\log(nd/\delta)}, 1\}$.

- *Let* $R \in (0, 0.01)$.

- *Let* $x_i \in \mathbb{R}^d$ *and* $\|x_i\|_2 \leq 1$ *for all* $i \in [n]$.

- *Let* $\widetilde{W} = [\widetilde{w}_1, \cdots, \widetilde{w}_m] \in \mathbb{R}^{d \times m}$, *where* $\widetilde{w}_1, \cdots, \widetilde{w}_m$ *are are i.i.d. draw from* $\mathcal{N}(0, \sigma^2 I_d)$.

- *Let* $W = [w_1, \cdots, w_m] \in \mathbb{R}^{d \times m}$ *and satisfy* $\|\widetilde{w}_r - w_r\|_2 \leq R$ *for any* $r \in [m]$.

- *Let* $v_{k,r} = \beta_{k,r} \cdot \mathbf{1}_m - \beta_k \in \mathbb{R}^m$, *for any* $k \in [d]$ *and for any* $r \in [m]$. *Note that* $\beta_{k,r}$ *is the r-th in* $\beta_k$.

- *Let* $\alpha_i = \langle \mathbf{1}_m, \exp(W^\top x_i) \rangle$ *and* $\widetilde{\alpha}_i = \langle \mathbf{1}_m, \exp(\widetilde{W}^\top x_i) \rangle$, $\forall i \in [n]$.

- *Let* $H$ *be defined as Definition H.1.*

*Then, we have*

- *Part 1. Then with probability at least* $1 - \delta/\operatorname{poly}(nd)$,

$$|[H_{k_1,k_2}]_{i,j}(W) - [H_{k_1,k_2}]_{i,j}(\widetilde{W})| \leq 8R \cdot \exp(22B).$$

- *Part 2. Then with probability at least* $1 - \delta$, *we have*

$$\|H(W) - H(\widetilde{W})\|_F \leq 8R\sqrt{nd} \cdot \exp(22B).$$

*Proof.* For simplicity, we give the following notations:

- Note that $\widetilde{\mathsf{S}}_i := \exp(\widetilde{W}(\tau)^\top x_i) \cdot \widetilde{\alpha}_i^{-1}$.

- Note that $\widetilde{\beta}_k := \widetilde{W}_{k,*} \circ a$.

- Note that $\widetilde{v}_{k,r} := \widetilde{\beta}_{k,r} \cdot \mathbf{1}_m - \widetilde{\beta}_k$.

**Proof of Part 1.** We have

$$|[H_{k_1,k_2}]_{i,j}(W) - [H_{k_1,k_2}]_{i,j}(\widetilde{W})| = m x_i^\top x_j \sum_{r=1}^m (B_{1,r} + B_{2,r} + B_{3,r} + B_{4,r} + B_{5,r} + B_{6,r})$$

here, we define:

$$B_{1,r} := \langle v_{k_1,r}, \mathsf{S}_i \rangle \cdot \mathsf{S}_{i,r} \cdot \langle v_{k_2,r}, \mathsf{S}_j \rangle \cdot \mathsf{S}_{j,r} - \langle v_{k_1,r}, \mathsf{S}_i \rangle \cdot \mathsf{S}_{i,r} \cdot \langle v_{k_2,r}, \mathsf{S}_j \rangle \cdot \widetilde{\mathsf{S}}_{j,r}$$

$$B_{2,r} := \langle v_{k_1,r}, \mathsf{S}_i \rangle \cdot \mathsf{S}_{i,r} \cdot \langle v_{k_2,r}, \mathsf{S}_j \rangle \cdot \widetilde{\mathsf{S}}_{j,r} - \langle v_{k_1,r}, \mathsf{S}_i \rangle \cdot \mathsf{S}_{i,r} \cdot \langle v_{k_2,r}, \widetilde{\mathsf{S}}_j \rangle \cdot \widetilde{\mathsf{S}}_{j,r}$$

$$B_{3,r} := \langle v_{k_1,r}, \mathsf{S}_i \rangle \cdot \mathsf{S}_{i,r} \cdot \langle v_{k_2,r}, \widetilde{\mathsf{S}}_j \rangle \cdot \widetilde{\mathsf{S}}_{j,r} - \langle v_{k_1,r}, \mathsf{S}_i \rangle \cdot \mathsf{S}_{i,r} \cdot \langle \widetilde{v}_{k_2,r}, \widetilde{\mathsf{S}}_j \rangle \cdot \widetilde{\mathsf{S}}_{j,r}$$

$$B_{4,r} := \langle v_{k_1,r}, \mathsf{S}_i \rangle \cdot \mathsf{S}_{i,r} \cdot \langle \widetilde{v}_{k_2,r}, \widetilde{\mathsf{S}}_j \rangle \cdot \widetilde{\mathsf{S}}_{j,r} - \langle v_{k_1,r}, \mathsf{S}_i \rangle \cdot \widetilde{\mathsf{S}}_{i,r} \cdot \langle \widetilde{v}_{k_2,r}, \widetilde{\mathsf{S}}_j \rangle \cdot \widetilde{\mathsf{S}}_{j,r}$$

$$B_{5,r} := \langle v_{k_1,r}, \mathsf{S}_i \rangle \cdot \widetilde{\mathsf{S}}_{i,r} \cdot \langle \widetilde{v}_{k_2,r}, \widetilde{\mathsf{S}}_j \rangle \cdot \widetilde{\mathsf{S}}_{j,r} - \langle v_{k_1,r}, \widetilde{\mathsf{S}}_i \rangle \cdot \widetilde{\mathsf{S}}_{i,r} \cdot \langle \widetilde{v}_{k_2,r}, \widetilde{\mathsf{S}}_j \rangle \cdot \widetilde{\mathsf{S}}_{j,r}$$

$$B_{6,r} := \langle v_{k_1,r}, \widetilde{\mathsf{S}}_i \rangle \cdot \widetilde{\mathsf{S}}_{i,r} \cdot \langle \widetilde{v}_{k_2,r}, \widetilde{\mathsf{S}}_j \rangle \cdot \widetilde{\mathsf{S}}_{j,r} - \langle \widetilde{v}_{k_1,r}, \widetilde{\mathsf{S}}_i \rangle \cdot \widetilde{\mathsf{S}}_{i,r} \cdot \langle \widetilde{v}_{k_2,r}, \widetilde{\mathsf{S}}_j \rangle \cdot \widetilde{\mathsf{S}}_{j,r}$$

Before we bound all terms, we provide a tool as follows:

$$\begin{aligned}
\|v_{k,r} - \widetilde{v}_{k,r}\|_2^2 &= \sum_{r_1=1}^m (v_{k,r,r_1} - \widetilde{v}_{k,r,r_1})^2 \\
&= \sum_{r_1=1}^m (\beta_{k,r} - \beta_{k,r_1} - \widetilde{\beta}_{k,r} + \widetilde{\beta}_{k,r_1})^2 \\
&= \sum_{r_1=1}^m (a_r W_{k,r} - a_{r_1} W_{k,r} - a_r \widetilde{W}_{k,r} + a_{r_1} \widetilde{W}_{k,r})^2 \\
&= \sum_{r_1=1}^m (a_r(W_{k,r} - \widetilde{W}_{k,r}) + a_{r_1}(\widetilde{W}_{k,r_1} - W_{k,r_1}))^2 \\
&\leq \sum_{r_1=1}^m (|W_{k,r} - \widetilde{W}_{k,r}| + |\widetilde{W}_{k,r_1} - W_{k,r_1}|)^2 \\
&\leq \sum_{r_1=1}^m 4R^2 \\
&\leq m 4R^2
\end{aligned} \tag{10}$$

where the first step follows from the definition of $\ell_2$ norm, the second step follows from the definition of $v_{k,r}$, the third step follows from Definition F.5, the fourth and fifth steps follow from simple algebras, the sixth step follows from $\|w_r - v_r\|_\infty \leq \|w_r - v_r\|_2 \leq R$, the last step follows from simple algebras.

To bound $B_{1,r}$, we have

$$\begin{aligned}
|B_{1,r}| &:= |\langle v_{k_1,r}, \mathsf{S}_i \rangle \cdot \mathsf{S}_{i,r} \cdot \langle v_{k_2,r}, \mathsf{S}_j \rangle \cdot \mathsf{S}_{j,r} - \langle v_{k_1,r}, \mathsf{S}_i \rangle \cdot \mathsf{S}_{i,r} \cdot \langle v_{k_2,r}, \mathsf{S}_j \rangle \cdot \widetilde{\mathsf{S}}_{j,r}| \\
&= |\langle v_{k_1,r}, \mathsf{S}_i \rangle \cdot \mathsf{S}_{i,r} \cdot \langle v_{k_2,r}, \mathsf{S}_j \rangle \cdot (\mathsf{S}_{j,r} - \widetilde{\mathsf{S}}_{j,r})| \\
&\leq \frac{\exp(15B)}{m} \cdot |\mathsf{S}_{j,r} - \widetilde{\mathsf{S}}_{j,r}| \\
&\leq \frac{R \exp(19B + 3R)}{m^2}
\end{aligned}$$

where the first step follows from the definition of $B_{1,r}$, the second step follows from simple algebras, the third step follows from Part 6 of Lemma L.2 and $0 \leq \mathsf{S}_{i,r} \leq \frac{\exp(3B)}{m}$ by Part 11 of Lemma L.1, the last step follows from Part 12 of Lemma L.1.

To bound $B_{2,r}$, we have

$$
\begin{aligned}
|B_{2,r}| &:= |\langle v_{k_1,r}, \mathsf{S}_i \rangle \cdot \mathsf{S}_{i,r} \cdot \langle v_{k_2,r}, \mathsf{S}_j \rangle \cdot \widetilde{\mathsf{S}}_{j,r} - \langle v_{k_1,r}, \mathsf{S}_i \rangle \cdot \mathsf{S}_{i,r} \cdot \langle v_{k_2,r}, \widetilde{\mathsf{S}}_j \rangle \cdot \widetilde{\mathsf{S}}_{j,r}| \\
&= |\langle v_{k_1,r}, \mathsf{S}_i \rangle \cdot \mathsf{S}_{i,r} \cdot \langle v_{k_2,r}, \mathsf{S}_j - \widetilde{\mathsf{S}}_j \rangle \cdot \widetilde{\mathsf{S}}_{j,r}| \\
&\leq \frac{2B \exp(12B)}{m^2} \cdot |\langle \frac{1}{2B} v_{k_2,r}, \mathsf{S}_j - \widetilde{\mathsf{S}}_j \rangle| \\
&\leq \frac{2BR \exp(16B + 3R)}{m^2}
\end{aligned}
$$

where the first step follows from the definition of $B_{2,r}$, the second step follows from simple algebras, the third step follows from Part 6 of Lemma L.2 and $0 \leq \mathsf{S}_{i,r} \leq \frac{\exp(3B)}{m}$ by Part 11 of Lemma L.1, the last step follows from Part 13 of Lemma L.1 and $\|v_{k,r}\|_\infty \leq 2B$ by simple algebras.

To bound $B_{3,r}$, we have

$$
\begin{aligned}
|B_{3,r}| &:= |\langle v_{k_1,r}, \mathsf{S}_i \rangle \cdot \mathsf{S}_{i,r} \cdot \langle v_{k_2,r}, \widetilde{\mathsf{S}}_j \rangle \cdot \widetilde{\mathsf{S}}_{j,r} - \langle v_{k_1,r}, \mathsf{S}_i \rangle \cdot \mathsf{S}_{i,r} \cdot \langle \widetilde{v}_{k_2,r}, \widetilde{\mathsf{S}}_j \rangle \cdot \widetilde{\mathsf{S}}_{j,r}| \\
&= |\langle v_{k_1,r}, \mathsf{S}_i \rangle \cdot \mathsf{S}_{i,r} \cdot \langle v_{k_2,r} - \widetilde{v}_{k_2,r}, \widetilde{\mathsf{S}}_j \rangle \cdot \widetilde{\mathsf{S}}_{j,r}| \\
&\leq \frac{\exp(12B)}{m^2} \cdot |\langle v_{k_2,r} - \widetilde{v}_{k_2,r}, \widetilde{\mathsf{S}}_j \rangle| \\
&\leq \frac{2R \exp(15B)}{m^2}
\end{aligned}
$$

where the first step follows from the definition of $B_{3,r}$, the second step follows from simple algebras, the third step follows from Part 6 of Lemma L.2 and $0 \leq \mathsf{S}_{i,r} \leq \frac{\exp(3B)}{m}$ by Part 11 of Lemma L.1, the last step follows from Cauchy-Schwarz inequality, Eq. (10) and $\|\mathsf{S}_i\|_2 \leq \frac{\exp(3B)}{\sqrt{m}}$.

The proof of bounding $B_{4,r}$ is similar to the proof of bounding $B_{1,r}$, we have $|B_{4,r}| \leq \frac{R \exp(19B + 3R)}{m^2}$.

The proof of bounding $B_{5,r}$ is similar to the proof of bounding $B_{2,r}$, we have $|B_{5,r}| \leq \frac{2BR \exp(16B + 3R)}{m^2}$.

The proof of bounding $B_{6,r}$ is similar to the proof of bounding $B_{3,r}$, we have $|B_{6,r}| \leq \frac{2R \exp(15B)}{m^2}$.

Now we combine all terms, we have

$$
\begin{aligned}
|[H_{k_1,k_2}]_{i,j}(W) - [H_{k_1,k_2}]_{i,j}(\widetilde{W})| &= m x_i^\top x_j \sum_{r=1}^m (B_{1,r} + B_{2,r} + B_{3,r} + B_{4,r} + B_{5,r} + B_{6,r}) \\
&\leq m \sum_{r=1}^m (B_{1,r} + B_{2,r} + B_{3,r} + B_{4,r} + B_{5,r} + B_{6,r}) \\
&\leq m \sum_{r=1}^m (|B_{1,r}| + |B_{2,r}| + |B_{3,r}| + |B_{4,r}| + |B_{5,r}| + |B_{6,r}|) \\
&\leq m \sum_{r=1}^m \frac{8R \exp(22B)}{m^2} \\
&\leq 8R \cdot \exp(22B)
\end{aligned}
$$

where the second step follows from $\|x_i\|_2 \leq 1$, the third step follows from simple algebras, the fourth step follows from $R \leq B$, $B \leq \exp(B)$ and the combination of all terms, the last step follows from simple algebras.

**Proof of Part 2.** This proof follows from Part 1 of this Lemma and the definition of Frobenius norm. $\qquad \square$

## H.2 KERNEL PSD DURING TRAINING PROCESS

**Claim H.4.** *If the following conditions hold:*

- *Let $\lambda = \lambda_{\min}(H^*)$*

- *Let $C > 10$ denote a sufficiently large constant*

- *Let $B := \max\{C\sigma\sqrt{\log(nd/\delta)}, 1\}$.*

- *Let $\delta \in (0, 0.1)$.*

- *Let timestamp $\tau \geq 0$ denotes as a integer.*

- *Denote $H^*$ as $H(W)$ in Definition H.1.*

- *Denote $H(\tau)$ as $H(\widetilde{W})$ in Definition H.1.*

- *Let $D := 2\lambda^{-1} \cdot \exp(20B)\frac{\sqrt{nd}}{m}\|Y - \mathsf{F}(0)\|_F$*

- *Let $\|w_r(t) - w_r(0)\|_2 \leq D < R = \lambda/\operatorname{poly}(n, d, \exp(B))$, $\forall r \in [m], \forall t \geq 0$*

*Then, with a probability at least $1 - \delta$, we have*

$$\lambda_{\min}(H(\tau)) \geq \lambda/2$$

*Proof.* By Lemma H.3, with a probability at least $1 - \delta$, we have

$$\|H^* - H(\tau)\|_F \leq 8R\sqrt{nd}\exp(22B)$$
$$\leq \lambda/2 \tag{11}$$

where the first step follows from Part 2 of Lemma H.3, the second step follows by choice of $R$.

By eigenvalue perturbation theory, we have

$$\begin{aligned}
\lambda_{\min}(H(\tau)) &\geq \lambda_{\min}(H^*) - \|H(\tau) - H^*\| \\
&\geq \lambda_{\min}(H^*) - \|H(\tau) - H^*\|_F \\
&\geq \lambda_{\min}(H^*) - \lambda/2 \\
&\geq \lambda/2
\end{aligned}$$

where the first step comes from triangle inequality, the second step is due to Frobenius norm, the third step is due to Eq. (11), the last step follows from $\lambda_{\min}(H^*) = \lambda$. $\square$

## I LOSS DECOMPOSITION

In this section, we provide the lemma below to decompose it into five terms, and then we will give bounds to four terms.

**Lemma I.1.** *Assuming the following condition is met:*

- *Let $W \in \mathbb{R}^{d \times m}$ and $a \in \mathbb{R}^m$ as Definition F.1.*

- *Let $\lambda = \lambda_{\min}(H^*)$*

- *For $i, j \in [n]$ and $k_1, k_2 \in [d]$.*

- *Let $\theta_{k,i}(\tau) \in \mathbb{R}^m$ be defined as Definition G.6.*

- *Let $\mathsf{u}_i(\tau) \in \mathbb{R}^m$ be defined as Definition G.2.*

- *Denote $\mathsf{F}(\tau) \in \mathbb{R}^{n \times d}$ as Definition G.8.*

- *Let $Y \in \mathbb{R}^{n \times d}$ denote the labels.*

- *Let $\eta > 0$ denote the learning rate.*

- *Let scalar $v_{0,k,i} \in \mathbb{R}$ be defined as follows*

$$v_{0,k,i} := m \sum_{r \in [m]} (\theta_{k,i,r}(\tau + 1) - \theta_{k,i,r}(\tau)) \cdot u_{i,r}(\tau + 1)$$

- *Let scalar $v_{1,k,i} \in \mathbb{R}$ be defined as follows*

$$v_{1,k,i} := m \sum_{r=1}^{m} \theta_{k,i,r}(\tau) \cdot u_{i,r}(\tau) \cdot (-\eta \langle \Delta w_r(\tau), x_i \rangle)$$

- *Let scalar $v_{2,k,i} \in \mathbb{R}$ be defined as follows*

$$v_{2,k,i} := m \sum_{r=1}^{m} \theta_{k,i,r}(\tau) \cdot u_{i,r}(\tau) \cdot \eta^2 \cdot \Theta(1) \cdot \langle \Delta w_r(\tau), x_i \rangle^2$$

- **Gradient Property.** $\eta \|\Delta w_r(i)\|_2 \leq 0.01, \forall r \in [m], \forall i \in [\tau]$

- $C_0 = 2\langle \text{vec}(\mathsf{F}(\tau) - Y), \text{vec}(v_0) \rangle$

- $C_1 = 2\langle \text{vec}(\mathsf{F}(\tau) - Y), \text{vec}(v_1) \rangle$

- $C_2 = 2\langle \text{vec}(\mathsf{F}(\tau) - Y), \text{vec}(v_2) \rangle$

- $C_3 = \|\mathsf{F}(\tau + 1) - \mathsf{F}(\tau)\|_F^2$

*Then, we can show*

$$\|\mathsf{F}(\tau + 1) - Y\|_F^2 = \|\mathsf{F}(\tau) - Y\|_F^2 + C_0 + C_1 + C_2 + C_3.$$

*Proof.* The expression $\|Y - \mathsf{F}(\tau + 1)\|_F^2 = \|\text{vec}(Y - \mathsf{F}(\tau + 1))\|_2^2$ can be rewritten in the following:

$$
\begin{aligned}
& \| \text{vec}(Y - \mathsf{F}(\tau + 1)) \|_2^2 \\
&= \| \text{vec}(Y - \mathsf{F}(\tau) - (\mathsf{F}(\tau + 1) - \mathsf{F}(\tau))) \|_2^2 \\
&= \| \text{vec}(Y - \mathsf{F}(\tau)) \|_2^2 - 2 \text{vec}(Y - \mathsf{F}(\tau))^\top \text{vec}(\mathsf{F}(\tau + 1) - \mathsf{F}(\tau)) \\
&\quad + \| \text{vec}(\mathsf{F}(\tau + 1) - \mathsf{F}(\tau)) \|_2^2.
\end{aligned}
\tag{12}
$$

where the first step follows from simple algebra, the last step follows from simple algebra.

Recall the update rule (Definition G.13),

$$w_r(\tau + 1) = w_r(\tau) - \eta \cdot \Delta w_r(\tau)$$

In the following manner, $\forall k \in [d]$, we can express $\mathsf{F}_k(\tau + 1) - \mathsf{F}_k(\tau) \in \mathbb{R}^n$:

$$
\begin{aligned}
& \mathsf{F}_{k,i}(\tau + 1) - \mathsf{F}_{k,i}(\tau) \\
&= m \sum_{r \in [m]} (\theta_{k,i,r}(\tau + 1) u_{i,r}(\tau + 1) - \theta_{k,i,r}(\tau) u_{i,r}(\tau)) \\
&= + \sum_{r \in [m]} (\theta_{k,i,r}(\tau + 1) - \theta_{k,i,r}(\tau)) \cdot u_{i,r}(\tau + 1) \\
&\quad + m \sum_{r \in [m]} \theta_{k,i,r} \cdot (u_{i,r}(\tau + 1) - u_{i,r}(\tau)) \\
&= + \sum_{r \in [m]} (\theta_{k,i,r}(\tau + 1) - \theta_{k,i,r}(\tau)) \cdot u_{i,r}(\tau + 1)
\end{aligned}
$$

$$+ m \sum_{r \in [m]} \theta_{k,i,r}(\tau) \cdot \mathsf{u}_{i,r}(\tau) \cdot (\exp(-\eta \langle \Delta w_r(\tau), x_i \rangle) - 1)$$

$$= + \sum_{r \in [m]} (\theta_{k,i,r}(\tau + 1) - \theta_{k,i,r}(\tau)) \cdot \mathsf{u}_{i,r}(\tau + 1)$$

$$+ m \sum_{r \in [m]} \theta_{k,i,r}(\tau) \mathsf{u}_{i,r}(\tau) \cdot (-\eta \langle \Delta w_r(\tau), x_i \rangle + \Theta(1) \eta^2 \langle \Delta w_r(\tau), x_i \rangle^2)$$

$$= v_{0,k,i} + v_{1,k,i} + v_{2,k,i} \tag{13}$$

where the first step is due to the definition of $\mathsf{F}_{k,i}(\tau)$, the second step is from the simple algebra, the third step is due to $|\eta \Delta w_r(\tau)^\top x_i| \leq 0.01$ (due to **Gradient Property** and $\|x_i\|_2 \leq 1$), the fourth step follows from the Taylor series approximation, the last step follows from

$$v_{0,k,i} := m \sum_{r \in [m]} (\theta_{k,i,r}(\tau + 1) - \theta_{k,i,r}(\tau)) \cdot \mathsf{u}_{i,r}(\tau + 1)$$

$$v_{1,k,i} := m \sum_{r=1}^{m} \theta_{k,i,r}(\tau) \cdot \mathsf{u}_{i,r}(\tau) \cdot (-\eta \langle \Delta w_r(\tau), x_i \rangle)$$

$$v_{2,k,i} := m \sum_{r=1}^{m} \theta_{k,i,r}(\tau) \cdot \mathsf{u}_{i,r}(\tau) \cdot \eta^2 \cdot \Theta(1) \cdot \langle \Delta w_r(\tau), x_i \rangle^2$$

Here $v_{0,k,i}$ and $v_{1,k,i}$ are linear in $\eta$ and $v_{2,k,i}$ is quadratic in $\eta$. Thus, $v_{0,k,i}$ and $v_{1,k,i}$ are the first order term, and $v_{2,k,i}$ is the second order term.

We can rewrite the second term in the Eq. (12) above as below:

$$\langle \mathrm{vec}(Y - \mathsf{F}(\tau)), \mathrm{vec}(\mathsf{F}(\tau + 1) - \mathsf{F}(\tau)) \rangle$$
$$= \langle \mathrm{vec}(Y - \mathsf{F}(\tau)), \mathrm{vec}(v_0 + v_1 + v_2) \rangle$$
$$= \langle \mathrm{vec}(Y - \mathsf{F}(\tau)), \mathrm{vec}(v_0) \rangle + \langle \mathrm{vec}(Y - \mathsf{F}(\tau)), \mathrm{vec}(v_1) \rangle + \langle \mathrm{vec}(Y - \mathsf{F}(\tau)), \mathrm{vec}(v_2) \rangle$$

where the first step follows from Eq.(13), the second step follows from simple algebras.

Therefore, we can conclude that

$$\|\mathsf{F}(\tau + 1) - Y\|_F^2 = \|\mathsf{F}(\tau) - Y\|_F^2 + C_0 + C_1 + C_2 + C_3.$$

$\square$

The below lemma analyzes the first-order term that is making progress.

**Lemma I.2** (Progress terms)**.** *If the following conditions hold*

- *Let $\lambda = \lambda_{\min}(H^*)$*

- *Let $C > 10$ denote a sufficiently large constant*

- *Let $B := \max\{C\sigma\sqrt{\log(nd/\delta)}, 1\}$.*

- *Let $\delta \in (0, 0.1)$.*

- *Let $m \geq \Omega(\lambda^{-2} n^2 d^2 \exp(30B) \sqrt{\log(nd/\delta)})$*

- *Let $r \in [m]$, let $i, j \in [n]$, let $k, k_2 \in [d]$.*

- *Let $\beta_k(\tau) \in \mathbb{R}^m$ be defined as Definition F.5.*

- *Let $\theta_{k,i}(\tau) \in \mathbb{R}^m$ be defined as Definition G.6.*

- *Let $\mathsf{u}_i(\tau) \in \mathbb{R}^m$ be defined as Definition G.2.*

- *Let $\mathsf{S}_i(\tau) \in \mathbb{R}^m$ be defined as Definition G.7.*

- *Let $v_{k,r} := \beta_{k,r}(\tau) \cdot \mathbf{1}_m - \beta_k(\tau) \in \mathbb{R}^m$*

- *Denote $\mathsf{F}(\tau) \in \mathbb{R}^{n \times d}$ as Definition G.8.*

- *Let $Y \in \mathbb{R}^{n \times d}$ denote the labels.*

- *Let $\eta > 0$ denote the learning rate.*

- *Let scalar $v_{1,1,k,i} \in \mathbb{R}$ be defined as follows (we omit $(\tau)$ in the following terms)*

$$v_{1,1,k,i} = m^2 \sum_{r \in [m]} \theta_{k,i,r}(\tau) \cdot \mathsf{u}_{i,r}(\tau)$$

$$\cdot (-\eta \sum_{j=1}^{n} \sum_{k_2=1}^{d} (\mathsf{F}_{k_2,j}(\tau) - y_{k_2,j}) \cdot \left( (\langle v_{k_2,r}, \mathsf{S}_j(\tau) \rangle) \cdot \mathsf{S}_{j,r}(\tau) \right) \cdot x_j^\top) x_i$$

- *Let $C_{1,1} := 2\langle \mathrm{vec}(\mathsf{F}(\tau) - Y), \mathrm{vec}(v_{1,1}) \rangle$*

*Then, we have*

- $C_{1,1} \leq -1.6m\eta \, \mathrm{vec}(\mathsf{F}(\tau) - Y)^\top H(\tau) \, \mathrm{vec}(\mathsf{F}(\tau) - Y)$

*Proof.* We have

$$v_{1,1,k,i} = m^2 \sum_{r \in [m]} \theta_{k,i,r}(\tau) \cdot \mathsf{u}_{i,r}(\tau)$$

$$\cdot (-\eta \sum_{j=1}^{n} \sum_{k_2=1}^{d} (\mathsf{F}_{k_2,j}(\tau) - y_{k_2,j}) \cdot \left( (\langle v_{k_2,r}, \mathsf{S}_j(\tau) \rangle) \cdot \mathsf{S}_{j,r}(\tau) \right) \cdot x_j^\top) x_i$$

$$= m^2 \sum_{r \in [m]} \beta_{k,r}(\tau) \cdot \alpha_i(\tau)^{-1} \cdot \mathsf{u}_{i,r}(\tau)$$

$$\cdot (-\eta \sum_{j=1}^{n} \sum_{k_2=1}^{d} (\mathsf{F}_{k_2,j}(\tau) - y_{k_2,j}) \cdot \left( (\langle v_{k_2,r}, \mathsf{S}_j(\tau) \rangle) \cdot \mathsf{S}_{j,r}(\tau) \right) \cdot x_j^\top) x_i$$

$$= m^2 \sum_{r \in [m]} \beta_{k,r}(\tau) \cdot \mathsf{S}_{i,r}(\tau)$$

$$\cdot (-\eta \sum_{j=1}^{n} \sum_{k_2=1}^{d} (\mathsf{F}_{k_2,j}(\tau) - y_{k_2,j}) \cdot \left( (\langle v_{k_2,r}, \mathsf{S}_j(\tau) \rangle) \cdot \mathsf{S}_{j,r}(\tau) \right) \cdot x_j^\top) x_i$$

$$= m^2 \sum_{r \in [m]} \langle \beta_{k,r}(\tau) \cdot \mathbf{1}_m, \mathsf{S}_i(\tau) \rangle \cdot \mathsf{S}_{i,r}(\tau)$$

$$\cdot (-\eta \sum_{j=1}^{n} \sum_{k_2=1}^{d} (\mathsf{F}_{k_2,j} - y_{k_2,j}) \cdot \left( (\langle v_{k_2,r}, \mathsf{S}_j(\tau) \rangle) \cdot \mathsf{S}_{j,r}(\tau) \right) \cdot x_j^\top) x_i$$

$$= m^2 \sum_{r \in [m]} (\langle v_{k,r}, \mathsf{S}_i(\tau) \rangle + \langle \beta_k(\tau), \mathsf{S}_i(\tau) \rangle) \cdot \mathsf{S}_{i,r}$$

$$\cdot (-\eta \sum_{j=1}^{n} \sum_{k_2=1}^{d} (\mathsf{F}_{k_2,j}(\tau) - y_{k_2,j}) \cdot \left( (\langle v_{k_2,r}, \mathsf{S}_j(\tau) \rangle) \cdot \mathsf{S}_{j,r}(\tau) \right) \cdot x_j^\top) x_i$$

$$= m^2 (Q_{1,1,k,i} + Q_{1,2,k,i})$$

where the first step follows from the definition of $v_{1,1,k,i}$, the second step follows from Definition G.6, the third step follows from Definition G.7, the fourth step follows from $\langle \beta_{k,r}(\tau) \cdot \mathbf{1}_m, \mathsf{S}_i \rangle = \beta_{k,r}(\tau)$, the fifth step follows from the definition of $v_k$ for $k \in [d]$ and simple algebras, the last step holds since we define

$$Q_{1,1,k,i} := \sum_{r \in [m]} \langle v_{k,r}, \mathsf{S}_i(\tau) \rangle \cdot \mathsf{S}_{i,r}(\tau)$$

$$\cdot (-\eta \sum_{j=1}^{n} \sum_{k_2=1}^{d} (\mathsf{F}_{k_2,j}(\tau) - y_{k_2,j}) \cdot \left( (\langle v_{k_2,r}, \mathsf{S}_j(\tau) \rangle) \cdot \mathsf{S}_{j,r}(\tau) \right) \cdot x_j^{\top}) x_i,$$

$$Q_{1,2,k,i} := \sum_{r \in [m]} \langle \beta_k(\tau), \mathsf{S}_i(\tau) \rangle \cdot \mathsf{S}_{i,r}(\tau)$$

$$\cdot (-\eta \sum_{j=1}^{n} \sum_{k_2=1}^{d} (\mathsf{F}_{k_2,j}(\tau) - y_{k_2,j}) \cdot \left( (\langle v_{k_2,r}, \mathsf{S}_j(\tau) \rangle) \cdot \mathsf{S}_{j,r}(\tau) \right) \cdot x_j^{\top}) x_i.$$

**Bounding first term.** Then for the first term $Q_{1,1,k,i}$, we have its quantity

$$\sum_{i=1}^{n} \sum_{k=1}^{d} Q_{1,1,k,i}(\mathsf{F}_{k,i}(\tau) - y_{k,i}) = -\frac{1}{m} \eta \operatorname{vec}(\mathsf{F}(\tau) - Y)^{\top} H(\tau) \operatorname{vec}(\mathsf{F}(\tau) - Y)$$

where this step follows from Definition H.1.

**Bounding second term.** On the other hand, for the second term $Q_{1,2,k,i}$, we have its quantity,

$$|\sum_{i=1}^{n} \sum_{k=1}^{d} Q_{1,2,k,i}(\mathsf{F}_{k,i}(\tau) - y_{k,i})|$$

$$\leq \eta |\frac{\exp(9B)}{m^3} \sum_{i=1}^{n} \sum_{j=1}^{n} \sum_{r=1}^{m} \sum_{k=1}^{d} \sum_{k_2=1}^{d} \sigma_r C_{k,k_2,r} (\mathsf{F}_{k,i}(\tau) - y_{k,i})(\mathsf{F}_{k_2,j}(\tau) - y_{k_2,j})|$$

$$\leq \eta \frac{\exp(9B)}{m^3} \cdot |\sum_{r=1}^{m} \sigma_r \max_{k,k_2 \in [d]} C_{k,k_2,r}| \cdot \|(\mathsf{F}(\tau) - Y) \otimes (\mathsf{F}(\tau) - Y)\|_1$$

$$\leq \eta \frac{\exp(9B)}{m^3} \cdot |\sum_{r=1}^{m} \sigma_r \max_{k,k_2 \in [d]} C_{k,k_2,r}| \cdot \|\mathsf{F}(\tau) - Y\|_1^2$$

$$\leq \eta \frac{nd \exp(9B)}{m^3} \cdot |\sum_{r=1}^{m} \sigma_r \max_{k,k_2 \in [d]} C_{k,k_2,r}| \cdot \|\mathsf{F}(\tau) - Y\|_F^2$$

$$\leq \eta \frac{\exp(9B)}{m^3 \lambda} |\sum_{r=1}^{m} \sigma_r \max_{k,k_2 \in [d]} C_{k,k_2,r}| \cdot \operatorname{vec}(\mathsf{F}(\tau) - Y)^{\top} H(\tau) \operatorname{vec}(\mathsf{F}(\tau) - Y)$$

where the first step follows from $0 \leq \mathsf{S}_{i,r} \leq \frac{\exp(3B)}{m}$ by Part 11 of Lemma L.1, $\|\mathsf{S}_i\|_2 \leq \frac{\exp(3B)}{\sqrt{m}}$, $\|x_i\| \leq 1$ and

$$C_{k,k_2,r} := \|\beta_k(\tau)\|_2 \cdot \|v_{k_2,r}\|_2, \sigma_r \in \{+1, -1\}$$

the second and third steps follow from the definition of Kronecker product, the fourth step follows from $\|U\|_1 \leq \sqrt{nd} \|U\|_F$ for $U \in \mathbb{R}^{n \times d}$, the last step follows from $\operatorname{vec}(\mathsf{F}(\tau) - Y)^{\top} H(\tau) \operatorname{vec}(\mathsf{F}(\tau) - Y) \geq \lambda \|\mathsf{F} - Y\|_F^2$.

Thus, by following Part 2 and Part 3 of Lemma L.2, we have

$$C_{k,k_2,r} = \|\beta_k(\tau)\|_2 \cdot \|v_{k_2,r}\|_2 \leq 2mB^2.$$

Besides, we apply Hoeffding inequality (Lemma E.4) to all random variables $\sigma_r \max_{k,k_2 \in [d]} C_{k,k_2,r}$ for $r \in [m]$, especially $\mathbb{E}[\sum_{r=1}^{m} \sigma_r \max_{k,k_2 \in [d]} C_{k,k_2,r}] = 0$ due to the symmetry of $a_r$, we have

$$|\sum_{i=1}^{n} \sum_{k=1}^{d} Q_{1,2,k,i}(\mathsf{F}_{k,i}(\tau) - y_{k,i})|$$

$$\leq C\eta \frac{nd \exp(9B)}{m^3 \lambda} \cdot \operatorname{vec}(\mathsf{F}(\tau) - Y)^{\top} H(\tau) \operatorname{vec}(\mathsf{F}(\tau) - Y) \cdot mB^2 \sqrt{m \log(nd/\delta)}$$

with probability at least $1 - \delta/\operatorname{poly}(nd)$.

Note that by Lemma condition, we have

$$C \frac{nd \exp(9B)}{m^3 \lambda} \cdot mB^2 \sqrt{m \log(nd/\delta)} \le 0.2 \frac{1}{m}.$$

Finally, we complete the proof with the result

$$C_{1,1} \le -1.6m\eta \operatorname{vec}(\mathsf{F}(\tau) - Y)^\top H(\tau) \operatorname{vec}(\mathsf{F}(\tau) - Y)$$

$\square$

Below, we prove all other terms are small when $m$ is large enough compared to the progressive term.

**Lemma I.3** (Minor effects on non-progress term)**.** *If the following*

- *Let $m \ge \Omega(\lambda^{-2} n^2 d^2 \exp(30B) \sqrt{\log(nd/\delta)})$.*

- *Let $r \in [m]$, let $i, j \in [n]$, let $k, k_2 \in [d]$*

- *Let scalar $v_{0,k,i} \in \mathbb{R}$ be defined as follows*

$$v_{0,k,i} := m \sum_{r \in [m]} (\theta_{k,i,r}(\tau+1) - \theta_{k,i,r}(\tau)) \cdot \mathsf{u}_{i,r}(\tau+1)$$

- *Let scalar $v_{1,2,k,i} \in \mathbb{R}$ be defined as follows (we omit $(\tau)$ in the following terms)*

$$v_{1,2,k,i} = m^2 \sum_{r \in [m]} \theta_{k,i,r}(\tau) \cdot \mathsf{u}_{i,r}(\tau) \cdot (-\eta \sum_{j=1}^n \sum_{k_2=1}^d (\mathsf{F}_{k_2,j}(\tau) - y_{k_2,j}) \cdot a_r \mathsf{S}_{j,r}(\tau) e_{k_2}^\top) x_i$$

- *Let scalar $v_{2,k,i} \in \mathbb{R}$ be defined as follows*

$$v_{2,k,i} := m \sum_{r=1}^m \theta_{k,i,r}(\tau) \cdot \mathsf{u}_{i,r}(\tau) \cdot \eta^2 \cdot \Theta(1) \cdot \langle \Delta w_r(\tau), x_i \rangle^2$$

- *Let $C_0 := 2\langle \operatorname{vec}(\mathsf{F}(\tau) - Y), \operatorname{vec}(v_0) \rangle$*

- *Let $C_{1,2} := 2\langle \operatorname{vec}(\mathsf{F}(\tau) - Y), \operatorname{vec}(v_{1,2}) \rangle$*

- *Let $C_2 := 2\langle \operatorname{vec}(\mathsf{F}(\tau) - Y), \operatorname{vec}(v_2) \rangle$*

- *Let $C_3 := \|\mathsf{F}(\tau+1) - \mathsf{F}(\tau)\|_F^2$*

*Then, we have*

- *$|C_0| \le 0.1 m\eta\lambda \cdot \|\mathsf{F}(\tau) - Y\|_F^2$*

- *$|C_{1,2}| \le 0.1 m\eta\lambda \cdot \|\mathsf{F}(\tau) - Y\|_F^2$*

- *$|C_2| \le \eta^2 m \cdot n^2 d^2 \exp(16B) \cdot \|\mathsf{F}(\tau) - Y\|_F^2$*

- *$|C_3| \le \eta^2 m^2 \cdot \|\mathsf{F}(\tau) - Y\|_F^2$*

*Proof.* This proof follows from Lemma I.4, Lemma I.5, Lemma I.6 and Lemma I.7. $\square$

## I.1 BOUNDING $C_0$

**Lemma I.4.** *If the following conditions hold*

- *Let $\lambda = \lambda_{\min}(H^*)$*

- *Let $C > 10$ denote a sufficiently large constant*

- *Let $B := \max\{C\sigma\sqrt{\log(nd/\delta)}, 1\}$.*

- *Let $\delta \in (0, 0.1)$.*

- *Let $m \geq \Omega(\lambda^{-2}n^2d^2\exp(30B)\sqrt{\log(nd/\delta)})$.*

- *Let $r \in [m]$, let $i, j \in [n]$, let $k, k_1 \in [d]$.*

- *Let $\beta_k(\tau) \in \mathbb{R}^m$ be defined as Definition F.5.*

- *Let $\alpha_i(\tau) \in \mathbb{R}$ be defined as Definition F.3.*

- *Let $\theta_{k,i}(\tau) \in \mathbb{R}^m$ be defined as Definition G.6.*

- *Let $\mathsf{u}_i(\tau) \in \mathbb{R}^m$ be defined as Definition G.2.*

- *Let $\mathsf{S}_i(\tau) \in \mathbb{R}^m$ be defined as Definition G.7.*

- *Let $v_k := \beta_{k,r}(\tau) \cdot \mathbf{1}_m - \beta_k(\tau) \in \mathbb{R}^m$*

- *Denote $\mathsf{F}(\tau) \in \mathbb{R}^{n \times d}$ as Definition G.8.*

- *Let $Y \in \mathbb{R}^{n \times d}$ denote the labels.*

- *Let $\eta \in (0, 1/m)$ denote the learning rate.*

- *Let scalar $v_{0,k,i} \in \mathbb{R}$ be defined as follows (we omit $(\tau)$ in the following terms)*

$$v_{0,k,i} = m \sum_{r \in [m]} (\theta_{k,i,r}(\tau+1) - \theta_{k,i,r}(\tau)) \cdot \mathsf{u}_{i,r}(\tau+1)$$

- *Let $C_0 := 2\langle \mathrm{vec}(\mathsf{F}(\tau) - Y), \mathrm{vec}(v_0)\rangle$*

*Then, with a probability at least $1 - \delta/\operatorname{poly}(nd)$, we have*

$$|C_0| \leq 0.1\eta m\lambda\|\mathsf{F}(\tau) - Y\|_F^2.$$

*Proof.* By Claim G.12, we have

$$\Delta w_r(\tau) = m \sum_{i=1}^{n} \sum_{k=1}^{d} (\mathsf{F}_{k,i}(\tau) - y_{k,i}) \cdot \Big( \langle v_{k,r}(\tau), \mathsf{S}_i(\tau)\rangle \cdot \mathsf{S}_{i,r}(\tau) \cdot x_i + a_r \mathsf{S}_{i,r}(\tau)e_k \Big)$$

Then the $k_1$-th entry $\Delta w_{r,k}(\tau)$ for $k_1 \in [d]$ should be

$$\Delta w_{r,k_1}(\tau) = m \sum_{i=1}^{n} \sum_{k=1}^{d} (\mathsf{F}_{k,i}(\tau) - y_{k,i}) \cdot \Big( \langle v_{k,r}(\tau), \mathsf{S}_i(\tau)\rangle \cdot \mathsf{S}_{i,r}(\tau) \cdot x_{i,k_1} + a_r \mathsf{S}_{i,r}(\tau)e_{k,k_1} \Big)$$

$$\tag{14}$$

We have

$$v_{0,k,i} = m \sum_{r \in [m]} (\theta_{k,i,r}(\tau+1) - \theta_{k,i,r}(\tau)) \cdot \mathsf{u}_{i,r}(\tau+1)$$

$$= m \sum_{r \in [m]} (\beta_{k,r}(\tau+1)\alpha_i(\tau+1)^{-1} - \beta_{k,r}(\tau)\alpha_i(\tau)^{-1}) \cdot \mathsf{u}_{i,r}(\tau+1)$$

$$= m \sum_{r \in [m]} (\beta_{k,r}(\tau+1)\alpha_i(\tau+1)^{-1} - \beta_{k,r}(\tau+1)\alpha_i(\tau)^{-1}$$

$$+ \beta_{k,r}(\tau+1)\alpha_i(\tau)^{-1} - \beta_{k,r}(\tau)\alpha_i(\tau)^{-1}) \cdot \mathsf{u}_{i,r}(\tau+1)$$

$$= m \sum_{r \in [m]} (\beta_{k,r}(\tau+1) \cdot (\alpha_i(\tau+1)^{-1} - \alpha_i(\tau)^{-1})$$

$$+ (\beta_{k,r}(\tau+1) - \beta_{k,r}(\tau)) \cdot \alpha_i(\tau)^{-1}) \cdot \mathsf{u}_{i,r}(\tau+1)$$
$$= m(Q_{0,1,k,i} + Q_{0,2,k,i})$$

where the first step follows from the definition of $v_{0,k,i}$, the second step follows from Definition G.6, the third and fourth steps follow from simple algebras, the last step hold since we define

$$Q_{0,1,k,i} := \sum_{r\in[m]} \beta_{k,r}(\tau+1) \cdot (\alpha_i(\tau+1)^{-1} - \alpha_i(\tau)^{-1}) \cdot \mathsf{u}_{i,r}(\tau+1),$$

$$Q_{0,2,k,i} := \sum_{r\in[m]} (\beta_{k,r}(\tau+1) - \beta_{k,r}(\tau)) \cdot \alpha_i(\tau)^{-1}) \cdot \mathsf{u}_{i,r}(\tau+1).$$

**Bounding first term.** For the first term $Q_{0,1,k,i}$, we have its quantity

$$|\sum_{i=1}^{n}\sum_{k=1}^{d} Q_{0,1,k,i}(\mathsf{F}_{k,i}(\tau) - y_{k,i})|$$

$$\leq |\sum_{i=1}^{n}\sum_{k=1}^{d}\sum_{r=1}^{m} \beta_{k,r}(\tau+1) \cdot (\alpha_i(\tau+1)^{-1} - \alpha_i(\tau)^{-1}) \cdot \mathsf{u}_{i,r}(\tau+1)(\mathsf{F}_{k,i}(\tau) - y_{k,i})|$$

$$\leq \exp(B) \cdot |\sum_{i=1}^{n}\sum_{k=1}^{d}\sum_{r=1}^{m} \beta_{k,r}(\tau+1) \cdot (\alpha_i(\tau+1)^{-1} - \alpha_i(\tau)^{-1})(\mathsf{F}_{k,i}(\tau) - y_{k,i})|$$

$$\leq B\exp(B) \cdot |\sum_{i=1}^{n}\sum_{k=1}^{d}\sum_{r=1}^{m} a_r(\alpha_i(\tau+1)^{-1} - \alpha_i(\tau)^{-1}) \cdot (\mathsf{F}_{k,i}(\tau) - y_{k,i})|$$

$$\leq B\exp(B) \cdot |\sum_{r=1}^{m} a_r(\alpha_i(\tau+1)^{-1} - \alpha_i(\tau)^{-1})| \cdot \sqrt{nd}\|\mathsf{F}(\tau) - Y\|_F \tag{15}$$

where the first step follows from the definition of $Q_{0,1,k,i}$, the second step follows from Part 4 of Lemma L.1 and Definition G.2, the third step follows from Part 1 of Lemma L.1 and $\|U\|_1 \leq \sqrt{nd}\|U\|_F$ for $U \in \mathbb{R}^{n\times d}$.

By Part 2 of Lemma I.9, we have

$$\alpha_i(\tau+1)^{-1} - \alpha_i(\tau)^{-1} \leq \eta\frac{\sqrt{nd}\exp(15B)}{m^3} \cdot \|\mathsf{F}(\tau) - Y\|_F + \eta^2\frac{nd\exp(27B)}{\sqrt{m}} \cdot \|\mathsf{F}(\tau) - Y\|_F.$$

Then we apply Hoeffding inequality (Lemma E.4) to random variables $a_r(\alpha_i(\tau+1)^{-1} - \alpha_i(\tau)^{-1})$ for $r \in [m]$, and by $\mathbb{E}[\sum_{r=1}^{m} a_r(\alpha_i(\tau+1)^{-1} - \alpha_i(\tau)^{-1})] = 0$, we have

$$|\sum_{r=1}^{m} a_r(\alpha_i(\tau+1)^{-1} - \alpha_i(\tau)^{-1})|$$

$$\leq (\eta\frac{\sqrt{nd}\exp(15B)}{m^3} + \eta^2\frac{nd\exp(27B)}{\sqrt{m}}) \cdot \|\mathsf{F}(\tau) - Y\|_F \cdot \sqrt{m\log(nd/\delta)}. \tag{16}$$

with probability at least $1 - \delta/\operatorname{poly}(nd)$.

Through combining Eq. (16) and Eq.(15), we can show that

$$|\sum_{i=1}^{n}\sum_{k=1}^{d} Q_{0,1,k,i}(\mathsf{F}_{k,i}(\tau) - y_{k,i})|$$

$$\leq (\eta\frac{nd\exp(17B)}{m^3} \cdot \|\mathsf{F}(\tau) - Y\|_F^2 + \eta^2\frac{nd\sqrt{nd}\exp(29B)}{\sqrt{m}} \cdot \|\mathsf{F}(\tau) - Y\|_F^2) \cdot \sqrt{m\log(nd/\delta)}$$

with a probability at least $1 - \delta/\operatorname{poly}(nd)$.

Thus, by Lemma condition, we can show

$$\eta\frac{nd\exp(17B)}{m^3} \cdot \sqrt{m\log(nd/\delta)} \leq 0.01\eta\lambda,$$

$$\eta^2 \frac{nd\sqrt{nd}\exp(29B)}{\sqrt{m}} \cdot \sqrt{m\log(nd/\delta)} \le \eta\frac{nd\sqrt{nd}\exp(29B)}{m} \cdot \sqrt{\log(nd/\delta)} \le 0.01\eta\lambda.$$

**Bounding second term.** On the other hand, for the second term $Q_{0,2,k,i}$, we have its quantity

$$|\sum_{i=1}^{n}\sum_{k=1}^{d} Q_{0,2,k,i}(\mathsf{F}_{k,i}(\tau) - y_{k,i})|$$

$$\le |\sum_{i=1}^{n}\sum_{k=1}^{d}\sum_{r=1}^{m}(\beta_{k,r}(\tau+1) - \beta_{k,r}(\tau)) \cdot \alpha_i(\tau)^{-1} \cdot \mathsf{u}_{i,r}(\tau+1) \cdot (\mathsf{F}_{k,i}(\tau) - y_{k,i})|$$

$$\le \exp(B) \cdot |\sum_{i=1}^{n}\sum_{k=1}^{d}\sum_{r=1}^{m}(\beta_{k,r}(\tau+1) - \beta_{k,r}(\tau)) \cdot \alpha_i(\tau)^{-1} \cdot (\mathsf{F}_{k,i}(\tau) - y_{k,i})|$$

$$\le \frac{\exp(2B)}{m} \cdot |\sum_{i=1}^{n}\sum_{k=1}^{d}\sum_{r=1}^{m}(\beta_{k,r}(\tau+1) - \beta_{k,r}(\tau)) \cdot (\mathsf{F}_{k,i}(\tau) - y_{k,i})|$$

$$\le \frac{\exp(2B)}{m} \cdot |\sum_{i=1}^{n}\sum_{k=1}^{d}\sum_{r=1}^{m}(W_{k,r}(\tau+1) \cdot a_r - W_{k,r}(\tau) \cdot a_r) \cdot (\mathsf{F}_{k,i}(\tau) - y_{k,i})|$$

$$\le \eta\frac{\exp(2B)}{m} \cdot |\sum_{i=1}^{n}\sum_{k=1}^{d}\sum_{r=1}^{m} a_r \cdot m \cdot \sum_{j=1}^{n}\sum_{k_1=1}^{d}(\mathsf{F}_{k_1,j}(\tau) - y_{k_1,j})$$

$$\cdot \Big(\langle v_{k_1,r}(\tau), \mathsf{S}_j(\tau)\rangle \cdot \mathsf{S}_{j,r}(\tau) \cdot x_{j,k} + a_r\mathsf{S}_{j,r}(\tau)e_{k_1,k}\Big) \cdot (\mathsf{F}_{k,i}(\tau) - y_{k,i})|$$

$$\le \eta\frac{\exp(5B)}{m} \cdot |\sum_{r=1}^{m}\sigma_r \max_{j,k,k_1\in[d]} C_{j,k,k_1,r}| \cdot \|(\mathsf{F}(\tau) - Y) \otimes (\mathsf{F}(\tau) - Y)\|_1$$

$$\le \eta\frac{\exp(5B)}{m} \cdot |\sum_{r=1}^{m}\sigma_r \max_{j,k,k_1\in[d]} C_{j,k,k_1,r}| \cdot \|\mathsf{F}(\tau) - Y\|_1^2$$

$$\le \eta\frac{nd\exp(5B)}{m} \cdot |\sum_{r=1}^{m}\sigma_r \max_{j,k,k_1\in[d]} C_{j,k,k_1,r}| \cdot \|\mathsf{F}(\tau) - Y\|_F^2$$

where the first step follows from the definition of $Q_{0,2,k,i}$, the second and third steps follow from Part 4 of Lemma L.1, the fourth step follows from Definition F.5, the fifth step follows from Eq.(14), the sixth step follows from the definition of Kronecker product, $1 \le \mathsf{S}_{i,r} \le \frac{\exp(3B)}{m}$ by Part 11 of Lemma L.1, $\|x_i\|_2 \le 1$ and defining

$$C_{j,k,k_1,r} := \langle \mathsf{S}_j, v_{k_1,r}\rangle + e_{k_1,k}, \sigma_r \in \{+1, -1\},$$

the seventh step follows from the definition of $\ell_1$ norm, the last step follows from $\|U\|_1 \le \sqrt{nd}\|U\|_F$ for $U \in \mathbb{R}^{n\times d}$.

Thus, by following Part 6 of Lemma L.2, we have

$$C_{j,k,k_1,r} = \langle \mathsf{S}_j, v_{k_1,r}\rangle + e_{k_1,k}$$
$$\le \exp(6B) + 1$$
$$\le \exp(7B)$$

where the last step follows from simple algebras.

We apply Hoeffding inequality (Lemma E.4) to $\sigma_r \max_{j,k,k_1\in[d]} C_{j,k,k_1,r}$ for $r \in [m]$.

By $\mathbb{E}[\sum_{r=1}^{m}\sigma_r \max_{j,k,k_1\in[d]} C_{j,k,k_1,r}] = 0$, we have

$$|\sum_{i=1}^{n}\sum_{k=1}^{d} Q_{0,2,k,i}(\mathsf{F}_{k,i}(\tau) - y_{k,i})| \le \eta\frac{nd\exp(5B)}{m} \cdot \|\mathsf{F}(\tau) - Y\|_F^2 \cdot \exp(6B)\sqrt{m\log(nd/\delta)}.$$

with probability at least $1 - \delta/\operatorname{poly}(nd)$.

Then, by Lemma condition, we have

$$\eta \frac{nd \exp(5B)}{m} \cdot \exp(7B)\sqrt{m \log(nd/\delta)} \leq 0.01\eta\lambda.$$

Now we can complete the proof by combining all terms, we have

$$|C_0| \leq 0.1\eta m\lambda \|\mathsf{F}(\tau) - Y\|_F^2.$$

$\square$

## I.2 Bounding $C_{1,2}$

**Lemma I.5.** *If the following conditions hold*

- *Let $\lambda = \lambda_{\min}(H^*)$*

- *Let $C > 10$ denote a sufficiently large constant*

- *Let $B := \max\{C\sigma\sqrt{\log(nd/\delta)}, 1\}$.*

- *Let $\delta \in (0, 0.1)$.*

- *Let $m \geq \Omega(\lambda^{-2}n^2d^2 \exp(30B)\sqrt{\log(nd/\delta)})$.*

- *Let $r \in [m]$, let $i, j \in [n]$, let $k, k_1 \in [d]$.*

- *Let $\beta_k(\tau) \in \mathbb{R}^m$ be defined as Definition F.5.*

- *Let $\alpha_i(\tau) \in \mathbb{R}$ be defined as Definition F.3.*

- *Let $\theta_{k,i}(\tau) \in \mathbb{R}^m$ be defined as Definition G.6.*

- *Let $\mathsf{u}_i(\tau) \in \mathbb{R}^m$ be defined as Definition G.2.*

- *Let $\mathsf{S}_i(\tau) \in \mathbb{R}^m$ be defined as Definition G.7.*

- *Let $v_k := \beta_{k,r}(\tau) \cdot \mathbf{1}_m - \beta_k(\tau) \in \mathbb{R}^m$*

- *Denote $\mathsf{F}(\tau) \in \mathbb{R}^{n \times d}$ as Definition G.8.*

- *Let $Y \in \mathbb{R}^{n \times d}$ denote the labels.*

- *Let $\eta > 0$ denote the learning rate.*

- *Let scalar $v_{1,2,k,i} \in \mathbb{R}$ be defined as follows (we omit $(\tau)$ in the following terms)*

$$v_{1,2,k,i} = m^2 \sum_{r \in [m]} \theta_{k,i,r}(\tau) \cdot \mathsf{u}_{i,r}(\tau) \cdot (-\eta \sum_{j=1}^{n} \sum_{k_2=1}^{d} (\mathsf{F}_{k_2,j}(\tau) - y_{k_2,j}) \cdot a_r \mathsf{S}_{j,r}(\tau) e_{k_2}^\top) x_i$$

- *Let $C_{1,2} := 2\langle \operatorname{vec}(\mathsf{F}(\tau) - Y), \operatorname{vec}(v_{1,2})\rangle$*

*Then, with a probability at least $1 - \delta/\operatorname{poly}(nd)$, we have*

$$|C_{1,2}| \leq 0.1\eta m\lambda \|\mathsf{F}(\tau) - Y\|_F^2$$

*Proof.* We have the quantity of $v_{1,2,k,i}$

$$|\sum_{i=1}^{n} \sum_{k=1}^{d} v_{1,2,k,i}(\mathsf{F}_{k,i}(\tau) - y_{k,i})|$$

$$\leq |\sum_{i=1}^{n}\sum_{k=1}^{d} m^2 \sum_{r=1}^{m} \theta_{k,i,r}(\tau) \cdot \mathsf{u}_{i,r}(\tau)$$

$$\cdot (-\eta \sum_{j=1}^{n}\sum_{k_2=1}^{d} (\mathsf{F}_{k_2,j}(\tau) - y_{k_2,j}) \cdot a_r \mathsf{S}_{j,r}(\tau) e_{k_2}^{\top}) x_i \cdot (\mathsf{F}_{k,i}(\tau) - y_{k,i})|$$

$$\leq |\sum_{i=1}^{n}\sum_{k=1}^{d} m^2 \sum_{r=1}^{m} \beta_{k,r}(\tau)\alpha_i(\tau)^{-1} \cdot \mathsf{u}_{i,r}(\tau)$$

$$\cdot (-\eta \sum_{j=1}^{n}\sum_{k_2=1}^{d} (\mathsf{F}_{k_2,j}(\tau) - y_{k_2,j}) \cdot a_r \mathsf{S}_{j,r}(\tau) e_{k_2}^{\top}) x_i \cdot (\mathsf{F}_{k,i}(\tau) - y_{k,i})|$$

$$\leq |\sum_{i=1}^{n}\sum_{k=1}^{d} m^2 \sum_{r=1}^{m} \beta_{k,r}(\tau)\mathsf{S}_{i,r}(\tau)$$

$$\cdot (-\eta \sum_{j=1}^{n}\sum_{k_2=1}^{d} (\mathsf{F}_{k_2,j}(\tau) - y_{k_2,j}) \cdot a_r \mathsf{S}_{j,r}(\tau) e_{k_2}^{\top}) x_i \cdot (\mathsf{F}_{k,i}(\tau) - y_{k,i})|$$

$$\leq \eta m^2 |\sum_{i=1}^{n}\sum_{k=1}^{d}\sum_{r=1}^{m} \beta_{k,r}(\tau)\mathsf{S}_{i,r}(\tau)$$

$$\cdot (-\sum_{j=1}^{n}\sum_{k_2=1}^{d} (\mathsf{F}_{k_2,j}(\tau) - y_{k_2,j}) \cdot a_r \mathsf{S}_{j,r}(\tau) e_{k_2}^{\top}) x_i \cdot (\mathsf{F}_{k,i}(\tau) - y_{k,i})|$$

$$\leq \eta \exp(6B) \sum_{r=1}^{m} |a_r \cdot \max_{k\in[d]} \beta_{k,r}(\tau)| \cdot \|(\mathsf{F}(\tau) - Y) \otimes (\mathsf{F}(\tau) - Y)\|_1$$

$$\leq \eta \exp(6B) \sum_{r=1}^{m} |a_r \cdot \max_{k\in[d]} \beta_{k,r}(\tau)| \cdot \|\mathsf{F}(\tau) - Y\|_1^2$$

$$\leq \eta nd \exp(6B) \sum_{r=1}^{m} |a_r \cdot \max_{k\in[d]} \beta_{k,r}(\tau)| \cdot \|\mathsf{F}(\tau) - Y\|_F^2$$

where the first step follows from the definition of $v_{1,2,k,i}$, the second step follows from Definition G.6, the third step follows from Definition F.5, the fourth step follows from Definition G.7, the fifth step follows from simple algebras, the sixth step follows from $0 \leq \mathsf{S}_{j,r} \leq \frac{\exp(3B)}{m}$, $\|x_i\|_2 \leq 1$ and the definition of Kronecker product, the seventh step follows from the definition of $\ell_1$ norm, the last step follows from $\|U\|_1 \leq \sqrt{nd}\|U\|_F$ for $U \in \mathbb{R}^{n\times d}$.

Then by Part 1 of Lemma L.1, we have

$$|\max_{k\in[d]} \beta_{k,r}(\tau)| \leq B$$

We apply Hoeffding inequality (Lemma E.4) to random variables $a_r \cdot \max_{k\in[d]} \beta_{k,r}(\tau)$ for $r \in [m]$. By $\mathbb{E}[\sum_{r=1}^{m} a_r \cdot \max_{k\in[d]} \beta_{k,r}(\tau)] = 0$, we have

$$|\sum_{i=1}^{n}\sum_{k=1}^{d} v_{1,2,k,i}(\mathsf{F}_{k,i}(\tau) - y_{k,i})| \leq \eta nd \exp(6B)B\|\mathsf{F}(\tau) - Y\|_F^2$$

with a probability at least $1 - \delta/\operatorname{poly}(nd)$.

By the Lemma condition, we have

$$nd \exp(6B)B \leq 0.1m\lambda$$

$\square$

### I.3 BOUNDING $C_2$

**Lemma I.6.** *If the following conditions hold*

- *Let $\lambda = \lambda_{\min}(H^*)$*

- *Let $C > 10$ denote a sufficiently large constant*

- *Let $B := \max\{C\sigma\sqrt{\log(nd/\delta)}, 1\}$.*

- *Let $\delta \in (0, 0.1)$.*

- *Let $m \geq \Omega(\lambda^{-2}n^2d^2\exp(30B)\sqrt{\log(nd/\delta)})$.*

- *Let $r \in [m]$, let $i, j \in [n]$, let $k, k_1 \in [d]$.*

- *Let $\beta_k(\tau) \in \mathbb{R}^m$ be defined as Definition F.5.*

- *Let $\alpha_i(\tau) \in \mathbb{R}$ be defined as Definition F.3.*

- *Let $\theta_{k,i}(\tau) \in \mathbb{R}^m$ be defined as Definition G.6.*

- *Let $\mathsf{u}_i(\tau) \in \mathbb{R}^m$ be defined as Definition G.2.*

- *Let $\mathsf{S}_i(\tau) \in \mathbb{R}^m$ be defined as Definition G.7.*

- *Let $v_k := \beta_{k,r}(\tau) \cdot \mathbf{1}_m - \beta_k(\tau) \in \mathbb{R}^m$*

- *Denote $\mathsf{F}(\tau) \in \mathbb{R}^{n \times d}$ as Definition G.8.*

- *Let $Y \in \mathbb{R}^{n \times d}$ denote the labels.*

- *Let $\eta > 0$ denote the learning rate.*

- *Let scalar $v_{2,k,i} \in \mathbb{R}$ be defined as follows (we omit $(\tau)$ in the following terms)*

$$v_{2,k,i} := m\sum_{r=1}^{m}\theta_{k,i,r}(\tau) \cdot \mathsf{u}_{i,r}(\tau) \cdot \eta^2 \cdot \Theta(1) \cdot \langle \Delta w_r(\tau), x_i\rangle^2$$

- *Let $C_2 := 2\langle \operatorname{vec}(\mathsf{F}(\tau) - Y), \operatorname{vec}(v_2)\rangle$*

*Then, with a probability at least $1 - \delta/\operatorname{poly}(nd)$, we have*

$$|C_2| \leq \eta^2 m \cdot n^2 d^2 \exp(16B)\|\mathsf{F}(\tau) - Y\|_F^2$$

*Proof.* We have

$$\langle \Delta w_r(\tau), x_i\rangle^2$$
$$\leq \left(m\sum_{j=1}^{n}\sum_{k=1}^{d}(\mathsf{F}_{k,i}(\tau) - y_{k,i}) \cdot \left(\langle v_{k,r}(\tau), \mathsf{S}_j(\tau)\rangle \cdot \mathsf{S}_{j,r}(\tau) \cdot x_j^\top + a_r\mathsf{S}_{j,r}(\tau)e_k^\top\right)x_i\right)^2$$
$$\leq \exp(12B) \cdot \|\mathsf{F}(\tau) - Y\|_1^2$$
$$\leq nd\exp(12B) \cdot \|\mathsf{F}(\tau) - Y\|_F^2 \tag{17}$$

where the first step follows from Claim G.12, the second step follows from the definition of $\ell_1$ norm, $0 \leq \mathsf{S}_{j,r} \leq \frac{\exp(3B)}{m}$ by Part 11 of Lemma L.1 and Part 6 of Lemma L.2, last step follows from $\|U\|_1 \leq \sqrt{nd}\|U\|_F$ for $U \in \mathbb{R}^{n \times d}$.

Then, we can show that

$$|\sum_{i=1}^{n}\sum_{k=1}^{d}v_{2,k,i}(\mathsf{F}_{k,i}(\tau) - y_{k,i})|$$

$$\leq |\sum_{i=1}^{n}\sum_{k=1}^{d} m \sum_{r=1}^{m} \theta_{k,i,r}(\tau) \cdot \mathsf{u}_{i,r}(\tau) \cdot \eta^2 \cdot \Theta(1) \cdot \langle \Delta w_r(\tau), x_i \rangle^2 \cdot (\mathsf{F}_{k,i}(\tau) - y_{k,i})|$$

$$\leq \eta^2 |\sum_{i=1}^{n}\sum_{k=1}^{d} m \sum_{r=1}^{m} \theta_{k,i,r}(\tau) \cdot \mathsf{u}_{i,r}(\tau) \cdot \langle \Delta w_r(\tau), x_i \rangle^2 \cdot (\mathsf{F}_{k,i}(\tau) - y_{k,i})|$$

$$\leq \eta^2 |\sum_{i=1}^{n}\sum_{k=1}^{d} m \sum_{r=1}^{m} \beta_{k,r}(\tau) \cdot \alpha_i(\tau)^{-1} \cdot \mathsf{u}_{i,r}(\tau) \cdot \langle \Delta w_r(\tau), x_i \rangle^2 \cdot (\mathsf{F}_{k,i}(\tau) - y_{k,i})|$$

$$\leq \eta^2 |\sum_{i=1}^{n}\sum_{k=1}^{d} m \sum_{r=1}^{m} \beta_{k,r}(\tau) \cdot \mathsf{S}_{i,r}(\tau) \cdot \langle \Delta w_r(\tau), x_i \rangle^2 \cdot (\mathsf{F}_{k,i}(\tau) - y_{k,i})|$$

$$\leq \eta^2 \exp(3B) |\sum_{i=1}^{n}\sum_{k=1}^{d}\sum_{r=1}^{m} \beta_{k,r}(\tau) \cdot \langle \Delta w_r(\tau), x_i \rangle^2 \cdot (\mathsf{F}_{k,i}(\tau) - y_{k,i})|$$

$$\leq \eta^2 \exp(4B) |\sum_{i=1}^{n}\sum_{k=1}^{d}\sum_{r=1}^{m} a_r \langle \Delta w_r(\tau), x_i \rangle^2 \cdot (\mathsf{F}_{k,i}(\tau) - y_{k,i})|$$

$$\leq \eta^2 \exp(4B) |\sum_{r=1}^{m} a_r \max_{i \in [n]} \langle \Delta w_r(\tau), x_i \rangle^2 | \cdot \sqrt{nd} \|\mathsf{F}(\tau) - Y\|_F$$

$$\leq \eta^2 \sqrt{m} n d \exp(4B) |\sum_{r=1}^{m} a_r \max_{i \in [n]} \langle \Delta w_r(\tau), x_i \rangle^2 |$$

where the first step follows from the definition of $v_{2,k,i}$, the second step follows from simple algebras, the third step follows from Definition G.6, the fourth step follows from Definition G.7, the fifth step follows from $0 \leq \mathsf{S}_{i,r} \leq \frac{\exp(3B)}{m}$ by Part 11 of Lemma L.1, the sixth step follows from Part 1 of Lemma L.1 and Definition F.5, the seventh step follows from definition of $\ell_1$ norm and $\|U\|_1 \leq \sqrt{nd}\|U\|_F$ for $U \in \mathbb{R}^{n \times d}$, the last step follows from Lemma I.8.

Next, by Eq.(17), applying Hoeffding inequality (Lemma E.4) to $a_r \max_{i \in [n]} \langle \Delta w_r(\tau), x_i \rangle^2$ for $r \in [m]$ and $\mathbb{E}[\sum_{r=1}^{m} a_r \max_{i \in [n]} \langle \Delta w_r(\tau), x_i \rangle^2] = 0$, we have

$$|\sum_{i=1}^{n}\sum_{k=1}^{d} v_{2,k,i}(\mathsf{F}_{k,i}(\tau) - y_{k,i})| \leq \eta^2 \sqrt{m} n^2 d^2 \exp(16B) \cdot \|\mathsf{F}(\tau) - Y\|_F^2 \cdot \sqrt{m \log(nd/\delta)}$$

with a probability at least $1 - \delta / \operatorname{poly}(nd)$.

By the Lemma condition, we have

$$\eta^2 \sqrt{m} n^2 d^2 \exp(16B) \cdot \sqrt{m \log(nd/\delta)} \leq \eta^2 m \cdot n^2 d^2 \exp(16B)$$

Then we complete the proof. $\qquad\square$

## I.4 Bounding $C_3$

**Lemma I.7.** *If the following conditions hold*

- *Let $\lambda = \lambda_{\min}(H^*)$*

- *Let $C > 10$ denote a sufficiently large constant*

- *Let $B := \max\{C\sigma\sqrt{\log(nd/\delta)}, 1\}$.*

- *Let $\delta \in (0, 0.1)$.*

- *Let $m \geq \Omega(\lambda^{-2}n^2 d^2 \exp(30B)\sqrt{\log(nd/\delta)})$.*

- *Let $r \in [m]$, let $i, j \in [n]$, let $k, k_1 \in [d]$.*

- *Let $\beta_k(\tau) \in \mathbb{R}^m$ be defined as Definition F.5.*

- *Let $\alpha_i(\tau) \in \mathbb{R}$ be defined as Definition F.3.*

- *Let $\theta_{k,i}(\tau) \in \mathbb{R}^m$ be defined as Definition G.6.*

- *Let $\mathsf{u}_i(\tau) \in \mathbb{R}^m$ be defined as Definition G.2.*

- *Let $\mathsf{S}_i(\tau) \in \mathbb{R}^m$ be defined as Definition G.7.*

- *Let $v_k := \beta_{k,r}(\tau) \cdot \mathbf{1}_m - \beta_k(\tau) \in \mathbb{R}^m$*

- *Denote $\mathsf{F}(\tau) \in \mathbb{R}^{n \times d}$ as Definition G.8.*

- *Let $Y \in \mathbb{R}^{n \times d}$ denote the labels.*

- *Let $\eta > 0$ denote the learning rate.*

- *Let $C_3 := \|\mathsf{F}(\tau + 1) - \mathsf{F}(\tau)\|_F^2$*

*Then, with a probability at least $1 - \delta/\operatorname{poly}(nd)$, we have*

$$|C_3| \le \eta^2 m^2 \|\mathsf{F}(\tau) - Y\|_F^2$$

*Proof.* We have

$$
\begin{aligned}
|C_3| &= \|\mathsf{F}(\tau + 1) - \mathsf{F}(\tau)\|_F^2 \\
&= \sum_{i=1}^n \sum_{k=1}^d (\mathsf{F}_{k,i}(\tau + 1) - \mathsf{F}_{k,i}(\tau))^2 \\
&= \sum_{i=1}^n \sum_{k=1}^d m^2 (\langle \beta_k(\tau + 1), \mathsf{S}_i(\tau + 1) \rangle - \langle \beta_k(\tau), \mathsf{S}_i(\tau) \rangle)^2 \\
&= \sum_{i=1}^n \sum_{k=1}^d m^2 \Big( \sum_{r=1}^m (\beta_{k,r}(\tau + 1) \cdot \mathsf{S}_{i,r}(\tau + 1) - \beta_{k,r}(\tau) \cdot \mathsf{S}_{i,r}(\tau)) \Big)^2 \\
&= \sum_{i=1}^n \sum_{k=1}^d m^2 \Big( \sum_{r=1}^m (\beta_{k,r}(\tau + 1) \cdot \mathsf{S}_{i,r}(\tau + 1) - \beta_{k,r}(\tau + 1) \cdot \mathsf{S}_{i,r}(\tau) \\
&\qquad + \beta_{k,r}(\tau + 1) \cdot \mathsf{S}_{i,r}(\tau) - \beta_{k,r}(\tau) \cdot \mathsf{S}_{i,r}(\tau)) \Big)^2 \\
&= \sum_{i=1}^n \sum_{k=1}^d m^2 \Big( \sum_{r=1}^m (\beta_{k,r}(\tau + 1) \cdot (\mathsf{S}_{i,r}(\tau + 1) - \mathsf{S}_{i,r}(\tau)) \\
&\qquad + (\beta_{k,r}(\tau + 1) - \beta_{k,r}(\tau)) \cdot \mathsf{S}_{i,r}(\tau)) \Big)^2 \\
&= \sum_{i=1}^n \sum_{k=1}^d m^2 (Q_{3,1,i,k} + Q_{3,2,i,k})^2
\end{aligned}
$$

where the first step follows from the definition $C_2$, the second step follows from the definition of Frobenius norm, the third step follows from Definition G.8, the fourth, fifth and sixth steps follow from simple algebras, the last step follows from defining

$$Q_{3,1,i,k} = \sum_{r=1}^m \beta_{k,r}(\tau + 1) \cdot (\mathsf{S}_{i,r}(\tau + 1) - \mathsf{S}_{i,r}(\tau)),$$

$$Q_{3,2,i,k} = \sum_{r=1}^m (\beta_{k,r}(\tau + 1) - \beta_{k,r}(\tau)) \cdot \mathsf{S}_{i,r}(\tau).$$

**Bounding first term.** For the first term, we have

$$
|Q_{3,1,i,k}| = |\sum_{r=1}^{m} \beta_{k,r}(\tau+1) \cdot (\mathsf{S}_{i,r}(\tau+1) - \mathsf{S}_{i,r}(\tau))|
$$

$$
= |\sum_{r=1}^{m} a_r \cdot w_{r,k}(\tau+1) \cdot (\mathsf{S}_{i,r}(\tau+1) - \mathsf{S}_{i,r}(\tau))|
$$

$$
\leq |B \cdot \sum_{r=1}^{m} a_r \cdot (\mathsf{S}_{i,r}(\tau+1) - \mathsf{S}_{i,r}(\tau))|
$$

$$
\leq |\exp(3B) \cdot \sum_{r=1}^{m} a_r \cdot \max_{i \in [n]} (\alpha_i(\tau+1)^{-1} - \alpha_i(\tau)^{-1})|
$$

where the first step follows from the definition of $Q_{3,1,i,k}$, the second step follows from Definition F.5, the third step follows from Part 1 of Lemma L.1, last step follows from Part 4 of Lemma L.1, Definition G.7 and $B \leq \exp(B)$.

Then by Part 2 of Lemma I.9, applying Hoeffding inequality (Lemma E.4) to the random variables $a_r \cdot \max_{i \in [n]} (\alpha_i(\tau+1)^{-1} - \alpha_i(\tau)^{-1}$ for $r \in [m]$ and $\mathbb{E}[\sum_{r=1}^{m} a_r \cdot \max_{i \in [n]} (\alpha_i(\tau+1)^{-1} - \alpha_i(\tau)^{-1}] = 0$, we have

$$
|Q_{3,1,i,k}| \leq (\eta \frac{\sqrt{nd}\exp(18B)}{m^3} \cdot \|\mathsf{F}(\tau) - Y\|_F + \eta^2 \frac{nd\exp(30B)}{\sqrt{m}} \cdot \|\mathsf{F}(\tau) - Y\|_F) \cdot \sqrt{m\log(nd/\delta)}
$$

with a probability of at least $1 - \delta/\operatorname{poly}(nd)$.

By the Lemma condition, we have

$$
(\eta \frac{\sqrt{nd}\exp(18B)}{m^3} + \eta^2 \frac{nd\exp(30B)}{\sqrt{m}}) \cdot \sqrt{m\log(nd/\delta)} \leq \frac{1}{2\sqrt{nd}}\eta
$$

**Bounding second term.** On the other hand, for the second term $Q_{3,2,k,i}$, we have

$$
|Q_{3,2,k,i}| = |\sum_{r=1}^{m} (\beta_{k,r}(\tau+1) - \beta_{k,r}(\tau)) \cdot \mathsf{S}_{i,r}(\tau)|
$$

$$
= \eta |\sum_{r=1}^{m} a_r \Delta w_{r,k}(\tau) \cdot \mathsf{S}_{i,r}(\tau)|
$$

$$
\leq \eta \frac{\exp(3B)}{m} |\sum_{r=1}^{m} a_r \Delta w_{r,k}(\tau)|
$$

$$
\leq \eta \exp(3B) \Big| \sum_{r=1}^{m} a_r \sum_{j=1}^{n} \sum_{k_1=1}^{d} (\mathsf{F}_{k_1,j}(\tau) - y_{k_1,j})
$$

$$
\cdot \Big( \langle v_{k_1,r}(\tau), \mathsf{S}_j(\tau) \rangle \cdot \mathsf{S}_{j,r}(\tau) \cdot x_{i,k} + a_r \mathsf{S}_{j,r}(\tau) e_{k,k_1} \Big) \Big|
$$

$$
\leq \eta \frac{\exp(6B)}{m} |\sum_{r=1}^{m} a_r \max_{j \in [n], k, k_1 \in [d]} C_{j,k,k_1,r}| \cdot \|\mathsf{F}(\tau) - Y\|_1
$$

$$
\leq \eta \frac{\sqrt{nd}\exp(6B)}{m} |\sum_{r=1}^{m} a_r \max_{j \in [n], k, k_1 \in [d]} C_{j,k,k_1,r}| \cdot \|\mathsf{F}(\tau) - Y\|_F
$$

where the first step follows from the definition of $Q_{3,2,k,i}$, the second step follows from Definition G.13, the third step follows from $0 \leq \mathsf{S}_{i,r} \leq \frac{\exp(3B)}{m}$ by Part 11 of Lemma L.1, the fourth step follows from Claim G.12, the fifth step follows from $0 \leq \mathsf{S}_{i,r} \leq \frac{\exp(3B)}{m}$ by Part 11 of Lemma L.1, $\|x_i\|_2 \leq 1$ and defining

$$
C_{j,k,k_1,r} := \langle v_{k_1,r}(\tau), \mathsf{S}_j(\tau) \rangle + e_{k,k_1},
$$

the last step follows from $\|U\|_1 \leq \sqrt{nd}\|U\|_F$ for $U \in \mathbb{R}^{n \times d}$.

Now we follow from Part 6 of Lemma L.2, applying Hoeffding inequality (Lemma E.4) to random variables $a_r \max_{j \in [n], k, k_1 \in [d]} C_{j,k,k_1,r}$ for $r \in [m]$ and $\mathbb{E}[\sum_{r=1}^m a_r \max_{j \in [n], k, k_1 \in [d]} C_{j,k,k_1,r}] = 0$, we have

$$|Q_{3,2,k,i}| \leq \eta \frac{\sqrt{nd} \exp(13B)}{m} \cdot \|\mathsf{F}(\tau) - Y\|_F \cdot \sqrt{m \log(nd/\delta)} \leq \frac{1}{2\sqrt{nd}} \eta$$

Finally, we combine all terms, we have

$$|C_3| = \sum_{i=1}^n \sum_{k=1}^d m^2 ((\frac{1}{2\sqrt{nd}}\eta + \frac{1}{2\sqrt{nd}}\eta) \cdot \|\mathsf{F}(\tau) - Y\|_F)^2$$
$$\leq \eta^2 m^2 \|\mathsf{F}(\tau) - Y\|_F^2$$

$\square$

## I.5 Bounding Loss during Training Process

**Lemma I.8.** *If the following conditions hold*

- *Denote $\mathsf{F}(\tau) \in \mathbb{R}^{n \times d}$ as Definition G.8.*

- *Let $Y \in \mathbb{R}^{n \times d}$ denote the labels.*

*Then we have*

$$\|\mathsf{F}(\tau) - Y\|_F \leq O(\sqrt{nmd})$$

*Proof.* This proof follows from $\|y_i\| \leq 1$ for $i \in [n]$ and Definition G.8. $\square$

## I.6 Helpful Lemma

**Lemma I.9.** *If the following conditions hold*

- *Let $\lambda = \lambda_{\min}(H^*)$.*

- *Let $C > 10$ denote a sufficiently large constant.*

- *Let $B := \max\{C\sigma\sqrt{\log(nd/\delta)}, 1\}$.*

- *Let $\delta \in (0, 0.1)$.*

- *Let $m \geq \Omega(\lambda^{-2}n^2d^2 \exp(30B)\sqrt{\log(nd/\delta)})$.*

- *Let $r \in [m]$, let $i, j \in [n]$, let $k, k_1 \in [d]$.*

- *Let $\alpha_i(\tau) \in \mathbb{R}$ be defined as Definition F.3.*

- *Let $\beta_k(\tau) \in \mathbb{R}^m$ be defined as Definition F.5.*

- *Let $\theta_{k,i}(\tau) \in \mathbb{R}^m$ be defined as Definition G.6.*

- *Let $\mathsf{u}_i(\tau) \in \mathbb{R}^m$ be defined as Definition G.2.*

- *Let $\mathsf{S}_i(\tau) \in \mathbb{R}^m$ be defined as Definition G.7.*

- *Let $v_k := \beta_{k,r}(\tau) \cdot \mathbf{1}_m - \beta_k(\tau) \in \mathbb{R}^m$.*

- *Denote $\mathsf{F}(\tau) \in \mathbb{R}^{n \times d}$ as Definition G.8.*

- *Let $Y \in \mathbb{R}^{n \times d}$ denote the labels.*

*Then with a probability at least $1 - \delta/\operatorname{poly}(nd)$, we have*

- *Part 1.*

$$\alpha_i(\tau + 1) - \alpha_i(\tau) \leq \eta \frac{\sqrt{nd}\exp(9B)}{m} \cdot \|\mathsf{F}(\tau) - Y\|_F + \eta^2 m^{1.5} \cdot nd\exp(21B) \cdot \|\mathsf{F}(\tau) - Y\|_F$$

- *Part 2.*

$$\alpha_i(\tau + 1)^{-1} - \alpha_i(\tau)^{-1} \leq \eta \frac{\sqrt{nd}\exp(15B)}{m^3} \cdot \|\mathsf{F}(\tau) - Y\|_F + \eta^2 \frac{nd\exp(27B)}{\sqrt{m}} \cdot \|\mathsf{F}(\tau) - Y\|_F$$

*Proof.* **Proof of Part 1.**

We have

$$\begin{aligned}
&\alpha_i(\tau + 1) - \alpha_i(\tau) \\
&= \langle \mathsf{u}_i(\tau + 1), \mathbf{1}_m \rangle - \langle \mathsf{u}_i(\tau), \mathbf{1}_m \rangle \\
&= \langle \mathsf{u}_i(\tau + 1) - \mathsf{u}_i(\tau), \mathbf{1}_m \rangle \\
&= \langle \exp(W(\tau + 1)^\top x_i) - \exp(W(\tau)^\top x_i), \mathbf{1}_m \rangle \\
&= \langle \exp(W(\tau)^\top x_i) \circ (\exp(-\eta \Delta W(\tau)^\top x_i) - \mathbf{1}_m), \mathbf{1}_m \rangle \\
&= \langle \exp(W(\tau)^\top x_i) \circ (-\eta \Delta W(\tau)^\top x_i + \Theta(1)\eta^2 \cdot (\Delta W(\tau)^\top x_i)^2), \mathbf{1}_m \rangle \\
&= \langle -\eta \Delta W(\tau)^\top x_i + \Theta(1)\eta^2 \cdot (\Delta W(\tau)^\top x_i)^2, \exp(W(\tau)^\top x_i) \rangle \\
&\leq \exp(B) \cdot \langle -\eta \Delta W(\tau)^\top x_i + \Theta(1)\eta^2 \cdot (\Delta W(\tau)^\top x_i)^2, \mathbf{1}_m \rangle \\
&\leq \eta \frac{\sqrt{nd}\exp(9B)}{m} \cdot \|\mathsf{F}(\tau) - Y\|_F + \eta^2 m^{1.5} \cdot nd\exp(21B) \cdot \|\mathsf{F}(\tau) - Y\|_F
\end{aligned}$$

where the first step follows from Definition F.3, the second step follows from simple algebras, the third step follows from Definition G.2, the fourth step follows from simple algebra, the fifth step follows from Fact E.1, the sixth step follows from simple algebras, the seventh step follows from Part 4 of Lemma L.1, last step follows from Part 1 and Part 2 of Lemma I.10.

**Proof of Part 2.** We have

$$\begin{aligned}
\alpha_i(\tau + 1)^{-1} - \alpha_i(\tau)^{-1} &= \alpha_i(\tau + 1)^{-1}\alpha_i(\tau)^{-1} \cdot (\alpha_i(\tau + 1) - \alpha_i(\tau)) \\
&\leq \frac{\exp(6B)}{m^2} \cdot (\alpha_i(\tau + 1) - \alpha_i(\tau)) \\
&\leq \eta \frac{\sqrt{nd}\exp(15B)}{m^3} \cdot \|\mathsf{F}(\tau) - Y\|_F + \eta^2 \frac{nd\exp(27B)}{\sqrt{m}} \cdot \|\mathsf{F}(\tau) - Y\|_F
\end{aligned}$$

where the first step follows from simple algebras, the second step follows from Part 4 of Lemma L.2, the last step follows from Part 1 of this Lemma. $\square$

**Lemma I.10.** *If the following conditions hold*

- *Let $\lambda = \lambda_{\min}(H^*)$.*

- *Let $W(\tau) \in \mathbb{R}^{m \times d}$ be defined as Definition G.13, let $a \in \mathbb{R}^m$ be defined as Definition F.1.*

- *Let $C > 10$ denote a sufficiently large constant.*

- *Let $B := \max\{C\sigma\sqrt{\log(nd/\delta)}, 1\}$.*

- *Let $\delta \in (0, 0.1)$.*

- *Let $m \geq \Omega(\lambda^{-2}n^2d^2\exp(30B)\sqrt{\log(nd/\delta)})$.*

- *Let $r \in [m]$, let $i, j \in [n]$, let $k, k_2 \in [d]$.*

- *Let $\mathsf{S}_i(\tau) \in \mathbb{R}^m$ be defined as Definition G.7.*

- *Let $v_{k,r} := \beta_{k,r}(\tau) \cdot \mathbf{1}_m - \beta_k(\tau) \in \mathbb{R}^m$.*

- *Denote $\mathsf{F}(\tau) \in \mathbb{R}^{n \times d}$ as Definition G.8.*

- *Let $Y \in \mathbb{R}^{n \times d}$ denote the labels.*

- *Let $\eta = \lambda/(m \cdot \mathrm{poly}(n, d, \exp(B)))$ denote the learning rate.*

*Then with a probability at least $1 - \delta/\mathrm{poly}(nd)$, we have*

- *Part 1.*

$$|\langle \eta \Delta W(\tau)^\top x_i, \mathbf{1}_m \rangle| \le \eta \frac{\sqrt{nd} \exp(8B)}{m} \cdot \|\mathsf{F}(\tau) - Y\|_F$$

- *Part 2.*

$$|\langle \eta^2 (\Delta W(\tau)^\top x_i)^2, \mathbf{1}_m \rangle| \le \eta^2 m^{1.5} \cdot nd \exp(20B) \cdot \|\mathsf{F}(\tau) - Y\|_F$$

*Proof.* **Proof of Part 1.** We have

$$|\langle \eta \Delta W(\tau)^\top x_i, \mathbf{1}_m \rangle|$$

$$= \eta |\sum_{r=1}^m \langle \Delta w_r(\tau), x_i \rangle|$$

$$\le \eta \Big| \sum_{r=1}^m m \sum_{j=1}^n \sum_{k=1}^d (\mathsf{F}_{k,i}(\tau) - y_{k,i}) \cdot \Big( \langle v_{k,r}(\tau), \mathsf{S}_j(\tau) \rangle \cdot \mathsf{S}_{j,r}(\tau) \cdot x_j^\top + a_r \mathsf{S}_{j,r}(\tau) e_k^\top \Big) x_i \Big|$$

$$\le \eta \Big| \sum_{r=1}^m m \sum_{j=1}^n \sum_{k=1}^d (\mathsf{F}_{k,i}(\tau) - y_{k,i}) \cdot \Big( \langle \beta_{k,r}(\tau) \cdot \mathbf{1}_m - \beta_k(\tau), \mathsf{S}_j(\tau) \rangle \cdot \mathsf{S}_{j,r}(\tau) \cdot x_j^\top + a_r \mathsf{S}_{j,r}(\tau) e_k^\top \Big) x_i \Big|$$

$$\le \eta \Big| \sum_{r=1}^m m \sum_{j=1}^n \sum_{k=1}^d (\mathsf{F}_{k,i}(\tau) - y_{k,i}) \cdot \Big( a_r w_{r,k} + \langle -a \circ W_{k,*}(\tau), \mathsf{S}_j(\tau) \rangle \cdot \mathsf{S}_{j,r}(\tau) \cdot x_j^\top + a_r \mathsf{S}_{j,r}(\tau) e_k^\top \Big) x_i \Big|$$

$$\le \eta \frac{\exp(3B)}{m} \sum_{r=1}^m \sigma_r \max_{j \in [n], k \in [d]} C_{j,k,r} \|\mathsf{F}(\tau) - Y\|_1$$

$$\le \eta \frac{\sqrt{nd} \exp(3B)}{m} \sum_{r=1}^m \sigma_r \max_{j \in [n], k \in [d]} C_{j,k,r} \|\mathsf{F}(\tau) - Y\|_F$$

where the first step follows from simple algebras, the second step follows from Claim G.12, the third step follows from the definition of $v_{k,r}$, the fourth step follows from Definition F.5 and simple algebras, the fifth step follows from $\|x_i\|_2 \le 1$, $1 \le \mathsf{S}_{i,r} \le \frac{\exp(3B)}{m}$ by Part 11 of Lemma L.1, definition of $\ell_1$ norm and defining

$$C_{j,k,r} := |w_{r,k}| + |\langle -W_{k,*}(\tau), \mathsf{S}_j(\tau) \rangle| + \|e_k\|, \sigma_r \in \{+1, -1\},$$

the last step follows from $\|U\|_1 \le \sqrt{nd}\|U\|_F$ for $U \in \mathbb{R}^{n \times d}$.

Thus, by following Part 1 and Part 11 of Lemma L.2 and Hoeffding inequality (Lemma E.4), we have

$$|\langle \eta \Delta W(\tau)^\top x_i, \mathbf{1}_m \rangle| \le \eta \frac{\sqrt{nd} \exp(8B)}{m} \cdot \|\mathsf{F}(\tau) - Y\|_F$$

with a probability at least $1 - \delta/\mathrm{poly}(nd)$.

**Proof of Part 2.** We have

$$|\langle \eta^2 (\Delta W(\tau)^\top x_i)^2, \mathbf{1}_m \rangle|$$

$$\le \eta^2 \sum_{r=1}^m (\langle \Delta w_r(\tau), x_i \rangle)^2$$

$$\leq \eta^2 \sum_{r=1}^{m} \Big( m \sum_{j=1}^{n} \sum_{k=1}^{d} (\mathsf{F}_{k,i}(\tau) - y_{k,i}) \cdot \Big( \langle v_{k,r}(\tau), \mathsf{S}_j(\tau) \rangle \cdot \mathsf{S}_{j,r}(\tau) \cdot x_j^\top + a_r \mathsf{S}_{j,r}(\tau) e_k^\top \Big) x_i \Big)^2$$

$$\leq \eta^2 \exp(6B) \sum_{r=1}^{m} \Big( \sum_{j=1}^{n} \sum_{k=1}^{d} (\mathsf{F}_{k,i}(\tau) - y_{k,i}) \cdot \Big( \langle v_{k,r}(\tau), \mathsf{S}_j(\tau) \rangle \cdot x_j^\top + a_r e_k^\top \Big) x_i \Big)^2$$

$$\leq \eta^2 m \exp(20B) \cdot \|\mathsf{F}(\tau) - Y\|_1^2$$

$$\leq \eta^2 m \sqrt{nmd} \exp(20B) \cdot \|\mathsf{F}(\tau) - Y\|_1$$

$$\leq \eta^2 m^{1.5} \cdot nd \exp(20B) \cdot \|\mathsf{F}(\tau) - Y\|_F$$

where the first step follows from simple algebras, the second step follows from Claim G.12, the third step follows from $0 \leq \mathsf{S}_{i,r} \leq \frac{\exp(3B)}{m}$ by Part 11 of Lemma L.1, the fourth step follows from $\langle v_{k,r}(\tau), \mathsf{S}_j(\tau) \rangle \leq \exp(6B)$ by Part 6 of Lemma L.2, $\|x_i\|_2 \leq 1$, $\exp(6B) + 1 \leq \exp(7B)$ and the definition of $\ell_1$ norm, the fifth step follows from Lemma I.8, the last step follows from $\|U\|_1 \leq \|U\|_F$ for $U \in \mathbb{R}^{n \times d}$. $\qquad \square$

## J CONVERGENCE OF PREFIX LEARNING

Here, we provide all the properties we need for math induction for NTK happening.

**Definition J.1** (Properties). *We state the following properties*

- *General Condition 1. Let $\lambda = \lambda_{\min}(H^*) > 0$*

- *General Condition 2. Let $B := \max\{C\sigma\sqrt{\log(nd/\delta)}, 1\}$.*

- *General Condition 3. Let $\eta$ be defined as*

$$\eta := \lambda/(m \operatorname{poly}(n, d, \exp(B))).$$

- *General Condition 4. Let $D := 2\lambda^{-1} \cdot \exp(20B) \frac{\sqrt{nd}}{m} \|Y - \mathsf{F}(0)\|_F$*

- *General Condition 5. Let $w_r$ and $a_r$ be defined as Definition F.1.*

- *General Condition 6. $D < R = \lambda/\operatorname{poly}(n, d, \exp(B))$*

- *General Condition 7. $m = \lambda^{-2} \operatorname{poly}(n, d, \exp(B))$*

- **Weight Condition.** $\|w_r(t) - w_r(0)\|_2 \leq D < R$, $\forall r \in [m]$

- **Loss Condition.** $\|\operatorname{vec}(\mathsf{F}(i) - Y)\|_2^2 \leq \|\operatorname{vec}(\mathsf{F}(0) - Y)\|_2^2 \cdot (1 - m\eta\lambda/2)^i$, $\forall i \in [t]$

- **Gradient Condition.** $\eta\|\Delta w_r(i)\|_2 \leq 0.01$ $\forall r \in [m]$, $\forall i \in [t]$

### J.1 MAIN RESULT

Our main result is presented as follows.

**Theorem J.2** (Main result, formal version of Theorem 3.2). *For any $\epsilon, \delta \in (0, 0.1)$, if the following conditions hold*

- *Let $\lambda = \lambda_{\min}(H^*) > 0$*

- *Let $B = \max\{C\sigma\sqrt{\log(nd/\delta)}, 1\}$*

- *Let $m = \lambda^{-2} \operatorname{poly}(n, d, \exp(B))$*

- *Let $\eta = \lambda/(m \operatorname{poly}(n, d, \exp(B)))$*

- *Let $\widehat{T} = \Omega((m\eta\lambda)^{-1} \log(nd/\epsilon))$*

*Then, after $\widehat{T}$ iterations, with probability at least $1 - \delta$, we have*

$$\|\mathsf{F}(\widehat{T}) - Y\|_F^2 \le \epsilon.$$

*Proof.* We have $\|\mathsf{F}(0) - Y\|_F^2 \le nd$ as Lemma J.6. Using the choice of $\widehat{T}$, it follows directly from the alternative application of Lemma J.3 and Lemma J.4. □

## J.2 INDUCTION PART 1. FOR WEIGHTS

In this section, we introduce the induction lemma for weights.

**Lemma J.3** (Induction Part 1 for weights). *If the following conditions hold*

  • *Suppose properties in Definition J.1 are true*

*For $t + 1$ and $\forall r \in [m]$, it holds that:*

$$\|w_r(t+1) - w_r(0)\|_2 \le D.$$

*Proof.* We have

$$\eta \sum_{i=0}^{\infty} (1 - m\eta\lambda/2)^i \le \eta \frac{4}{m\lambda} \tag{18}$$

where this step follows from Fact E.2.

$$
\begin{aligned}
\|w_r(t+1) - w_r(0)\|_2 &\le \eta \sum_{\tau=0}^{t} \|\Delta w_r(\tau)\|_2 \\
&\le \eta \sum_{\tau=0}^{t} \sqrt{n}d \exp(11B) \cdot \|\mathsf{F}(t) - Y\|_F \\
&\le \eta \sqrt{n}d \exp(11B) \cdot \sum_{\tau=0}^{t} (1 - m\eta\lambda/2)^i \cdot \|\mathsf{F}(0) - Y\|_F \\
&\le 2\eta \frac{1}{m\lambda} \sqrt{n}d \exp(11B) \cdot \|\mathsf{F}(0) - Y\|_F \\
&\le D
\end{aligned}
$$

where the third step follows from the triangle inequality, the second step follows from Eq. (22), the third step follows from Lemma J.4, the fourth step follows from Eq. (18), the last step follows from *General Condition 4.* in Definition J.1.

□

## J.3 INDUCTION PART 2. FOR LOSS

Now, we present our next induction lemma.

**Lemma J.4** (Induction Part 2 for loss). *Let $t$ be a fixed integer.*

*If the following conditions hold*

  • *Suppose properties in Definition J.1 are true*

*Then we have*

$$\|\mathsf{F}(t+1) - y\|_F^2 \le (1 - m\eta\lambda/2)^{t+1} \cdot \|\mathsf{F}(0) - y\|_F^2.$$

*Proof.* We have

$$\|\mathsf{F}(t+1) - y\|_F^2$$

$$\leq \|\mathsf{F}(t) - y\|_F^2 + C_0 + C_1 + C_2 + C_3$$

$$= \|\mathsf{F}(t) - y\|_F^2 + C_0 + C_{1,1} + C_{1,2} + C_2 + C_3$$

$$\leq \|\mathsf{F}(t) - y\|_F^2 \cdot (1 + 0.1\eta m\lambda - 1.6\eta m\lambda + 0.1\eta m\lambda + \eta^2 m \cdot n^2 d^2 \exp(16B) + \eta^2 m^2)$$

$$\leq \|\mathsf{F}(t) - y\|_F^2 \cdot (1 - 1.4\eta m\lambda + \eta^2 m \cdot n^2 d^2 \exp(16B) + \eta^2 m^2) \tag{19}$$

where the first step follows from Lemma I.1, the second step follows from the definitions of $C_1$, $C_{1,1}$ and $C_{1,2}$, the third step follows from Lemma I.2 and Lemma I.3.

**Choice of parameter.** Here, we explain the condition setting in Definition J.1:

- To get our results in Lemma I.2 and Lemma I.3, we have to let $m \geq \Omega(\lambda^{-2}n^2 d^2 \cdot \exp(30B) \cdot \sqrt{\log(nd/\delta)})$.

- If we let $\eta \leq O(\lambda/(mn^2 d^2 \exp(16B)))$, we can have

$$\eta^2 m \cdot n^2 d^2 \exp(16B) + \eta^2 m^2 \leq 0.9\eta m\lambda. \tag{20}$$

Thus, combining Eq. (19) and Eq. (20), we have

$$\|\mathsf{F}(t+1) - y\|_F^2 \leq (1 - m\eta\lambda/2) \cdot \|\mathsf{F}(t) - y\|_F^2 \tag{21}$$

Then by Eq. (21), we conclude all $\|\mathsf{F}(\tau) - y\|_F^2$ for $\tau \in [t]$, we have

$$\|\mathsf{F}(t+1) - y\|_F^2 \leq (1 - m\eta\lambda/2)^{t+1} \cdot \|\mathsf{F}(0) - y\|_F^2$$

$\square$

## J.4 INDUCTION PART 3. FOR GRADIENT

In this section, we present the induction lemma for gradients.

**Lemma J.5** (Induction Part 3 for gradient). *Let $t$ be a fixed integer.*

*If the following conditions hold*

- *Suppose properties in Definition J.1 are true*

*Then we have*

$$\eta\|\Delta w_r(t)\|_2 \leq 0.01, \forall r \in [m]$$

*Proof.* Firstly, we have

$$\|\Delta w_r(t)\|_2 \leq \|\Delta w_r(t)\|_1$$

$$\leq \sum_{k_1=1}^{d} \left| m \sum_{i=1}^{n} \sum_{k=1}^{d} (\mathsf{F}_{k,i}(t) - y_{k,i}) \cdot \left( \langle v_{k,r}(t), \mathsf{S}_i(t) \rangle \cdot \mathsf{S}_{i,r}(t) \cdot x_{i,k_1} + a_r \mathsf{S}_{i,r}(t) e_{k,k_1} \right) \right|$$

$$\leq \sqrt{n}d \exp(11B)\|\mathsf{F}(t) - Y\|_F \tag{22}$$

where the first step follows from $\|U\|_F \leq \|U\|_1$ for $U \in \mathbb{R}^{n \times d}$, the second step follows from Claim G.12, the last step follows from the definition of 4 $\ell_1$ norm, $0 \leq \mathsf{S}_{i,r} \leq \frac{\exp(3B)}{m}$ by Part 11 of Lemma L.1, $\|x_i\|_2 \leq 1$ and Part 6 of Lemma L.2.

Then by the property of $\eta$ in Definition J.1, we have

$$\eta\|\Delta w_r(t)\|_2 \leq 0.01, \forall r \in [m]$$

$\square$

### J.5 BOUNDING LOSS AT INITIALIZATION

**Lemma J.6.** *If the following conditions hold*

- *Denote* $\mathsf{F}(\tau) \in \mathbb{R}^{n \times d}$ *as Definition G.8.*

- *Let* $Y \in \mathbb{R}^{n \times d}$ *denote the labels.*

*Then we have*

$$\|\mathsf{F}(0) - Y\|_F \leq O(\sqrt{nd})$$

*Proof.* This proof follows from $\|y_i\| \leq 1$ for $i \in [n]$ and Definition G.8. $\square$

## K NTK-ATTENTION

In this section, we compute the error bound of our NTK-Attention in approximating prefix matrix $P \in \mathbb{R}^{m \times d}$. In Appendix K.1, we provide the formal definition of our NTK-Attention. In Appendix K.2, we give our main theorem of error bound. In Appendix K.3, we state tools from (Alman & Song, 2023).

### K.1 DEFINITIONS

**Definition K.1.** *If the following conditions hold:*

- *Given input* $X \in \mathbb{R}^{L \times d}$, *prefix matrix* $P \in \mathbb{R}^{m \times d}$.

- *Let* $S := \begin{bmatrix} P \\ X \end{bmatrix} \in \mathbb{R}^{(m+L) \times d}$.

- *Given projections* $W_Q, W_K, W_V \in \mathbb{R}^{d \times d}$

- *Let* $Q := XW_Q \in \mathbb{R}^{L \times d}$.

- *Let* $K_P := SW_Q \in \mathbb{R}^{(m+L) \times d}$

- *Let* $V_P := SW_V \in \mathbb{R}^{(m+L) \times d}$

- *Let* $A := \exp(QK_P^\top) \in \mathbb{R}^{L \times (m+L)}$.

- *Let* $D := \mathrm{diag}(A\mathbf{1}_{(m+L)}) \in \mathbb{R}^{L \times L}$.

*We define:*

$$\mathsf{Attn}(Q, K, V) := D^{-1}AV_P.$$

### K.2 ERROR BOUND

Here, we provide our two statements about error bound.

**Theorem K.2** (Formal version of Theorem 4.1)**.** *Given an input matrix* $X \in \mathbb{R}^{L \times d}$ *and prefix matrix* $P \in \mathbb{R}^{m \times d}$, *we denote* $Q = XW_Q$, $K_C = PW_K$ *and* $V_C = PW_V$. *If the condition Eq.* (7), $\|Q\|_\infty \leq o(\sqrt{\log m})$, $\|K_C\|_\infty \leq o(\sqrt{\log m})$, $\|V_C\|_\infty \leq o(\sqrt{\log m})$ *and* $d = O(\log m)$ *holds, then Algorithm 2 outputs a matrix* $T \in \mathbb{R}^{L \times d}$ *within time complexity of* $O(L^2 d)$ *that satisfies:*

$$\|T - \mathsf{PrefixAttn}(X, P)\|_\infty \leq 1/\mathrm{poly}(m).$$

*Proof.* Following Definition K.1, we can have matrix $A \in \mathbb{R}^{L \times (m+L)}$ as follows:

$$A = QK^\top$$
$$= \begin{bmatrix} \exp(XW_QW_K^\top X^\top) & \exp(XW_QW_K^\top P^\top) \end{bmatrix}$$

where the second step follows from $K = SW_K$ and $S = \begin{bmatrix} P \\ X \end{bmatrix}$.

Our Algorithm 2 actually implement on using $Q = XW_Q$ and $PW_K$ to approximate $\exp(XW_QW_K^\top P^\top)$ by Lemma K.7.

Trivially, this proof follows from Theorem K.5 and Lemma K.7. $\qquad\square$

**Corollary K.3.** *Given an input matrix $X \in \mathbb{R}^{L \times d}$ and prefix matrix $P \in \mathbb{R}^{m \times d}$, we denote $Q = XW_Q$, $K_C = PW_K$ and $V_C = PW_V$. If the condition Eq. (7), $\|Q\|_\infty \leq o(\sqrt{\log m})$, $\|K_C\|_\infty \leq o(\sqrt{\log m})$, $\|V_C\|_\infty \leq o(\sqrt{\log m})$ and $d = O(\log m)$ holds, then there exists an algorithm that outputs a matrix $T \in \mathbb{R}^{L \times d}$ within time complexity of $O(L^{1+o(1)}d)$ that satisfies:*

$$\|T - \mathsf{PrefixAttn}(X, P)\|_\infty \leq 1/\operatorname{poly}(m).$$

*Proof.* The algorithm and proof can trivially follow from Algorithm 1, 2, 3 and Theorem 1 in HyperAttention (Han et al., 2024). $\qquad\square$

### K.3 Tools from Fast Attention

In this section, we introduce some tools from previous work which we have used.

**Definition K.4** (Approximate Attention Computation $\mathsf{AAttC}(n, d, B, \epsilon_a)$, Definition 1.2 in (Alman & Song, 2023)). *Let $\epsilon_a > 0$ and $B > 0$ be parameters. Given three matrices $Q, K, V \in \mathbb{R}^{n \times d}$, with the guarantees that $\|Q\|_\infty \leq B$, $\|K\|_\infty \leq B$, and $\|V\|_\infty \leq B$, output a matrix $T \in \mathbb{R}^{n \times d}$ which is approximately equal to $D^{-1}AV$, meaning,*

$$\|T - D^{-1}AV\|_\infty \leq \epsilon_a.$$

*Here, for a matrix $M \in \mathbb{R}^{n \times n}$, we write $\|M\|_\infty := \max_{i,j} |M_{i,j}|$.*

**Theorem K.5** (Upper bound, Theorem 1.4 in (Alman & Song, 2023)). *There is an algorithm that solves $\mathsf{AAttC}(n, d = O(\log n), B = o(\sqrt{\log n}), \epsilon_a = 1/\operatorname{poly}(n))$ in time $n^{1+o(1)}$.*

**Definition K.6** (Definition 3.1 in (Alman & Song, 2023)). *Let $r \geq 1$ denote a positive integer. Let $\epsilon \in (0, 0.1)$ denote an accuracy parameter. Given a matrix $A \in \mathbb{R}_{\geq 0}^{n \times n}$, we say $\widetilde{A} \in \mathbb{R}_{\geq 0}^{n \times n}$ is an $(\epsilon, r)$-approximation of $A$ if*

- $\widetilde{A} = U_1 \cdot U_2^\top$ *for some matrices $U_1, U_2 \in \mathbb{R}^{n \times r}$ (i.e., $\widetilde{A}$ has rank at most $r$), and*

- $|\widetilde{A}_{i,j} - A_{i,j}| \leq \epsilon \cdot A_{i,j}$ *for all $(i, j) \in [n]^2$.*

**Lemma K.7** (Lemma 3.4 in (Alman & Song, 2023)). *Suppose $Q, K \in \mathbb{R}^{n \times d}$, with $\|Q\|_\infty \leq B$, and $\|K\|_\infty \leq B$. Let $A := \exp(QK^\top/d) \in \mathbb{R}^{n \times n}$. For accuracy parameter $\epsilon \in (0, 1)$, there is a positive integer $g$ bounded above by*

$$g = O\Big( \max\Big\{ \frac{\log(1/\epsilon)}{\log(\log(1/\epsilon)/B^2)}, B^2 \Big\} \Big),$$

*and a positive integer $r$ bounded above by*

$$r \leq \binom{2(g+d)}{2g}$$

*such that: There is a matrix $\widetilde{A} \in \mathbb{R}^{n \times n}$ that is an $(\epsilon, r)$-approximation (Definition K.6) of $A \in \mathbb{R}^{n \times n}$. Furthermore, we can construct the matrices $U_1 := \phi(Q)$ and $U_2 := \phi(K)$ through a function $\phi(\cdot)$ defining $\widetilde{A} = U_1 U_2^\top$ can be computed in $O(n \cdot r)$ time.*

## L Taylor Series

In this section, we provide some perturbation analysis for NTK analysis.

**Lemma L.1** (Lemma B.1 in (Li et al., 2024a)). *If the following conditions hold*

- *Let $C > 10$ denote a sufficiently large constant*

- *Let $B := \max\{C\sigma\sqrt{\log(nd/\delta)}, 1\}$.*

- *Let $W = [w_1, \cdots, w_m]$ and $w_r$ be random Gaussian vectors from $\mathcal{N}(0, \sigma^2 I_d)$.*

- *Let $V = [v_1, \cdots, v_m]$ and $v_r$ denote the vector where $\|v_r - w_r\|_2 \leq R$, $\forall r \in [m]$.*

- *Let $x_i \in \mathbb{R}^d$ and $\|x_i\|_2 \leq 1$, $\forall i \in [n]$.*

- *Let $R \in (0, 0.01)$.*

- *Let $\mathsf{S}_i$ and $\widetilde{\mathsf{S}}_i$ be the softmax function corresponding to $W$ and $V$ respectively.*

- *Let $\alpha_i = \langle \mathbf{1}_m, \exp(W^\top x_i) \rangle$ and $\widetilde{\alpha}_i = \langle \mathbf{1}_m, \exp(V^\top x_i) \rangle$, $\forall i \in [n]$.*

*Then, with probability at least $1 - \delta/\operatorname{poly}(nd)$, we have*

- *Standard inner product*

  - *Part 1. $|\langle w_r, x_i \rangle| \leq B$, $\forall i \in [n]$, $\forall r \in [m]$*
  - *Part 2. $|\langle v_r, x_i \rangle| \leq B + R$, $\forall i \in [n]$, $\forall r \in [m]$*
  - *Part 3. $|\langle w_r - v_r, x_i + x_j \rangle| \leq 2R$, $\forall i, j \in [n]$, $\forall r \in [m]$*

- $\exp$ *function*

  - *Part 4. $\exp(-B) \leq \exp(\langle w_r, x_i \rangle) \leq \exp(B)$, $\forall i \in [n]$, $\forall r \in [m]$*
  - *Part 5. $\exp(-B - R) \leq \exp(\langle v_r, x_i \rangle) \leq \exp(B + R)$, $\forall i \in [n]$, $\forall r \in [m]$*
  - *Part 6. $|\exp(\langle w_r - v_r, x_i + x_j \rangle) - 1| \leq 4R$, $\forall i, j \in [n]$, $\forall r \in [m]$*
  - *Part 7. $|\exp(\langle w_r, x_i \rangle) - \exp(\langle v_r, x_i \rangle)| \leq R \exp(B + R)$, $\forall i \in [n]$, $\forall r \in [m]$*

- *softmax $\mathsf{S}$ function*

  - *Part 8. $|\alpha_i - \widetilde{\alpha}_i| \leq mR \exp(B + R)$, $\forall i \in [n]$*
  - *Part 9. $|\alpha_i^{-1} - \widetilde{\alpha}_i^{-1}| \leq \frac{R}{m} \exp(3B + 2R)$, $\forall i \in [n]$*
  - *Part 10. $|\mathsf{S}_{i,r}| \leq \exp(2B)/m$, $\forall i \in [n]$, $\forall r \in [m]$*
  - *Part 11. $|\widetilde{\mathsf{S}}_{i,r}| \leq \exp(2B + 2R)/m$, $\forall i \in [n]$, $\forall r \in [m]$*
  - *Part 12. $|\mathsf{S}_{i,r} - \widetilde{\mathsf{S}}_{i,r}| \leq \frac{R}{m} \exp(4B + 3R)$, $\forall i \in [n]$, $\forall r \in [m]$*
  - *Part 13. for any $z \in \mathbb{R}^m$ and $\|z\|_\infty \leq 1$, we have $|\langle z, \mathsf{S}_i \rangle - \langle z, \widetilde{\mathsf{S}}_i \rangle| \leq R \exp(4B + 3R)$, $\forall i \in [n]$*

**Lemma L.2.** *If the following conditions hold*

- *Let $C > 10$ denote a sufficiently large constant*

- *Let $B := \max\{C\sigma\sqrt{\log(nd/\delta)}, 1\}$.*

- *Let $W = [w_1, \cdots, w_m]$ and $w_r$ be random Gaussian vectors from $\mathcal{N}(0, \sigma^2 I_d)$.*

- *$w_r$ for $r \in [m]$ satisfies $\|w_r\|_2 \leq B$ with probability at least $1 - \delta/\operatorname{poly}(nd)$ as in Lemma L.1.*

- *Let $a \in \mathbb{R}^m$ be defined as Definition F.1.*

- *Define $\beta_k := W_{k,*} \circ a \in \mathbb{R}^m$ for $k \in [d]$ as Definition F.5.*

- *Define $v_{k,r} := \beta_{k,r} \cdot \mathbf{1}_m - \beta_k \in \mathbb{R}^m$ for $k \in [d]$ and $r \in [m]$ as Definition H.1.*

- *Define $\alpha_i$ for $i \in [n]$ as Definition F.3.*

*Then, with probability at least $1 - \delta/\operatorname{poly}(nd)$, we have*

- *Part 1. $|\beta_{k,r}| \leq B$*

- *Part 2.* $\|\beta_k\|_2 \le B\sqrt{m}$

- *Part 3.* $\|v_{k,r}\|_2 \le 2\sqrt{m}B$

- *Part 4.* $|\alpha^{-1}| \le \exp(B)/m$

- *Part 5.* $\langle \beta_k, \mathsf{S}_i \rangle \le \exp(4B)$

- *Part 6.* $\langle v_{k,r}, \mathsf{S}_i \rangle \le \exp(6B)$

*Proof.* **Proof of Part 1.** We can get the proof by Gaussian tail bound.

**Proof of Part 2.** We have

$$\|\beta_k\|_2 = \sqrt{\sum_{r=1}^{m} \beta_{k,r}^2}$$

$$\le \sqrt{\sum_{r=1}^{m} B^2}$$

$$\le \sqrt{m} \cdot B$$

where the first step follows from the definition of $\ell_2$ norm, the second step follows from Part 1 of this Lemma, the last step follows from simple algebras.

**Proof of Part 3.** We have

$$\|v_{k,r}\|_2 = \sqrt{\sum_{r_1=1}^{m} (\beta_{k,r} - \beta_{k,r_1})^2}$$

$$\le \sqrt{\sum_{r_1=1}^{m} \beta_{k,r}^2 + \beta_{k,r_1}^2 + |2\beta_{k,r}\beta_{k,r_1}|}$$

$$\le \sqrt{\sum_{r_1=1}^{m} 4B^2}$$

$$\le 2\sqrt{m} \cdot B$$

where the first step follows from the definition of $\ell_2$ norm, the second step follows from simple algebras, the third step follows from Part 1 of this Lemma, the last step follows from simple algebras.

**Proof of Part 4.** This proof follows from Part 4 of Lemma L.1 and Definition F.3.

**Proof of Part 5.** We have

$$\langle \beta_k, \mathsf{S}_i \rangle \le \|\beta_k\|_2 \cdot \|\mathsf{S}_i\|_2$$

$$\le \sqrt{m}B \cdot \|\mathsf{S}_i\|_2$$

$$\le \sqrt{m}B \cdot \sqrt{\sum_{r=1}^{m} \mathsf{S}_{i,r}^2}$$

$$\le \sqrt{m}B \cdot \sqrt{\sum_{r=1}^{m} \frac{\exp(6B)}{m^2}}$$

$$\le \sqrt{m}B \cdot \sqrt{\frac{\exp(6B)}{m}}$$

$$\le B \exp(3B)$$

$$\le \exp(4B)$$

where the first step follows from Cauchy-Schwarz inequality, the second step follows from Part 2 of this Lemma, the third step follows from the definition of $\ell_2$ norm, the fourth step follows from Part 11 of Lemma L.1, the fifth step follows from triangle inequality, the sixth step follows from $B \le \exp(B)$, last step follows from simple algebras.

**Proof of Part 6.** This proof follows from Part 3 of this Lemma, $B \le \exp(B)$ and Part 11 of Lemma L.1. $\qquad\square$

