# OpenReview forum: "Towards Infinite-Long Prefix in Transformer"
_ICLR.cc/2025/Conference — Submitted to ICLR 2025_

### Official Review · Reviewer_oFkK · 2024-10-31

**Soundness:** 2
**Presentation:** 3
**Contribution:** 2
**Rating:** 3
**Confidence:** 4

**Summary:**

This paper presents an analysis of ultra-long prefix vectors (P-Tuning V2), focusing exclusively on this version of prefix-tuning. The study utilizes the Neural Tangent Kernel (NTK) to examine ultra-long prefix vector convergence and subsequently proposes NTK-attention, derived from observations in the analysis.

**Strengths:**

The paper analyzes P-Tuning V2 through the lens of the Neural Tangent Kernel (NTK), offering a fresh perspective on prefix-tuning. The analysis is precise, highly detailed, and appears to have involved considerable effort.

**Weaknesses:**

1. This analysis does not cover various prompt tuning methods but is instead focused solely on P-Tuning V2, which employs different embeddings per layer. In this case, it is evident that the model is equivalent to using two linear layers with tied weights, making this insight neither new nor surprising. This raises questions about the novelty of the analysis itself. The difficulty in analyzing prefix-tuning lies in the fact that the prefixes passed through the input layer interact with each other in every layer via attention before moving to the next layer. However, this paper lacks analysis in that regard.

2. In my view, contrary to the author’s claim, NTK-attention is not parameter-efficient. The number of parameters is quite large:  as shown on Line 345, $r = d$, resulting in $d^2+d$ parameters. This count is significantly higher than the $md$ parameters used in prefix-tuning (experiments were conducted with prefix counts $m$ ranging from 1 to 200). This discrepancy calls into question whether the comparative experiments with P-Tuning V2 and LoRA are indeed fair. The author notes using LLama 3.1 1B for the experiments, and given that this model has an embedding dimension of 2048, a considerable number of parameters are indeed used (more than x10). If the author intends to argue that their method is more parameter-efficient than prefix-tuning, they should disclose the parameter counts used in Tables 1 and 2.

3. Computational efficiency also does not surpass that of prefix-tuning. The $\text{O}(\text{L}^2)$ notation in Figure 1 is incorrect; due to the $\Phi(Q)Z$ operation in Line 5 of Algorithm 2, the complexity of NTK-attention is $\text{O}(\text{(L+d+1)L})$ ($d=r$ as mentioned above). Although Appendix Figure 3 provides a running time comparison, it is not a fair comparison with $d=32$ as the value of $m$ is ranging from $1$~$2^{16}$ (The complexity of prefix-tuning $\text{O}(\text{(L+m)L})$ is quite large in this setting. And nobody use more than 1K tokens for the prefix-tuning.). For a fair experiment, the author should test with common embedding size $d$ such as 2048 or 4096, frequently used in various foundation models.  In this experimental setting, since $m\leq d$ in most cases, NTK-attention is likely to incur an even greater computational load.

4. The advantage of prefix-tuning lies in achieving fine-tuning level performance by adjusting only the input, enabling parallel execution across various tasks. However, since NTK-attention modifies the model itself, this advantage is lost.

5. The author should conduct additional experiments across various foundation models and tasks.

**Questions:**

See weaknesses.

+) If the mentioned weaknesses are adequately addressed in the rebuttal, I would be willing to raise the score.

---

> ### Author Response · Authors · 2024-11-22
>
> We sincerely thank you for your reviews and helpful suggestions. We address your concerns as follows:
>
> ### W1: This analysis does not cover various prompt tuning methods but is instead focused solely on P-Tuning V2.
> Thank you for pointing out these insightful comments. In our method, we can have different trainable parameters for different layers to approximate P-Tuning V2. Also, we can  (1) only introduce trainable parameters in the first layer or (2) share the same trainable parameters among different layers to approximate other prompt tuning methods. However, on the theoretical side, these methods may require new analysis and we leave these interesting directions as our future works.
>
>
> ### W2: NTK-attention is not parameter-efficient.
>
> Thank you for your comments. We used low-rank adaptation to achieve an efficient parameter representation of $Z$. In detailed, we decompose $Z(0) = Z_A(0) \cdot Z_B(0)$, where $Z_A(0) \in \mathbb{R}^{r \times s}$ and $Z_B \in \mathbb{R}^{s \times d}$. We choose $s \leq \lfloor d/2 \rfloor$ as an appropriately small integer, in Line 361-370 of the revision.
>
> On the other hand, in updated Table 1, 2, and 3, we disclose the number of parameters each fine-tuning uses. Since the number of trainable parameters of NTK-Attention is close to LoRA but indicates better performance, this confirms the efficiency and effectiveness of NTK-Attention. We refer the reviewer to the Global Response section **Supplementary Implementation Detail** for more details.
>
>
> ### W3: Computational efficiency also does not surpass that of prefix-tuning.
>
> We acknowledge this computational efficiency limitation of NTK-Attention compared to prefix attention with $m \leq d$. However, in the scope of this paper, we focus on the comparison with prefix attention with large prefix length $m$. Since we confirm the strong learning ability of an ultra-long prefix learning in Section 3, the proposition of NTK-Attention is how to catch this strong ability efficiently. NTK-Attention provides a probability for the longer extension of prefix length in prefix-based fine-tuning methods, while only costing a few computational complexity (see Figure 3) and few trainable parameters (see Table 1, 2, 3, 4). Moreover, the superior performance of NTK-Attention we showed in our experiments is worth using such complexity in exchange for.
>
> On the other hand, the assumption of a big $m$ is common, and it could also be supported by **Reclaim the Practicality of Assumption towards $m \gg L$ and $m \gg d$** section in Global Response.
>
> ### W4: The advantage of prefix-tuning lies in achieving fine-tuning level performance by adjusting only the input, enabling parallel execution across various tasks. However, since NTK-attention modifies the model itself, this advantage is lost.
>
> Thank you for pointing this concern out. We argue this is not our issue by providing a very easy Python code (only 10 lines) in Appendix C, which is a naive implementation of NTK-Attention. This substantiates the simplicity of our method of implementation. As we access the Flash Attention function [1], we only need to add such a shortcode to implement our method.
>
> ### W5: The author should conduct additional experiments across various foundation models and tasks.
>
> Thank you for suggesting this. We add new experiments to extend the experimental scope, which is regarding scalability with large language models in real-world scenarios. We refer the reviewer to **Supplementary Experiments** in Global Response to see the details and results of our new experiments.
>
> On the other hand, a prompt-based fine-tuning method processes to fit on various tasks at once fine-tuning, while our NTK-Attention can also qualify it since we can approximate a prefix attention computation with arbitrary prefix length by following Eq. (7).
>
> ### References
>
> [1] Tri Dao, Dan Fu, Stefano Ermon, Atri Rudra, and Christopher Ré. Flashattention: Fast and memory- efficient exact attention with io-awareness. NeurIPS’22

---

> ### Comment · Reviewer_oFkK · 2024-11-27
>
> Thank the authors for the response.
>
> Regarding the first question in the "Weakness" section, compared to the previous NTK analysis, what distinguishes your approach from the original NTK analysis? It seems the only addition is an extra layer, which makes the analysis appear less novel.
>
> Additionally, the main theme of your paper is that infinitely long prefixes can be reduced to your NTK attention. However, the final form appears quite similar to LoRA (The authors did not even mention the use of low-rank approximation in $Z$ in the initial version. ).
>
> Regarding the practicality of long prefixes, I believe it mostly relates to hard prompting and in-context learning. However, as demonstrated in recent RAG papers (which pre-compress the context), the concept of an infinitely long prefix seems unnatural. Moreover, prior research on soft prompting suggests that longer prefixes don’t necessarily lead to better performance and may degrade it at certain points.
>
> I also noticed that the computational complexity in Figure 1 appears to be incorrect, and this hasn't been addressed in the current version.
>
> I feel that my concerns have not been adequately addressed in the paper.

---

> > ### Comment · Reviewer_oFkK · 2024-11-27
> >
> > +)
> > Regarding weakness 1, I am not concerned about whether NTK can use different parameters for different layers, but rather whether this NTK analysis can be extended beyond P-TuningV2 to other prefix-tuning techniques. This is an important consideration that seems to be overlooked.

---

> > > ### Author Response · Authors · 2024-11-28
> > > **Thank you and further reply to new concerns (Part 1)**
> > >
> > > We sincerely thank reviewer’s reply. We appreciate the reviewer's insightful comments. We try to address your further concerns below.
> > >
> > > ### Q1: Regarding the first question in the "Weakness" section, compared to the previous NTK analysis, what distinguishes your approach from the original NTK analysis? It seems the only addition is an extra layer, which makes the analysis appear less novel.
> > >
> > > Thank you for your question! Our analysis are novel in the following sense:
> > > - The model function (Eq. (3)) we study in this work is derived from a simplification of prefix learning, to the best of our knowledge, which has not been studied before. Studying its convergence and training dynamic could help us to understand prefix learning.
> > > - Our theoretical analysis focuses on the novel setting of softmax activation function that rarely appears in prior NTK-style analysis. In particular, the output of the softmax function would converge to nearly zero since the input length becomes too long, which makes analyzing its dynamic challenging.
> > > - At the same time, training two weight-sharing linear networks is not similar to training two different linear networks (NTK-style convergence of three layers [1]), to our best knowledge, this is the first analysis of such a specific network structure.
> > >
> > > Below, we summarize our technical contribution in detail:
> > > - The kernel convergence during the training to keep the PSD property of $H(W)$ is totally different from the classical NTK-style linear networks. We didn’t use concentration bound as classical proofs since we consider the worst case. In general, $\mathbb{E}[H(W ) − H( \widetilde{W} )] \ne {\bf 0}_{nd \times nd}$. Thus, using the concentration bound may not gain any benefits (see Remark H.2). This remains difficult in analysis of the upper bound on the kernel convergence and we solve it in Lemma H.3 and Lemma H.4.
> > > - In our proofs, we study the discrete-time optimization case, Lemma I.1 first decomposes the dynamic of the training objective in each timestep. Naive decomposing derivations usually fail since the softmax is too complicated to compute. We introduced the Taylor expansion of exponential function to address this (see Fact E.1). Next, computing upper bounds for each decomposed term (Lemma I.2, I.3, I.4, I.5, I.6, I.7) is also challenging.
> > > - Otherwise, introducing two weight-sharing linear networks brings an additional term in gradient computation as we highlighted in Claim G.12. This term cannot help proving NTK and is a brand-new theoretical issue to derive the kernel function in training our model function ${\sf F}(x, W, a)$. We also succeed in solving it by the skillful decomposition to the training objective.
> > >
> > >
> > >
> > >
> > > ### Q2.1: the final form appears quite similar to LoRA
> > >
> > > Thank you for pointing this out! We would like to clarify that we have discussed and emphasized the difference between our NTK-Attention and LoRA in Section 4.2 of the original version manuscript. Currently, we move this part to Line 1334-1343 in Appendix D. Simply put, LoRA makes adaptation on Query and Value projections $W_Q, W_V \in \mathbb{R}^{d \times d}$; denote the adaptation as $W_{\Delta Q}, W_{\Delta V} \in \mathbb{R}^{d \times d}$. Given an input $X\in \mathbb{R}^{L \times d}$, LoRA computes $\widetilde{D}^{-1} \widetilde{A} X ( W_V + W_{\Delta V})$, where $\widetilde{A} := \exp(X (W_Q + W_{\Delta Q})W_K^\top X^\top)$, $\widetilde{D} := {\rm diag}(\widetilde{A} {\bf 1}_L)$, and $W_K \in \mathbb{R}^{d \times d}$ is the Key projection weights. So LoRA updates query and value weights during training, while our NTK-Attention compresses the additional prefix $P $ into $Z  $ and $k $  (Algorithm 2) but does not modification of query and value projection $W_Q$ and $W_K$, which is a completely different mechanism.
> > >
> > >
> > >
> > > ### Q2.2: The authors did not even mention the use of low-rank approximation in Z in the initial version
> > >
> > > We are sorry our presentation led to a misunderstanding. In our first submitted manuscript, for the convenience of readers' understanding and to make the paper focus on our main message of theoretical analysis, we did not disclose the specific implementation of a trainable matrix $Z$ with using low-rank representation, but the details we claim below can be found in the code of the supplementary materials in our first submitted version. When we realize the importance of presenting the parameter-efficiency for comparison, we re-clarify its details.

---

> > > ### Author Response · Authors · 2024-11-28
> > > **Thank you and further reply to new concerns (Part 2)**
> > >
> > > ### Q3: Regarding the practicality of long prefixes, I believe it mostly relates to hard prompting and in-context learning. However, as demonstrated in recent RAG papers (which pre-compress the context), the concept of an infinitely long prefix seems unnatural. Moreover, prior research on soft prompting suggests that longer prefixes don’t necessarily lead to better performance and may degrade it at certain points.
> > >
> > > Thank you for your question. What we want to indicate in our theory is not that longer prefixes bring better performance. In contrast, refer to Line 270-294 in Section 3.3, our theory confirms the existence of *scaling law in prefix learning*. This means that a considerably long prefix is the crucial reason for better performance but not all it needs. Actually, we confirm a longer prefix would only bring stronger learning ability, whereas a better performance also requires more computation resources and more high-quality data [2].
> > >
> > > In our NTK-Attention that approximates long prefix attention, the motivation is to maximize this theoretically guaranteed strong learning ability for fine-tuning LLMs on downstream tasks.
> > >
> > >
> > > ### Q4: I also noticed that the computational complexity in Figure 1 appears to be incorrect, and this hasn't been addressed in the current version.
> > >
> > > Thank you for reminding us of this point. We have modified Figure 1 and its caption by emphasizing $m \gg d$ and adding an illustration of the structure of our NTK-Attention with low-rank approximating $Z$. We refer the reviewer to the second version of the manuscripts and the new global response for our modifications. Let us know if the reviewer has more concerns.
> > >
> > > ### Q5: Regarding weakness 1, I am not concerned about whether NTK can use different parameters for different layers but rather whether this NTK analysis can be extended beyond P-TuningV2 to other prefix-tuning techniques. This is an important consideration that seems to be overlooked.
> > >
> > > Thank you so much for your questions. We analyze P-TuningV2 because it is a popular state-of-the-art method. Also, we can easily extend our theory to the case where the prefix matrix appears only in the first layer, which is a degenerate version of P-TuningV2.
> > >
> > > On the other hand, we would like to argue the analysis hardness of other prefix learning methods. We give some examples here:
> > > - There are some works that select the discrete embedding or their linear combination from the embedding dictionary as the prefix. This method of analysis requires understanding the delicate structure of the embedding vector space and requires sophisticated assumptions to describe the relationship between embedding space and downstream tasks.
> > > - Some works use the same prefix among different layers. To study this scenario, the number of variables will be large, which makes the analysis redundant and complicated, e.g., there may be many terms in the gradient, and it is hard to give a concrete analysis.
> > >
> > > Furthermore, we would like to argue that theoretical analysis is hard, e.g., one line of code change may require a totally different analysis strategy, e.g., many analyses in optimizer. This corresponds to the empirical side. There are many good methods, that only require one line code difference, and have different behaviors, meaning that the backend mechanism is different, leading to different analyses. On the other hand, we agree that a general framework will be better. We will make it our ultimate goal. We thank you again for the reviewer's valuable comments.
> > >
> > > We hope that our response can relieve your concerns. We are willing to discuss more if the reviewer has more follow-up questions.
> > >
> > > ### References:
> > >
> > > [1] Learning and Generalization in Overparameterized Neural Networks, Going Beyond Two Layers. Zeyuan Allen-Zhu, Yuanzhi Li, Yingyu Liang. NeurIPs’2019.
> > >
> > > [2] Jared Kaplan, Sam McCandlish, Tom Henighan, Tom B Brown, Benjamin Chess, Rewon Child, Scott Gray, Alec Radford, Jeffrey Wu, and Dario Amodei. Scaling laws for neural language models. OpenAI’20.

---

> > > > ### Author Response · Authors · 2024-12-02
> > > > **Looking forward to receiving your feedback**
> > > >
> > > > Dear Reviewer oFkK,
> > > >
> > > > We hope we have adequately addressed your new concerns. We would be very grateful if you could provide feedback since the discussion deadline is approaching in one day. If you require further clarification or have any additional concerns, please do not hesitate to contact us. We are more than willing to continue communicating with you.
> > > >
> > > > Warmest regards,
> > > >
> > > > Authors

---

> > > > > ### Comment · Reviewer_oFkK · 2024-12-02
> > > > >
> > > > > Dear Authors,
> > > > >
> > > > > Thank you for your feedback and for revising the paper.
> > > > >
> > > > > However, I still have some concerns regarding the connection between the low-rank approximation of $Z$ and Eq. (7). I believe a more detailed justification is necessary for the use of the low-rank approximation in this context.
> > > > >
> > > > > Additionally, you mention that the NTK analysis in the paper can be generalized to the case of prefix tuning, where the prefix matrix only appears in the first layer. However, in your analysis, the prefix matrix $P$ is not updated throughout the layers. Could you clarify how the analysis can be extended to this case where the prefix matrix is updated during training?
> > > > >
> > > > > Finally, I am still unsure about the scaling law for the prefix. If that extremely long prefix is required, why not train the model from scratch? The original NTK analysis suggests that training from scratch would guarantee convergence.

---

> > > > > > ### Author Response · Authors · 2024-12-02
> > > > > > **Thank you and further reply to new concerns**
> > > > > >
> > > > > > We sincerely thank you for your constructive suggestions and questions. Here we address your follow-up concerns below.
> > > > > >
> > > > > > ### Q1.  detailed justification regarding the connection between the low-rank approximation of Z and Eq. (7)
> > > > > >
> > > > > > The low-rank approximation of $Z$ is used for empirical purposes only to save parameter numbers and computation resources. This proxy method is well used in many deep learning algorithms [1, 2] and so on. On the other hand, we find that using the low-rank version does not hurt the performance compared to the full-rank version.
> > > > > >
> > > > > > ### Q2. clarify how the analysis can be extended to this case where the prefix matrix is updated during training
> > > > > >
> > > > > > Thank you for your question! Recall the Hierarchical learning theory [3], training prefix-tuning is equivalent to training the first-layer transformer with a prefix matrix to learn some ideal outputs. Our analysis can easily extend to this case since it only introduces some additional gradient from the latter layers (i.e., layers except the first layer) to the original NTK-style gradient, which does not hurt the NTK analysis framework. Following our decomposition of the training objective in Lemma I.1, these additional gradients can be decomposed into at most two additional terms and finally be bound by a considerably large $m$. We will add this additional analysis in the next version since we are not allowed to modify the manuscript.
> > > > > >
> > > > > > ### Q3. why not train the model from scratch?
> > > > > >
> > > > > > Thank you for your question! You are correct, we can indeed train LLM with prefixes from scratch to enhance the learning ability of language models since the confirmed scaling law property in our work. Actually, this method has already appeared in the LLM community, it is called PrefixLM [4]. Some prior works confirmed that PrefixLM [4] performs better than standard LM (a.k.a CasualLM) in some tasks (e.g., in-context learning) [5, 6]. Training PrefixLM from scratch to outperform CasaulLM is a potential future direction. However, under this setting, scaling prefix length up to extremely long is still impossible. The prefix length will aggravate the quadratic complexity of attention. We thus think that is the reason why the AI community hasn’t kept developing super-long PrefixLM before.
> > > > > >
> > > > > > On the other hand, training our NTK-Attention from scratch is also impossible. The parameters in $Z$ and $k$ are highly dependent on the attention weights of key and value projections $W_K$ and $W_V$. Once $W_K$ and $W_V$ change, the guarantee of our Theorem 4.1 will be immediately destroyed. We have tested that updating $W_K$, $W_V$, $Z$, and $k$ at the same time leads only to bad performance.
> > > > > >
> > > > > > Finally, we would like to point out that our algorithm design and evaluation focus on PEFT tasks. PrefixLM might be more like extra application scenarios that provide further motivation and justify long prefix learning, but they are not our focus.
> > > > > >
> > > > > > We hope that our new response can relieve your concerns.
> > > > > >
> > > > > > References:
> > > > > >
> > > > > > [1] LoRA: Low-Rank Adaptation of Large Language Models. Edward J. Hu, Yelong Shen, Phillip Wallis, Zeyuan Allen-Zhu, Yuanzhi Li, Shean Wang, Lu Wang, and Weizhu Chen. ICLR’22
> > > > > >
> > > > > > [2] QLoRA: Efficient Finetuning of Quantized LLMs. Tim Dettmers, Artidoro Pagnoni, Ari Holtzman, Luke Zettlemoyer. arXiv preprint: 2305.14314
> > > > > >
> > > > > > [3] Yoshua Bengio, Pascal Lamblin, Dan Popovici, and Hugo Larochelle. Greedy layer-wise training of deep networks. NeurIPS’06
> > > > > >
> > > > > > [4] Exploring the limits of transfer learning with a unified text-to-text transformer. Colin Raffel, Noam Shazeer, Adam Roberts, Katherine Lee, Sharan Narang, Michael Matena, Yanqi Zhou, Wei Li, and Peter J. Liu. JMLR’20
> > > > > >
> > > > > > [5] CausalLM is not optimal for in-context learning. Nan Ding, Tomer Levinboim, Jialin Wu, Sebastian Goodman, and Radu Soricut. ICLR’24
> > > > > >
> > > > > > [6] Examining scaling and transfer of language model architectures for machine translation. Biao Zhang, Behrooz Ghorbani, Ankur Bapna, Yong Cheng, Xavier Garcia, Jonathan Shen, and Orhan Firat. ICML’22

---

> > > > > > > ### Comment · Reviewer_oFkK · 2024-12-03
> > > > > > >
> > > > > > > Thank you for your feedback.
> > > > > > >
> > > > > > > Q1.
> > > > > > > In the case of LoRA and other methods, a low-rank approximation is justified based on their findings, such as the intrinsic dimension. However, in this version of the paper, I do not see a clear justification for applying a low-rank approximation to
> > > > > > > $Z$. Could you provide more details or rationale for this?
> > > > > > >
> > > > > > > Q2.
> > > > > > > If the transformer layers receive the prefix matrix $P$, the matrix $P$ will change its values from layer to layer, affecting the other inputs (e.g., image or text embeddings) differently at each layer during the forward pass. However, in your analysis, $P$ is treated as fixed. Could you clarify how your analysis can be extended to the case where $P$ is updated or modified across layers in the forward pass?
> > > > > > >
> > > > > > > This will be my final question, and I will evaluate the final score based on your response.

---

> ### Author Response · Authors · 2024-12-04
> **Thank you and final reply by authors**
>
> We appreciate your time and further reply. We hope that our answer to the following can relieve your concerns.
>
> ### Q1: Justification for applying a low-rank approximation to Z.
> Thank you so much for your questions! In the revision **Table 4** (we also highlight part of them below) on page 24 (previously used to reply to Q1 of Reviewer LAin), we can see that the low-rank approximation version, e.g, $(r,s)=(128,4)$, and large-rank version $(r,s)=(128,64)$ have similar performance, as shown in below table. Thus, we claim that $Z$ has a low-dimensional structure. We hope that this ablation study can be a justification for addressing your concerns. We will conduct more justification experiments in the next version. Furthermore, theoretically studying why $Z$ can be approximated by low rank is an interesting future direction. We leave it as a future work.
>
> |  Parameter  |  Evaluation Loss | Training Loss  |
> |--------|--------|--------|
> | $(r,s)=(128,4)$   |  2.48  | 2.38 |
> | $(r,s)=(128,64)$   |  2.41  | 2.31 |
>
> ### Q2: Could you clarify how your analysis can be extended to the case where P is updated or modified across layers in the forward pass?
>
> To analyze this situation, we would like to reclaim that we are able to consider the case where the prefix matrix parameters $P$ are only in the first layer. Although the prefix matrix is involved in the computations of further layers in the model and eventually input to the final linear classifier, during the back-propagation, this could only introduce some additional terms on gradients that compare to P-Tuning V2.
>
> In detail, assume we have $h$ layers of transformers, denoted as $\mathsf{TF}\_1, \dots, \mathsf{TF}\_h$. Then, the model is $f(x) = \mathsf{TF}\_h \circ \mathsf{TF}\_{h-1} \circ \dots \circ \mathsf{TF}\_1(x)$. Given a data point $(x,y)$, for one layer setting, we have the gradient be $\frac{d L(f(x), y)}{ d P}  = \frac{d L(f(x), y)}{ d \mathsf{TF}\_1(x)} \cdot \frac{d \mathsf{TF}\_1(x) }{d P}  $ by chain rule. For $h$ layer, by chain rule we still have $\frac{d L(f(x), y)}{ d P}  = \frac{d L(f(x), y)}{ d \mathsf{TF}\_1(x)} \cdot \frac{d \mathsf{TF}\_1(x) }{ d P}  $. Here, we can view the loss function as $L( (\mathsf{TF}\_h \circ \mathsf{TF}\_{h-1} \circ \dots \circ \mathsf{TF}\_2) \circ f’(x), y)$, where $f’ = \mathsf{TF}\_1$, rather than $L(f(x), y)$.  In other words, we can absorb the latter layers into the loss function as they are deterministic.
>
> Now we focus on these additional terms on gradients, e.g., $\frac{d L(f(x), y)}{ d \mathsf{TF}\_1(x)}$, it can also be trivially handled by a considerably large $m$ as we proved in Appendix I. Therefore, we can conclude this situation within some extra equations in our proof. We will add this additional analysis in the next version since we are now not allowed to modify the manuscript.
>
> ### Thank you!
> Moreover, we would like to sincerely thank you again since you provided lots of constructive suggestions and insightful questions, which helped us greatly enhance the contribution of our work and further polish our presentation. This work luckily gains good progress during the revision.

---

### Official Review · Reviewer_ud7D · 2024-11-03

**Soundness:** 2
**Presentation:** 3
**Contribution:** 3
**Rating:** 5
**Confidence:** 2

**Summary:**

This paper aims to enhance the theoretical understanding of "Prefix Learning" in Transformer-based models. The authors use the Neural Tangent Kernel (NTK) framework to guarantee convergence when training ultra-long prefixes, essentially addressing the theoretical limits of prefix length. They propose a new NTK-Attention algorithm that approximates infinite-long prefix learning using a few extra trainable parameters per layer, thus reducing computational complexity and memory usage. Experimental results show that NTK-Attention achieves competitive performance compared to methods like P-Tuning V2 and LoRA on datasets involving vision, natural language understanding, and mathematical inference.

**Strengths:**

Theoretical Contribution: The paper provides a solid theoretical analysis based on NTK to understand the efficiency and convergence of Prefix Learning, which deepens the theoretical foundation of this area.
Efficient Training: The NTK-Attention algorithm significantly reduces computational complexity and memory requirements by replacing ultra-long prefixes with a small number of additional trainable parameters.
Experimental Validation: The authors validate the performance of NTK-Attention on diverse datasets (vision, natural language understanding, and math inference) and demonstrate superior or competitive results compared to existing methods.

**Weaknesses:**

Lack of Practical Applicability of Theoretical Analysis: While this paper presents an NTK-based analysis to establish the theoretical foundation of Prefix Learning, there is a lack of empirical evaluation regarding its applicability to large-scale models. In particular, there is little discussion on how the approach involving ultra-long prefixes can be practically applied in real-world industrial settings.

Implementation Complexity of NTK-Attention: The NTK-Attention algorithm introduces a small number of additional trainable parameters to replace ultra-long prefixes, but this also brings about implementation complexity and the need for additional hyperparameter tuning. This could limit the method's applicability and make it challenging to ensure model stability in practical applications.

Limited Experimental Scope: The experiments validate the performance of NTK-Attention on vision, natural language, and mathematical inference datasets, but the scope of the experiments is limited. In particular, there is a lack of validation regarding scalability with very large language models and performance in diverse real-world scenarios, leaving questions about the generalizability of the proposed method. Additional evaluations on a wider range of datasets and real-world applications are needed.

**Questions:**

see weakness

---

> ### Author Response · Authors · 2024-11-22
>
> We sincerely thank you for your reviews and helpful suggestions. We address your concerns as follows:
>
> ### W1: Lack of Practical Applicability of Theoretical Analysis
>
> Thank you for your question. We acknowledge that a very large setting of prefix length $m$ is impractical in prompt-based parameter-efficient-fine-tuning methods. However, in in-context learning (ICL), chain-of-thought (CoT), and LLMs as an agent, an ultra-long prompt as the prefix for input is quite common. We refer the reviewer to **Reclaim the Practicality of Assumption towards $m \gg L$ and $m \gg d$** section in the Global Response to see more practical examples that support the rationality of our assumption in this paper.
>
> ### W2: Implementation Complexity of NTK-Attention
>
> Thank you for your question. We would like to argue that the implementation is not complex, as seen in our pseudo code Line 390-404, eg, compared to prefix attention, and also the PyTorch code Line 1291-1311.
>
> On the other hand, we would like to clarify that a main contribution of our NTK-Attention is improving the complexity from a prompt-based fine-tuning method, e.g. P-Tuning V2. Currently, P-Tuning V2 and Prefix Tuning are gaining popularity in industries, as these methods also introduce extra parameters and computational complexity, but may not affect themselves to become popular.
>
> ### W3: Limited Experimental Scope
>
> Thank you for suggesting this. We add new experiments to extend the experimental scope to relieve your issue, which is regarding scalability with large language models in real-world scenarios. It shows that our methods are effective under a wide range of model sizes. We refer the reviewer to the **Supplementary Experiments** section in Global Response to see the details and results of our new experiments.

---

> > ### Comment · Reviewer_ud7D · 2024-11-27
> > **Thank you for the response**
> >
> > I appreciate the authors' efforts in addressing my concerns. After reviewing the clarifications provided, I have decided to retain my previous score.

---

> > > ### Author Response · Authors · 2024-11-27
> > > **Thank you**
> > >
> > > We appreciate the reviewer for the time and effort involved in the review. If the reviewer has any future concerns, we are willing to address them by using the extended discussion deadline. We thank the reviewer's valuable suggestion again.

---

### Official Review · Reviewer_LAin · 2024-11-03

**Soundness:** 3
**Presentation:** 3
**Contribution:** 3
**Rating:** 6
**Confidence:** 5

**Summary:**

This paper explores “Prefix Learning,” a method that enhances large language models (LLMs) by fine-tuning an added prefix matrix in each layer of a transformer. The authors investigate the effect of infinitely long prefixes on model performance through theoretical analysis using the Neural Tangent Kernel (NTK) framework. Based on this theoretical insight, they propose “NTK-Attention,” an efficient approximation algorithm that replaces the infinite prefix matrix with a finite number of trainable parameters. This method achieves polynomially small errors while significantly reducing computational and memory costs.

**Strengths:**

1. Strong Theoretical Foundation: The paper provides a rigorous theoretical analysis of prefix learning with ultra-long prefixes, leveraging the Neural Tangent Kernel (NTK) framework. This contributes a new perspective to understanding prefix-based methods in transformers, especially regarding convergence and scaling laws.
2. Efficient Fine-Tuning Method: The proposed NTK-Attention significantly reduces the number of trainable parameters and computational complexity by replacing the large prefix matrix with two smaller trainable parameters,  Z  and  k . This approach is practical and resource-efficient, addressing scalability issues that arise with traditional long prefix methods.
3. High Performance Across Multiple Domains: Experimental results demonstrate NTK-Attention’s competitive or superior performance on tasks in natural language understanding, vision, and mathematical inference. This versatility and effectiveness across diverse datasets and tasks underscore its broad applicability.
4. Reduced Memory and Computational Costs: NTK-Attention achieves considerable savings in memory and computation compared to standard prefix learning methods. This efficiency is particularly beneficial for large-scale models where memory and computational constraints are a concern.
5. Reproducibility: Although the code is not included in the OpenReview submission, the arXiv version of the paper provides the accompanying code, enhancing reproducibility.

**Weaknesses:**

1.Incomplete Evaluation Across Different Model Architectures: The experiments presented focus primarily on a few specific architectures, such as pretrained ViT and ChatGLM3-6B. The generalizability of NTK-Attention across a broader range of transformer architectures and model sizes remains unexplored, which limits the applicability of the findings to other models commonly used in different domains​.
2. Limited Applicability of NTK-Attention’s Efficiency Claims: The reduced computational complexity of NTK-Attention compared to Prefix Attention relies on the condition  m \gg L , where  m  is the prefix length and  L  is the input length. However, such conditions may not be common in many real-world scenarios, which restricts the practical applications of NTK-Attention to specific cases.
3. Inappropriate Comparison with Prefix Attention: The paper’s comparison with Prefix Attention may be less relevant, as NTK-Attention does not introduce additional prefill tokens and is structurally closer to LoRA. From Table 1, NTK-Attention achieves comparable performance to LoRA, but it introduces additional inference costs that LoRA does not have. Given these trade-offs, LoRA may provide better practical value, limiting NTK-Attention’s advantage over existing methods.

**Questions:**

1. Sensitivity to Hyperparameters: The paper does not extensively cover the sensitivity of NTK-Attention to key hyperparameters, such as the dimension of the feature mapping  r  and the learning rates for  Z  and  k . Could the authors provide an ablation study or guidelines for tuning these parameters? This would assist researchers in optimizing NTK-Attention across different model architectures and tasks.
2. Applicability of the  m >> L  Assumption: The computational efficiency of NTK-Attention compared to Prefix Attention assumes  m >> L , which may not hold in many real-world scenarios. Could the authors provide more context or examples where this assumption is practical? Alternatively, could they discuss how NTK-Attention performs in settings where m and L are closer in scale?
3. Justification for Comparison with Prefix Attention: Given that NTK-Attention does not introduce additional prefill tokens, it is more similar to LoRA than Prefix Attention. Could the authors clarify why they chose to emphasize the comparison with Prefix Attention rather than focusing on comparisons with methods like LoRA? This would help align the experimental evaluations with the method’s structural similarities.
4. Additional Inference Costs Compared to LoRA: NTK-Attention introduces additional inference costs, as highlighted in Table 1, where its performance is comparable to LoRA. Could the authors elaborate on scenarios where the increased inference cost of NTK-Attention would be justified over LoRA? This could clarify the practical advantages of NTK-Attention in resource-constrained environments.

---

> ### Author Response · Authors · 2024-11-22
>
> We sincerely thank you for your reviews and helpful suggestions. We address your concerns as follows:
>
> ### W1: Incomplete Evaluation Across Different Model Architectures
>
> Thank you for pointing this out. We state additional experiments across LLAMA and OPT architecture to relieve this issue, which shows that our methods are effective on these model architectures. We refer the reviewer to the **Supplementary Experiments** section in the Global Response to see the details and results of our new experiments.
>
> ### W2: Limited Applicability of NTK-Attention’s Efficiency Claims
>
> Since we confirm the strong learning ability of an ultra-long prefix learning in Section 3, the proposition of NTK-Attention is how to catch this strong ability efficiently. NTK-Attention provides a probability for the longer extension of prefix length in prefix-based fine-tuning methods, while only costing few computational complexities (see Figure 3) and fewer trainable parameters than P-Tuning V2 when $m$ is large (see Table 1, 2, 3, 4 in revision). In short, we have compatible or better performance than LoRA, when we share the same number of trainable parameters. We refer the reviewer to the Global Response section **Supplementary Implementation Detail** for more details.
>
>
> ### W3 and Q3: Inappropriate Comparison with Prefix Attention
>
> Thank you for pointing out your concern. Carefully following the derivation in Section 4.1, the NTK-Attention method comes up from introducing a polynomial attention trick to modify the prefix attention computation. It needs to be clarified that by following Section 4.1, Section 4.2, and Section 4.3, our NTK-Attention is more similar to prefix attention both in mathematics and architecture instead of LoRA.
>
> Moreover, prefix attention introduces extra parameters and computational complexity as we state in Figure 1, while our NTK-Attention enhances it from $mL+L^2$ to $L^2$ in complexity, from $md$ to $rd+r$ in the number of trainable parameters.
>
> ### Q1: Sensitivity to Hyperparameters
>
> Thanks for your suggestion; we add an ablation study on the hyper-parameters of NTK-Attention in the revision. We refer the reviewer to the **Supplementary Experiments** section of Global Response to see the details and results of the ablation study.
>
>
> ### Q2: Applicability of the m >> L Assumption
>
> In fact, in the next-token-prediction mechanism, in order to improve the models’ outputs, all previous tokens can be considered as a prefix of the language model. Hence, in this situation, how to find an optimal prefix for the best next-token prediction becomes a considerable problem that is interesting to study. In particular, the above is only one case that Prefix Learning can stand for, we recommend the reviewer check **Reclaim the Practicality of Assumption towards $m \gg L$ and $m \gg d$** section in Global Response to see more practical examples that support the rationality of our assumption in this paper.
>
> ### Q4: Additional Inference Costs Compared to LoRA
>
> Thank you for pointing this out. Since our NTK-Attention is more similar to prefix attention, the born disadvantage of prefix attention - extra parameters and computational complexity - is unavoidable. However, our NTK-Attention still makes a great improvement compared to traditional prompt-based fine-tuning methods (see rebuttal to **W3 and Q3: Inappropriate Comparison with Prefix Attention**) and demonstrates high performance compared to LoRA, full fine-tuning, and P-Tuning V2 (see Table 1, 2, 3). We believe that it is worth using such complexity in exchange for.

---

> > ### Comment · Reviewer_LAin · 2024-11-27
> > **Acknowledgment of Response**
> >
> > Thank you to the authors for their efforts in addressing my concerns. After reviewing the clarifications provided, I have decided to maintain my previous score.

---

> > > ### Author Response · Authors · 2024-11-27
> > > **Thank you**
> > >
> > > We sincerely thank your valuable suggestions and comments. We appreciate your time and positive feedback.

---

### Official Review · Reviewer_jxUA · 2024-11-04

**Soundness:** 2
**Presentation:** 3
**Contribution:** 2
**Rating:** 6
**Confidence:** 2

**Summary:**

The paper proposes NTK-Attention, an alternative to Prefix-Tuning method for Transformer fine-tuning. Specifically, NTK-Attention is derived from theoretical understanding that Prefix-Tuning can be approximated using Neural Tangent Kernel (NTK) when the prefix length is sufficiently large. The key difference of NTK-Attention is to utilize a learnable weight matrix multiplied to Query, instead of concatenating a learnable prefix to Key and Value. Due to the architecture of NTK-Attention, it can avoid quadratic memory increase as the prefix length increases. Experimental results show that NTK-Attention achieves comparable performance to competitors, P-Tuning-v2 and LoRA.

**Strengths:**

* The proposed NTK-based analysis is novel and opens a new direction to understand prefix-tuning. Although the extremely-long prefix setup is not widely used yet, it seems to be an interesting direction for Transformer-based models.

**Weaknesses:**

* Is the "infinite-long" or "sufficiently long" prefix assumption practical (i.e., beneficial in terms of the performance)? In Table 1, m=200 is worse than m=100 case. The "Many-shot In-context Learning" paper (Agarwal et al., 2024) empirically showed that using lots of few-shot examples help, but the paper increases the length of input prompt rather than the number of training parameters; the result would not directly mapped one-to-one.
* Experimental results (Tables and Figures) should indicate the number of trainable parameters for each configuration. To be fair, it is important to compare the performance under similar resource usage constraint.

**Questions:**

* Theoretical questions
  * Does NTK-Attention still theoretically supported for the small m (~=100) case? How about the performance of NTK-Attention for small m, in terms of the number of training parameters?
  * I am not sure if the theoretical derivation also applies to L >> 1 case and causal attention setup, which is the most common case for Transformer-based generative models.
* How should we determine the size "r" of NTK-Attention?

---

> ### Author Response · Authors · 2024-11-22
> **Official Comment by Authors (Part 1)**
>
> We sincerely thank you for your reviews and helpful suggestions. We address your concerns as follows:
>
> ### W1: Is the "infinite-long" or "sufficiently long" prefix assumption practical (i.e., beneficial in terms of the performance)?
>
> Thank you for your question. Our paper considers the learning setting when $m \to \infty$, and then approximates the $m = \infty$ learning setting with NTK-attention. Thus, our NTK-attention method is not approximating small $m$.
>
> As we claimed in Section 2, Line 174, prior works [1, 2, 3, 4] practically discovered the benefit of scaling the prefix length up in prompt-based methods, where all of them could be abstract as a learning problem of prefix matrix as we stated in Eq. (2). We clarify that the larger $m$ would provide the transformer-based model with strong learning ability, whereas it doesn’t guarantee the superior performance on generalization, the model can be over-fitted on training set to perform not well on evaluation set. Furthermore, scaling law requires comprehensive high-quality datasets since models can master complicated skills from training on them due to their strong learning capabilities under scaling law, but it may not mean that we can enhance performance by only improving the model size [5].
>
> In short, large $m$ means better capacity, but this capacity may not be fully exploited due to different learning algorithms. On the other hand, our method does have convergence guarantees which means it can exploit the capacity and typically shows better performance.
>
>
> ### W2: Experimental results (Tables and Figures) should indicate the number of trainable parameters for each configuration.
>
> Thank you for pointing this out! We have updated Table 1 and Table 2 in the pdf file of our revision indicating the number of parameters of each method in our experiments, where we can change the number of parameters by adjusting $r$ and $s$, which is defined in Line 361-370. For the new experiments we add during revision, we also clearly claim the number of parameters of each method for fair comparison.
>
> In short, we have compatible or better performance than LoRA, when we share the same number of trainable parameters. We refer the reviewer to the Global Response section **Supplementary Implementation Detail** for more details.
>
> ### Q1.1: Does NTK-Attention still theoretically supported for the small m (~=100) case? How about the performance of NTK-Attention for small m, in terms of the number of training parameters?
>
> As we claimed in **W1**, our paper considers the learning setting when $m \to \infty$, and then approximates the $m = \infty$ learning setting with NTK-attention. Thus, our NTK-attention method is not approximating small $m$. Also, we would like to clarify that $m$ is not a parameter of NTK-Attention but prefix attention. Only the choices of $r$ and $s$ (see Global Response section **Supplementary Implementation Detail**) will affect the number of trainable parameters in NTK-Attention.
>
> ### Q1.2: I am not sure if the theoretical derivation also applies to L >> 1 case and causal attention setup, which is the most common case for Transformer-based generative models.
>
> Thank you for pointing this concern out. In fact, assuming $L=1$ in Section 3.1 is for simplifying the setting of our analysis, it helps us to consider only the learning of one token query state (Q) in attention. We are able to extend our theoretical results to the $L \gg 1$ case easily by the NTK analysis framework. We are willing to provide more discussion per the reviewer’s request.
>
> ### Q2: How should we determine the size "r" of NTK-Attention?
>
> It depends on the performance of NTK-Attention we aim to achieve and the computational resource we have for parameter-efficient-fine-tuning. Theorem 4.1 literally says that the larger choice of the value of $r$ will lead to NTK-Attention approximating prefix attention with ultra-long prefix length with small errors. However, larger $r$ would also bring requirements of larger computational complexity, which may be difficult to implement in some computational resource-limited situations.
>
> The choice of $r$ is based on the degree we choose for the Taylor expansion of the exponential function. When choosing $r \in [d, d^2)$, function $\phi(Q_i)^\top \phi(K_j)$ is first-order approximation to $\exp(Q_i^\top K_j / \sqrt{d})$ (see Eq. (6)), when choosing $r \in [d^2, d^3)$, function $\phi(Q_i)^\top \phi(K_j)$ is second-order approximation to $\exp(Q_i^\top K_j / \sqrt{d})$, etc. We provide the detailed implementation of the function $\phi()$ in our supplementary code as well.

---

> ### Author Response · Authors · 2024-11-22
> **Official Comment by Authors (Part 2)**
>
> ### References
>
> [1] Xiao Liu, Kaixuan Ji, Yicheng Fu, Weng Lam Tam, Zhengxiao Du, Zhilin Yang, and Jie Tang. P-tuning v2: Prompt tuning can be comparable to fine-tuning universally across scales and tasks. ACL’22.
>
> [2] Brian Lester, Rami Al-Rfou, and Noah Constant. The power of scale for parameter-efficient prompt tuning. ACL’21.
>
> [3] Rishabh Agarwal, Avi Singh, Lei M Zhang, Bernd Bohnet, Stephanie Chan, Ankesh Anand, Zaheer Abbas, Azade Nova, John D Co-Reyes, Eric Chu, et al. Many-shot in-context learning. NeurIPS’24.
>
> [4] Yao Fu, Hao Peng, Ashish Sabharwal, Peter Clark, and Tushar Khot. Complexity-based prompting for multi-step reasoning. ICLR’22.
>
> [5] Jared Kaplan, Sam McCandlish, Tom Henighan, Tom B Brown, Benjamin Chess, Rewon Child, Scott Gray, Alec Radford, Jeffrey Wu, and Dario Amodei. Scaling laws for neural language models. OpenAI’20.

---

> > ### Comment · Reviewer_jxUA · 2024-11-23
> > **Thank you for the rebuttal**
> >
> > Thank you for the authors's response.
> >
> > The rebuttal addressed my concerns regarding the number of parameters, the L=1 setup, and the selection of "r". In particular, the revised manuscript clarified the architecture of NTK attention that it also equips low-rank architecture, significantly reducing the expected number of parameters.
> >
> > On the other hand, the authors pointed out that "our NTK-attention method is not approximating small m" and "small m case is prefix attention, not NTK-attention".
> > I understand that extremely large m case is what NTK attention aims for, and ICL and CoT are promising directions.
> > I also agree that, if m is sufficiently long, NTK would help.
> > However, ICL and CoT cases would use the real input sequences that are task-specific or query-specific, which is quite flexible and maybe better than fixed parameters.
> > In this perspective, I think infinite-long ICL and CoT are somewhat different from infinite-long "trainable" prefixes...
> >
> > If NTK-Attention is just an efficient PEFT architecture that provides more performance improvements than LoRA, then it is okay, and we can discuss the well-known problems, such as catastrophic forgetting, sensitivity to training data selection, etc.
> > However, if the paper claims this is an efficient approximation of infinite-long prefix learning, I think we should also discuss the practical point of infinite-long prefix and whether their advantages still holds for NTK-attention.

---

> > > ### Author Response · Authors · 2024-11-24
> > > **Thank you and further reply to new concerns**
> > >
> > > We are glad that our reply has addressed some concerns from the reviewer! We sincerely thank you for your new response. We appreciate your time and we would like to fix your further concern.
> > >
> > > ### Q1: ICL and CoT
> > > The reviewer may like to check whether the NTK-attention is good for ICL/CoT. However, we would like to point out that our algorithm design and evaluation focus on PEFT tasks. ICL/CoT are more like extra application scenarios that provide further motivation justifying long prefix learning, but are not our focus.
> > >
> > > First, we would like to reclaim our main focus: NTK-attention approximates long prefix learning and can exploit the power of long prefixes (stronger than the sort prefixes) thanks to the convergence guarantee. This allows it to be an efficient PEFT method improving over existing prefix learning, which provides the main motivation and guides our evaluation.
> > >
> > > Second, we would like to clarify why ICL/CoT provides motivation for long prefix learning. People need to design a considerably good prompt for ICL or CoT on some specific datasets/tasks, while p-tuning or prefix tuning is also task-specific and requires training its virtual/prefix tokens. Also, ICL or CoT may rely on manual design and step-by-step search iteration to find the optimal prompt, while P-Tuning or Prefix-Tuning relies on gradient descent to optimize the prefix. We believe that the essence of both is to optimize the prefix in LLMs. The difference is that P-Tuning and Prefix-Tuning are flexible because they can train continuous parameters, while ICL and CoT can only optimize discrete tokens. Hence, this is the reason we use it as a motivation for long prefix learning.
> > >
> > > Furthermore, we share a similar motivation as CNTK [1], they give the first approximate algorithm that extends NTK in CNN and found a strong baseline for CNN in vision tasks, while our work gives an NTK-approximating method in attention from the perspective of scaling prefix length and make the prompt-based PEFT currently competitive to SOTA method like LoRA and full-parameters fine-tuning.
> > >
> > > In short, ICL/CoT can be a good potential for applying NTK-attention for such applications but they are not the focus of this paper.
> > >
> > > ### Q2: If the paper claims this is an efficient approximation of infinite-long prefix learning, I think we should also discuss the practical point of infinite-long prefix and whether their advantages still hold for NTK-attention.
> > >
> > > As we emphasized in Section 3.3, Line 270-294 in revision, the practical point of our NTK-Attention is we confirmed the existence of *scaling law in prefix learning*. Therefore, the usage of such an infinite-long prefix represents a strong learning capability and it gains a similar property as scaling law in language models - large model size, more comprehensive high-quality datasets, and longer training computes are leading the model to perform better. It is precisely because we have theoretically proven this that approximating infinite long prefix attention practically has become interesting and worthy of further investigation. At the same time, our NTK-Attention provides a possible solution for this approximation.
> > >
> > > We hope our answer may address your concern and we are willing to discuss more per reviewer requests.
> > >
> > > ### References
> > >
> > > [1] Sanjeev Arora, Simon S Du, Wei Hu, Zhiyuan Li, Russ R Salakhutdinov, and Ruosong Wang. On exact computation with an infinitely wide neural net. NeurIPS’19.

---

> > > > ### Comment · Reviewer_jxUA · 2024-11-25
> > > > **Thank you for the response**
> > > >
> > > > Thank you for clarifying the paper's motivation and direction.
> > > > Whether or not I agree with the importance of the long prefix, the claim seems to be logically structured, so I increased my score from 5 to 6.
> > > >
> > > > I think the discussion now regresses to the correctness of the derivation and generalizability of the assumptions. As I am not an expert to NTK (so my confidence is 2), I hope other reviewers verify the paper thoroughly...

---

> > > > > ### Author Response · Authors · 2024-11-25
> > > > > **Thank you**
> > > > >
> > > > > We are glad that our response addressed your concerns and thank you for your thoughtful feedback. We appreciate your valuable time and increasing score!

---

### Author Response · Authors · 2024-11-22
**Global Response (Part 2)**

### Reclaim the Practicality of Assumption towards $m \gg L$ and $m \gg d$

Since Reviewer LAin, ud7D, and oFkK mentioned their concerns about the practicality of prefix length $m \rightarrow +\infty$ assumption, we here explain and address as follows:

We acknowledge that a very large setting of prefix length $m$ is impractical in prompt-based parameter-efficient-fine-tuning methods. However, in in-context learning (ICL), chain-of-thought (CoT), and LLMs as an agent, an ultra-long prompt as the prefix for input is quite common. In this paper, we examine the implications of using a significantly large prefix length, denoted as $m \gg L$ and $m \gg d$, which is prevalent across various prompt-based methods. The primary objective of Prefix Learning is to enhance the LLMs' outputs by identifying an advanced prefix during the generation process. For instance, the search for optimal example pairs to improve ICL and the development of prompt engineering tailored for agent frameworks to address specific task requirements often necessitate the use of exceptionally long prefixes. Moreover, given the modern application demands related to long-context scenarios, optimizing previous tokens to improve next-token prediction can be framed as a prefix optimization problem. Thus, a thorough investigation into the optimization of infinitely long prefixes is essential for understanding the theoretical significance of the prefix matrix in LLMs.

### Clarification of notation

We found that the $r$ in LoRA is reused with the $r$ notation in our NTK Attention. We re-denote the $r$ in LoRA as $r’$ in the revision.


### References

[1] Susan Zhang, Stephen Roller, Naman Goyal, Mikel Artetxe, Moya Chen, Shuohui Chen, Christopher Dewan, Mona Diab, Xian Li, Xi Victoria Lin, et al. Opt: Open pre-trained transformer language models. Meta’22.

[2] Stephen Merity, Caiming Xiong, James Bradbury, and Richard Socher. Pointer sentinel mixture models. ICLR’17.

[3] Mitch Marcus, Beatrice Santorini, and Mary Ann Marcinkiewicz. Building a large annotated corpus of english: The penn treebank. Computational linguistics’93.

[4] Denis Paperno, Germán Kruszewski, Angeliki Lazaridou, Quan Ngoc Pham, Raffaella Bernardi, Sandro Pezzelle, Marco Baroni, Gemma Boleda, and Raquel Fernández. The lambada dataset: Word prediction requiring a broad discourse context. ACL’16.

[5] Edward J Hu, Yelong Shen, Phillip Wallis, Zeyuan Allen-Zhu, Yuanzhi Li, Shean Wang, Lu Wang, and Weizhu Chen. Lora: Low-rank adaptation of large language models. ICLR’22.

[6] Xiao Liu, Yanan Zheng, Zhengxiao Du, Ming Ding, Yujie Qian, Zhilin Yang, and Jie Tang. Gpt understands, too. AI Open’23.

[7] Xiao Liu, Kaixuan Ji, Yicheng Fu, Weng Lam Tam, Zhengxiao Du, Zhilin Yang, and Jie Tang. P-tuning v2: Prompt tuning can be comparable to fine-tuning universally across scales and tasks. ACL’22.

---

### Author Response · Authors · 2024-11-22
**Global Response (Part 1)**

We thank all constructive and valuable reviews from all reviewers.

We are grateful that reviewers all agree that our NTK analysis in Transformers is novel and theoretically solid, contributing considerably to understanding prefix learning in pretrained transformers-based architectures. Besides, reviewers (LAin, ud7D) admit the potential of our NTK-Attention as a parameter-efficient-fine-tuning method that approximates that ultra-long prefix attention computation and demonstrates superior empirical results.


We have updated a **revision** for our draft. We summarize all the major updates (in **brown** color) we made in the revision. All line numbers in the rebuttal correspond to the revised version.
- Line 102-107: we supplement to introduce new experiments we add in the revision.
- Line 188-197: we add an explanation for the practicality of infinite-long prefix length assumption $m \gg L$ and $m \gg d$
- Line 367-394: we reclaim the implementation detail of our method, and add an analysis of the number of trainable parameters compared to other popular methods.
- Line 381-388: we clarify the number of trainable parameters of each fine-tuning method in Table 1
- Line 440-441: we clarify the setting of hyper-parameters we use for fine-tuning ChatGLM3-6B with NTK-Attention
- Line 459-467 & 486-505: we add a detailed instruction for new experiments and we state the results of new experiments in Table 3
- Line 474: we clarify the setting of hyper-parameters we use for fine-tuning LLAMA with NTK-Attention on math inference tasks
- Line 489-495: we clarify the number of trainable parameters of each fine-tuning method in Table 2
- Line 520-523 & 1253-1288: we add detailed instructions for the ablation study and we provide the results of the ablation study in Table 4
- Line 1291-1311: we provide a 10-line naive implementation of NTK-Attention by Python code to support the implementation simplicity of our method.

Here, we address some common questions.

### Supplementary Implementation Detail
We used low-rank adaptation to achieve an efficient parameter representation of $Z$. In detailed, we decompose $Z(0) = Z_A(0) \cdot Z_B(0)$, where $Z_A(0) \in \mathbb{R}^{r \times s}$ and $Z_B \in \mathbb{R}^{s \times d}$. We choose $s \leq \lfloor d/2 \rfloor$ as an appropriately small integer, in Line 361-370 of the revision.

Since given $r$ and $s$ as two hyper-parameters in NTK-Attention, for each attention layer in transformer-based architecture, we denote $\beta:= \frac{r}{d}$, then the number of trainable parameters could be computed by $(\beta s+ \beta + s)d$ where integer $\beta \ge 1$ and $s \leq \lfloor d/2 \rfloor$. This is more flexible when adjusting the practical efficiency needs. For LoRA with its hyper-parameter $r' \leq \lfloor d/2 \rfloor$, where $r'$ is the rank number used for approximation, its number of trainable parameters is $4r'd$ and for prefix attention with its hyper-parameter $m \ge 1$, its number of trainable parameters is $md$ in each attention layer. By choosing $(\beta s+ \beta + s) \leq 4r'$, the higher efficiency of NTK-Attention compared to LoRA will be satisfied. The above comparison is also in Line 371-377 of the revision.

In our first submitted manuscript, for the convenience of readers' understanding and to make the paper focus on our main message of theoretical analysis, we did not disclose the specific implementation of a trainable matrix $Z$, but the details we claim below can be found in the code of the supplementary materials in our first submitted version.

### Supplementary Experiments

Since Reviewer LAin, ud7D, and oFkK mentioned our limitation on the experimental scope, we add two more experiments to support our work.

**Evaluation on Language Modeling Tasks.** In this experiment, we focus on the scalability of NTK-Attention on a family of language models of different sizes, the OPT family with the model sizes 125M, 350M, 1.3B, 2.7B and 6.7B [1]. We introduce three text datasets, which are WikiText-103 [2], Penn TreeBank [3], and LAMBADA [4], to compare the scalability of NTK-Attention with LoRA [5] and P-Tuning V2 [6, 7]. As we choose $r'=8$ for LoRA, $m=32$ for P-Tuning V2, and $r=2d$ and $s = 10$ for our NTK-Attention, the numbers of trainable parameters are aligned to the same as $32d$ for each attention layer. The results are stated in Table 3, which shows the improvement of NTK-Attention compared to baselines when scaling the model size.

**Ablation Study.** We validate the sensitivity of hyper-parameters $r$ and $s$ and give the results in Appendix B.3. The results firstly indicate that choosing $r=d$ and $s=4$ is enough for high-performance fine-tuning on LLAMA-3.1-8B. Also, we follow Table 4 to suggest choosing a larger value of $r$ primarily instead of $s$ to achieve supernal accuracy.

---

### Author Response · Authors · 2024-11-28
**Second version revision update**

Per reviewer oFkK valuable suggestions, we made a second revision, where the update is in the color **purple** (we also keep our old updates). We summarize all major updates below. Note that all line numbers in the rebuttal correspond to the second revision version.

- Line 54-70: we modify Figure 1 and its caption to clarify the complexity and number of parameters of our NTK-Attention as reviewer oFkK requires.
- Line 94-95: we modify the introduction for low-rank approximation to matrix $Z$.
- Line 1140-1182: we update the detailed version of NTK-Attention with low-rank approximation to matrix $Z$ and update the newest complexity and the memory usage of trainable parameters.

---

### Meta-Review · Area_Chair_YeSt · 2024-12-20

**Metareview:**

The paper addresses the theoretical underpinnings and practical applications of Prefix Learning in enhancing the performance of large language models. By utilizing the Neural Tangent Kernel (NTK) framework, the study provides a theoretical guarantee for the convergence when training an ultra-long prefix, which was previously not well-understood. Some reviewers questioned the practicality of the assumption of ultra-long prefixes and the complexity of implementing the proposed NTK-Attention, suggesting that these aspects may limit the method’s real-world applicability.  There are concerns about the fairness and adequacy of the comparative analysis with existing methods. Specifically, details about parameter counts and computational efficiency need clearer exposition to validate the claimed advantages. AC agrees with reviewers, while this work presents significant theoretical contributions and innovative methods, requires more rigorous empirical support and a clearer exposition of its comparative advantages over existing methods.

**Additional Comments On Reviewer Discussion:**

The authors have actively responded to reviewer feedback by conducting further experiments to verify the applicability of NTK-Attention in typical Transformer scenarios, broadening the testing across various architectures such as LLAMA and OPT, and substantiating parameter efficiency through a low-rank adaptation strategy while revealing detailed parameter counts in their experiments. The decision to reject was influenced by weighing the theoretical innovations against practical implementation challenges and insufficient empirical validation. Concerns remain about the method’s computational efficiency and the validity of its claimed benefits under typical usage conditions, where the prefix and input lengths are not excessively large. Additionally, some aspects of the work related to the design of $Z$ and $P$ have not been clearly articulated in the current version of the manuscript.

---

### Decision · Program_Chairs · 2025-01-22

Reject